# Generalizability of Memorization Neural Networks

**Lijia Yu[1], Xiao-Shan Gao[2,3,*], Lijun Zhang[1,3], Yibo Miao[2,3]**
[1] Key Laboratory of System Software (Chinese Academy of Sciences)
and State Key Laboratory of Computer Science, Institute of Software, Chinese Academy of Sciences
[2] Academy of Mathematics and Systems Science, Chinese Academy of Sciences
Beijing 100190, China
[3] University of Chinese Academy of Sciences, Beijing 100049, China

## Abstract

The neural network memorization problem is to study the expressive power of neural networks to interpolate a finite dataset. Although memorization is widely believed to have a close relationship with the strong generalizability of deep learning when using over-parameterized models, to the best of our knowledge, there exists no theoretical study on the generalizability of memorization neural networks. In this paper, we give the first theoretical analysis of this topic. Since using i.i.d. training data is a necessary condition for a learning algorithm to be generalizable, memorization and its generalization theory for i.i.d. datasets are developed under mild conditions on the data distribution. First, algorithms are given to construct memorization networks for an i.i.d. dataset, which have the smallest number of parameters and even a constant number of parameters. Second, we show that, in order for the memorization networks to be generalizable, the width of the network must be at least equal to the dimension of the data, which implies that the existing memorization networks with an optimal number of parameters are not generalizable. Third, a lower bound for the sample complexity of general memorization algorithms and the exact sample complexity for memorization algorithms with constant number of parameters are given. It is also shown that there exist data distributions such that, to be generalizable for them, the memorization network must have an exponential number of parameters in the data dimension. Finally, an efficient and generalizable memorization algorithm is given when the number of training samples is greater than the efficient memorization sample complexity of the data distribution.

## 1 Introduction

Memorization is to study the expressive power of neural networks to interpolate a finite dataset [9]. The main focus of the existing work is to study how many parameters are needed to memorize. For any dataset $\mathcal{D}_{tr}$ of size $N$ and neural networks of the form $\mathcal{F} : \mathbb{R}^n \to \mathbb{R}$, memorization networks with $\overline{O}(N)$ parameters have been given with various model structures and activation functions [31, 50, 30, 29, 26, 47, 56, 11, 65]. On the other hand, it is shown that in order to memorize an arbitrary dataset of size $N$ [64, 56], the network must have at least $\overline{\Omega}(N)$ parameters, so the above algorithms are approximately optimal. Under certain assumptions, it is shown that sublinear $\overline{O}(N^{2/3})$ parameters are sufficient to memorize $\mathcal{D}_{tr}$ [49]. Furthermore, Vardi et al. [55] give a memorization network with optimal number of parameters: $\overline{O}(\sqrt{N})$.

Recently, it is shown that memorization is closely related to one of the most surprising properties of deep learning, that is, over-parameterized neural networks are trained to nearly memorize noisy data and yet can still achieve a very nice generalization on the test data [45, 7, 4]. More precisely, the

---

[*] Corresponding author.

double descent phenomenon [45] indicates that when the networks reach the interpolation threshold, larger networks tend to have more generalizability [41, 10]. It is also noted that memorizing helps generalization in complex learning tasks, because data with the same label have quite diversified features and need to be nearly memorized [19, 20]. A line of research to harvest the help of memorization to generalization is *interpolation learning*. Most of recent work in interpolation learning shows generalizability of memorization models in linear regimes [7, 12, 38, 53, 59, 66].

As far as we know, the generazability of memorization neural networks has not been studied theoretically, which is more challenging compared to the linear models, and this paper provides a systematic study of this topic. In this paper, we consider datasets that are sampled i.i.d. from a data distribution, because i.i.d. training dataset is a necessary condition for learning algorithms to have generalizability [54, 44]. More precisely, we consider binary data distributions $\mathcal{D}$ over $\mathbb{R}^n \times \{-1, 1\}$ and use $\mathcal{D}_{tr} \sim \mathcal{D}^N$ to mean that $\mathcal{D}_{tr}$ is sampled i.i.d. from $\mathcal{D}$ and $|\mathcal{D}_{tr}| = N$. All neural networks are of the form $\mathcal{F} : \mathbb{R}^n \to \mathbb{R}$. The main contributions of this paper include four aspects.

First, we give the smallest number of parameters required for a network to memorize an i.i.d. dataset.

**Theorem 1.1** (Informal. Refer to Section 4). *Under mild conditions on $\mathcal{D}$, if $\mathcal{D}_{tr} \sim \mathcal{D}^N$, it holds*

*(1) There exists an algorithm to obtain a memorization network of $\mathcal{D}_{tr}$ with width 6 and depth $\overline{O}(\sqrt{N})$.*

*(2) There exists a constant $N_{\mathcal{D}} \in \mathbb{Z}_+$ depending on $\mathcal{D}$ only, such that a memorization network of $\mathcal{D}_{tr}$ with at most $N_{\mathcal{D}}$ parameters can be obtained algorithmically.*

$N_{\mathcal{D}}$ is named as the **memorization parameter complexity** of $\mathcal{D}$, which measures the complexity of $\mathcal{D}$ under which a memorization network with $\leq N_{\mathcal{D}}$ parameters exists for almost all $D_{tr} \sim \mathcal{D}^N$.

Theorem 1.1 allows us to give the memorization network for i.i.d dataset with the optimal number of parameters. When $N$ is small so that $\sqrt{N} \ll N_{\mathcal{D}}$, the memorization network needs at least $\overline{\Omega}(\sqrt{N})$ parameters as proved in [6] and (1) of Theorem 1.1 gives the optimal construction. When $N$ is large, (2) of Theorem 1.1 shows that a constant number of parameters is enough to memorize.

Second, we give a necessary condition for the structure of the memorization networks to be generalizable, and shows that even if there is enough data, memorization network may not have generalizability.

**Theorem 1.2** (Informal. Refer to Section 5). *Under mild conditions on $\mathcal{D}$, if $\mathcal{D}_{tr} \sim \mathcal{D}^N$, it holds*

*(1) Let $\mathbf{H}$ be a set of neural networks with width $w$. Then, there exist an integer $n > w$ and a data distribution $\mathcal{D}$ over $\mathbb{R}^n \times \{-1, 1\}$ such that, any memorization network of $\mathcal{D}_{tr}$ in $\mathbf{H}$ is not generalizable.*

*(2) For almost any $\mathcal{D}$, there exists a memorization network of $\mathcal{D}_{tr}$, which has $\overline{O}(\sqrt{N})$ parameters and is not generalizable.*

Theorem 1.2 indicates that memorization networks with the optimal number of parameters $\overline{O}(\sqrt{N})$ may have poor generalizability, and commonly used algorithms for constructing fixed-width memorization networks have poor generalization for some distributions. These conclusions demonstrate that the commonly used network structures for memorization is not generalizable and new network structures are needed to achieve generalization.

Third, we give a lower bound for the sample complexity of general memorization networks and the exact sample complexity for certain memorization networks.

**Theorem 1.3** (Informal. Refer to Section 6). *Let $N_{\mathcal{D}}$ be the memorization parameter complexity defined in Theorem 1.1. Under mild conditions on $\mathcal{D}$, we have*

*(1) **Lower bound**. In order for a memorization network of any $\mathcal{D}_{tr} \sim \mathcal{D}^N$ to be generalizable, $N$ must be $\geq \overline{\Omega}(\frac{N_{\mathcal{D}}^2}{\ln^2(N_{\mathcal{D}})})^2$.*

*(2) **Upper bound**. For any memorization network with at most $N_{\mathcal{D}}$ parameters for $\mathcal{D}_{tr} \sim \mathcal{D}^N$, if $N = \overline{O}(N_{\mathcal{D}}^2 \ln N_{\mathcal{D}})$, then the network is generalizable.*

---

[2]Here, $\overline{\Omega}$ and $\overline{O}$ mean that certain small quantities are omitted. Also, we keep the logarithm factor of $N_{\mathcal{D}}$ for comparison with the upper bound

Notice that the lower bound is for general memorization networks and the upper bound is for memorization networks with $\le N_{\mathcal{D}}$ parameters, which always exist by (2) of Theorem 1.1. In the latter case, the lower and upper bounds are approximately the same, which gives the exact sample complexity $\overline{O}(N_{\mathcal{D}}^2)$ in this case. In other words, a necessary and sufficient condition for the memorization network in (2) of Theorem 1.1 to be generalizable is $N = \overline{O}(N_{\mathcal{D}}^2)$.

*Remark* 1.4. Unfortunately, these generalizable memorization networks cannot be computed efficiently, as shown by the following results proved by us.

(1) If $P \ne NP$, then all networks in (2) of Theorem 1.3 cannot be computed in polynomial time.

(2) For some data distributions, an exponential (in the data dimension) number of samples is required for memorization networks to achieve generalization.

Finally, we want to know that does there exist a polynomial time memorization algorithm that can ensure generalization, and what is the sample complexity of such memorization algorithm? An answer is given in the following theorem.

**Theorem 1.5** (Informal. Refer to Section 7). *There exists an $S_{\mathcal{D}} \in \mathbb{Z}_+$ depending on $\mathcal{D}$ only such that, under mild conditions on $\mathcal{D}$, if $N = \overline{O}(S_{\mathcal{D}})$, then we can construct a generalizable memorization network with $O(N^2 n)$ parameters for any $\mathcal{D}_{\mathrm{tr}} \sim \mathcal{D}^N$ in polynomial time.*

$S_{\mathcal{D}}$ is named as the **efficient memorization sample complexity** for $\mathcal{D}$, which measures the complexity of $\mathcal{D}$ so that the generalizable memorization network of any $D_{\mathrm{tr}} \sim \mathcal{D}^N$ can be computed efficiently if $N = \overline{O}(S_{\mathcal{D}})$.

The memorization network in Theorem 1.5 has more parameters than the optimal number $\overline{O}(\sqrt{N})$ of parameters required for memorization. The main reason is that building memorization networks with $\overline{O}(\sqrt{N})$ parameters requires special technical skill that may break the generalization. On the other hand, as mention in [7], over-parametrization is good for generalization, so it is reasonable for us to use more parameters for memorization to achieve generalization.

*Remark* 1.6. We explain the relationship between our results and interpolation learning [7]. Interpolation learning uses optimization to achieve memorization, which is a more practical approach, while our approach gives a theoretical foundation for memorization networks. Once an interpolation is achieved, Theorem 1.2, (1) of Theorem 1.3, and Theorem 1.5 are valid for interpolation learning. For example, according to (1) of Theorem 1.3, $\overline{\Omega}(N_{\mathcal{D}}^2)$ is a lower bound for the sample complexity of interpolation learning, and by Theorem 1.5, $\overline{O}(S_{\mathcal{D}})$ is an upper bound for the sample complexity of efficient interpolation learning.

**Main Contributions**. Under mild conditions for the data distribution $\mathcal{D}$, we have

- We define the *memorization parameter complexity* $N_{\mathcal{D}} \in \mathbb{Z}_+$ of $\mathcal{D}$ such that, a memorization network for any $\mathcal{D}_{\mathrm{tr}} \sim \mathcal{D}^N$ can be constructed, which has $\overline{O}(\sqrt{N})$ or $\le N_{\mathcal{D}}$ parameters. Here, the memorization network has the optimal number of parameters.

- We give two necessary conditions for the construction of generalizable memorization networks for any $\mathcal{D}_{\mathrm{tr}}$ in terms of the width and number of parameters of the memorization network.

- We give a lower bound $\overline{\Omega}(N_{\mathcal{D}}^2)$ of the sample complexity for general memorization networks as well as the exact sample complexity $\overline{O}(N_{\mathcal{D}}^2)$ for memorization networks with $\le N_{\mathcal{D}}$ parameters. We also show that for some data distribution, an exponential number of samples in $n$ is required to achieve generalization.

- We define the *efficient memorization sample complexity* $S_{\mathcal{D}} \in \mathbb{Z}_+$ for $\mathcal{D}$, so that generalizable memorization network of any $D_{\mathrm{tr}} \sim \mathcal{D}^N$ can be computed in polynomial time, if $N = \overline{O}(S_{\mathcal{D}})$.

## 2   Related work

**Memorization**. The problem of memorization has a long history. In [9], it is shown that networks with depth 2 and $\overline{O}(N)$ parameters can memorize a binary dataset of size $N$. In subsequent work,

it is shown that networks with $\overline{O}(N)$ parameters can be a memorization for any dataset [31, 50, 11, 30, 65, 29, 64, 56, 26, 47] and such memorization networks are approximately optimal for generic dataset [64, 56]. Since the VC dimension of neural networks with $N$ parameters and depth $D$ and with ReLU as the activation function is at most $\overline{O}(ND)$ [24, 5, 6], memorizing some special datasets of size $N$ requires at least $\overline{\Omega}(\sqrt{N})$ parameters and there exists a gap between this lower bound $\overline{\Omega}(\sqrt{N})$ and the upper bound $\overline{O}(N)$. Park et al. [49] show that a network with $\overline{O}(N^{2/3})$ parameters is enough for memorization under certain assumptions. Vardi et al. [55] further give the memorization network with optimal number of parameters $\overline{O}(\sqrt{N})$. In [22], strengths of both generalization and memorization are combined in a single neural network. Recently, robust memorization has been studied [35, 62]. As far as we know, the generazability of memorization neural networks has not been studied theoretically.

**Interpolation Learning**. Another line of related research is interpolation learning, that is, leaning under the constraint of memorization, which can be traced back to [52]. Most recent works establish various generalizability of interpolation learning in linear regimes [7, 12, 38, 53, 59, 66]. For instance, Bartlett et al. [7] prove that over-parametrization allows gradient methods to find generalizable interpolating solutions for the linear regime. In relation to this, how to achieve memorization via gradient descent is studied in [13, 14]. Results of this paper can be considered to give sample complexities for interpolation learning.

**Generalization Guarantee**. There exist several ways to ensure generalization of networks. The common way is to estimate the generalization bound or sample complexity of leaning algorithms. Generalization bounds for neural networks are given in terms of the VC dimension [24, 5, 6], under the normal training setting [27, 44, 8], under the differential privacy training setting [1], and under the adversarial training setting [60, 58]. In most cases, these generalization bounds imply that when the training set is large enough, a well-trained network with fixed structure has good generalizability. On the other hand, the relationship between memorization and generalization has also been extensively studied [45, 41, 10, 19, 20]. In [25], sample complexity of neural networks is given when the norm of the transition matrix is limited, in [36], sample complexity of shallow transformers is considered. This paper gives the lower bound and upper bound (in certain cases) of the sample complexities for interpolation learning.

# 3 Notation

In this paper, we use $O(A)$ to mean a value not greater than $cA$ for some constant $c$, and $\overline{O}$ to mean that small quantities, such as logarithm, are omitted. We use $\Omega(A)$ to mean a value not less than $cA$ for some constant $c$, and $\overline{\Omega}$ to mean that small quantities, such as logarithm, are omitted. We say for all $(x, y) \sim \mathcal{D}$ there is event A stand means that $P_{(x,y)\sim\mathcal{D}}(A) = 1$.

## 3.1 Neural network

In this paper, we consider feedforward neural networks of the form $\mathcal{F} : \mathbb{R}^n \to \mathbb{R}$ and the $l$-th hidden layer of $\mathcal{F}(x)$ can be written as

$$X_l = \sigma(W_l X_{l-1} + b_l) \in \mathbb{R}^{n_l},$$

where $\sigma = \text{Relu}$ is the activation function, $X_0 = x$ and $N_0 = n$. The last layer of $\mathcal{F}$ is $\mathcal{F}(x) = W_{L+1} X_L + b_{L+1} \in \mathbb{R}$, where $L$ is the number of hidden layers in $\mathcal{F}$. The depth of $\mathcal{F}$ is $\text{depth}(\mathcal{F}) = L + 1$, the width of $\mathcal{F}$ is $\text{width}(\mathcal{F}) = \max_{i=1}^{L}\{n_i\}$, the number of parameters of $\mathcal{F}$ is $\text{para}(\mathcal{F}) = \sum_{i=0}^{L} n_i(n_{i+1} + 1)$. Denote $\mathbf{H}(n)$ to be the set of all neural networks in the above form.

## 3.2 Data distribution

In this paper, we consider binary classification problems and use $\mathcal{D}$ to denote a joint distribution on $\mathbf{D}(n) = [0, 1]^n \times \{-1, 1\}$. To avoid extreme cases, we focus mainly on a special kind of distribution to be defined in the following.

**Definition 3.1.** For $n \in \mathbb{Z}_+$ and $c \in \mathbb{R}_+$, $\mathcal{D}(n, c)$ is the set of distributions $\mathcal{D}$ on $\mathbf{D}(n)$, which has a *positive separation bound:* $\inf_{(x,1),(z,-1)\sim\mathcal{D}} \|x - z\|_2 \geq c$.

The accuracy of a network $\mathcal{F}$ on a distribution $\mathcal{D}$ is defined as

$$A_{\mathcal{D}}(\mathcal{F}) = \mathbb{P}_{(x,y)\sim\mathcal{D}}(\mathrm{Sgn}(\mathcal{F}(x)) = y).$$

We use $\mathcal{D}_{tr} \sim \mathcal{D}^N$ to mean that $\mathcal{D}_{tr}$ is a set of $N$ data sampled i.i.d. according to $\mathcal{D}$. For convenience, dataset under distribution means that the dataset is i.i.d selected from a data distribution.

*Remark* 3.2. We define the distribution with positive separation bound in for the following reasons. (1) If $\mathcal{D}_{tr} \sim \mathcal{D}^N$ and $\mathcal{D} \in \mathcal{D}(n,c)$, then $x_i \neq x_j$ when $y_i \neq y_j$. Such property ensures that $\mathcal{D}_{tr}$ can be memorized. (2) Proposition 3.3 shows that there exists a $\mathcal{D}$ such that any network is not generalizable over $\mathcal{D}$, and this should be avoided. Therefore, distribution $\mathcal{D}$ needs to meet certain requirements for a dataset sampled from $\mathcal{D}$ to have generalizability. Proof of Proposition 3.3 is given in Appendix A. (3) Most commonly used classification distributions should have positive separation bound.

**Proposition 3.3.** *There exists a distribution $\mathcal{D}$ such that $A_{\mathcal{D}}(\mathcal{F}) \leq 0.5$ for any neural network $\mathcal{F}$.*

### 3.3 Memorization neural network

**Definition 3.4.** A neural network $\mathcal{F} \in \mathbf{H}(n)$ is a memorization of a dataset $\mathcal{D}_{tr}$ over $\mathbf{D}(n)$, if $\mathrm{Sgn}(\mathcal{F}(x)) = y$ for any $(x,y) \in \mathcal{D}_{tr}$.

*Remark* 3.5. Memorization networks can also be defined more strictly as $\mathcal{F}(x) = y$ for any $(x,y) \in \mathcal{D}_{tr}$. In Proposition 4.10 of [62], it is shown that these two types of memorization networks need essentially the same number of parameters.

To be more precise, we treat memorization as a learning algorithm in this paper, as defined below.

**Definition 3.6.** $\mathcal{L} : \cup_{n\in\mathbb{Z}_+} 2^{\mathbf{D}(n)} \to \cup_{n\in\mathbb{Z}_+} \mathbf{H}(n)$ is called a *memorization algorithm* if for any $n$ and $\mathcal{D}_{tr} \in \mathbf{D}(n)$, $\mathcal{L}(\mathcal{D}_{tr})$ is a memorization network of $\mathcal{D}_{tr}$.

Furthermore, a memorization algorithm $\mathcal{L}$ is called an *efficient memorization algorithm* if there exists a polynomial poly : $\mathbb{R} \to \mathbb{R}$ such that $\mathcal{L}(\mathcal{D}_{tr})$ can be computed in time $\mathrm{poly}(\mathrm{size}(\mathcal{D}_{tr}))$, where $\mathrm{size}(\mathcal{D}_{tr})$ is the bit-size of $\mathcal{D}_{tr}$.

*Remark* 3.7. It is clear that if $\mathcal{L}$ is an efficient memorization algorithm, then $\mathrm{para}(\mathcal{L}(\mathcal{D}_{tr}))$ is also polynomial in $\mathrm{size}(\mathcal{D}_{tr})$.

There exist many methods which can construct memorization networks in polynomial times, and all these memorization methods are efficient memorization algorithms, which are summarized in the following proposition.

**Proposition 3.8.** *The methods given in [9, 62] are efficient memorization algorithms. The methods given in [55, 49] are probabilistic efficient memorization algorithms, which can be proved similar to that of Theorem 4.1. More precisely, they are Monte Carlo polynomial-time algorithms.*

## 4 Optimal memorization network for dataset under distribution

By the term "dataset under distribution", we mean datasets that are sampled i.i.d. from a data distribution, and is denoted as $\mathcal{D}_{tr} \sim \mathcal{D}^N$. In this section, we show how to construct the memorization network with the optimal number of parameters for dataset under distribution.

### 4.1 Memorization network with optimal number of parameters

To memorize $N$ samples, $\widetilde{\Omega}(\sqrt{N})$ parameters are necessary [6]. In [55], a memorization network is given which has $\overline{O}(\sqrt{N})$ parameters under certain conditions, where $\overline{O}$ means that some logarithm factors in $N$ and polynomial factors of other values are omitted. Therefore, $\overline{O}(\sqrt{N})$ is the optimal number of parameters for a network to memorize certain dataset. In the following theorem, we show that such a result can be extended to dataset under distribution.

**Theorem 4.1.** *Let $\mathcal{D} \in \mathcal{D}(n,c)$ and $\mathcal{D}_{tr} \sim \mathcal{D}^N$. Then there exists a memorization algorithm $\mathcal{L}$ such that $\mathcal{L}(\mathcal{D}_{tr})$ has width 6 and depth (equivalently, the number of parameters) $O(\sqrt{N}\ln(Nn/c))$. Furthermore, for any $\epsilon \in (0,1)$, $\mathcal{L}(\mathcal{D}_{tr})$ can be computed in time $\mathrm{poly}(\mathrm{size}(\mathcal{D}_{tr}), \ln(1/\epsilon))$ with probability $\geq 1 - \epsilon$.*

**Proof Idea.** *This theorem can be proven using the idea from [55]. Let $\mathcal{D}_{tr} = \{(x_i, y_i)\}_{i=1}^N$. The mainly different is that in [55], it requires $\|x_i - x_j\| \geq c$ for all $i \neq j$, which is no longer valid when $\mathcal{D}_{tr}$ is sampled i.i.d. from distribution $\mathcal{D}$. Since $\mathcal{D}$ has separation bound $c > 0$, we have $\|x_i - x_j\| \geq c$ for all $i, j$ satisfying $y_i \neq y_j$, which is weaker. Despite this difference, the idea of [55] can still be modified to prove the theorem. In constructing such a memorization network, we need to randomly select a vector, and each selection has a probability of 0.5 to give the correct vector. So, repeat the selection $\ln(1/\epsilon)$ times, with probability $1 - \epsilon$, we can get at least one correct vector. Then we can construct the memorization network based on this vector. Detailed proof is given in Appendix B.*

*Remark* 4.2. The algorithm in Theorem 4.1 is a Monte Carlo polynomial-time algorithm, that is, it gives a correct answer with arbitrarily high probability. The algorithm given in [55] is also a Monte Carlo algorithm.

## 4.2 Memorization network with constant number of parameters

In this section, we prove an interesting fact of memorization for dataset under distribution. We show that for a distribution $\mathcal{D} \in \mathcal{D}(n, c)$, there exists a constant $N_{\mathcal{D}} \in \mathbb{Z}_+$ such that for all datasets sampled i.i.d. from $\mathcal{D}$, there exists a memorization network with $N_{\mathcal{D}}$ parameters.

**Theorem 4.3.** *There exists a memorization algorithm $\mathcal{L}$ such that for any $\mathcal{D} \in \mathcal{D}(n, c)$, there is an $N_{\mathcal{D}}' \in \mathbb{Z}_+$ satisfying that for any $N > 0$, with probability 1 of $\mathcal{D}_{tr} \sim \mathcal{D}^N$, we have $\mathrm{para}(\mathcal{L}(\mathcal{D}_{tr})) \leq N_{\mathcal{D}}'$. The smallest $N_{\mathcal{D}}'$ of the distribution $\mathcal{D}$ is called the memorization parameter complexity of $\mathcal{D}$, written as $N_{\mathcal{D}}$.*

**Proof Idea.** *It suffices to show that we can find a memorization network of $\mathcal{D}_{tr}$ with a constant number of parameters, which depends on $\mathcal{D}$ only. The main idea is to take a subset $\mathcal{D}_{tr}'$ of $\mathcal{D}_{tr}$ such that $\mathcal{D}_{tr}$ is contained in the neighborhood of $\mathcal{D}_{tr}'$. It can be proven that the number of elements in this subset is limited. Then construct a robust memorization network of $\mathcal{D}_{tr}'$ with certain budget [62], we obtain a memorization network of $\mathcal{D}_{tr}$, which has a constant number of parameters. The proof is given in Appendix C.*

Combining Theorems 4.1 and 4.3, we can give a memorization network with the optimal number of parameters.

*Remark* 4.4. What we have proven in Theorem 4.3 is that a memorization algorithm with a constant number of parameters can be found, but in most of times, we have $N_{\mathcal{D}}' > N_{\mathcal{D}}$. Furthermore, if $N_{\mathcal{D}}'$ is large for the memorization algorithm, the algorithm can be efficient. Otherwise, if $N_{\mathcal{D}}'$ is closed to $N_{\mathcal{D}}$, the algorithm is usually not efficient.

*Remark* 4.5. It is obvious that the memorization parameter compelxity $N_{\mathcal{D}}$ is the minimum number of parameters required to memorize any dataset sampled i.i.d. from $\mathcal{D}$. $N_{\mathcal{D}}$ is mainly determined by the characteristic of $\mathcal{D} \in \mathcal{D}(n, c)$, so $N_{\mathcal{D}}$ may be related to $n$ and $c$. It is an interesting problem to estimate $N_{\mathcal{D}}$.

# 5  Condition on the network structure for generalizable memorization

In the preceding section, we show that for the dataset under distribution, there exists a memorization algorithm to generate memorization networks with the optimal number of parameters. In this section, we give some conditions for the generalizable memorization networks in terms of width and number of parameters of the network. As a consequence, we show that the commonly used memorization networks with fixed width is not generalizable.

First, we show that networks with fixed width do not have generazability in some situations. Reducing the width and increasing depth is a common way for parameter reduction, but it inevitably limits the network's power, making it unable to achieve good generalization for specific distributions, as shown in the following theorem.

**Theorem 5.1.** *Let $w \in \mathbb{Z}_+$ and $\mathcal{L}$ be a memorization algorithm such that $\mathcal{L}(\mathcal{D}_{tr})$ has width not more than $w$ for all $\mathcal{D}_{tr}$. Then, there exist an integer $n > w$, $c \in \mathbb{R}_+$, and a distribution $\mathcal{D} \in \mathcal{D}(n, c)$ such that, for any $\mathcal{D}_{tr} \sim \mathcal{D}^N$, it holds $A_{\mathcal{D}}(\mathcal{L}(\mathcal{D}_{tr})) \leq 0.51$.*

**Proof Idea.** *As shown in [40, 48], networks with small width are not dense in the space of measurable functions, but this is not enough to estimate the upper bound of the generalization. In order to further measure the upper bound of generalization, we define a special class of distributions. Then, we calculate the upper bound of the generalization of networks with fixed width on this class of distribution. Based on the calculation results, it is possible to find a specific distribution within this class of distributions, such that the fixed-width network exhibits a poor generalization of this distribution. The proof is given in Appendix D.*

It is well known that width of the network is important for the network to be robust [2, 17, 18, 37, 67]. Theorem 5.1 further shows that large width is a necessary condition for generalizabity.

Note that Theorem 5.1 is for a specific data distribution. We will show that for most distributions, providing enough data does not necessarily mean that the memorization algorithm has generalization ability. This highlights the importance of constructing appropriate memorization algorithms to ensure generalization. We need to introduce another parameter for data distribution.

**Definition 5.2.** The distribution $\mathcal{D}$ is said to have *density* $r$, if $\mathbb{P}_{x \sim \mathcal{D}}(x \in A)/V(A) \leq r$ for any closed set $A \subset [0,1]^n$, where $V(A)$ is the volume of $A$.

Loosely speaking, the density of a distribution is the upper bound of the density function.

**Theorem 5.3.** *For any $n \in \mathbb{Z}_+, r, c \in \mathbb{R}_+$, if distribution $\mathcal{D} \in \mathcal{D}(n, c)$ has density $r$, then for any $N \in \mathbb{Z}_+$ and $\mathcal{D}_{tr} \sim \mathcal{D}^N$, there exists a memorization network $\mathcal{F}$ for $\mathcal{D}_{tr}$ such that $\mathrm{para}(\mathcal{F}) = O(n + \sqrt{N} \ln(Nnr/c))$ and $A_{\mathcal{D}}(\mathcal{F}) \leq 0.51$.*

**Proof Idea.** *We refer to the classical memorization construction idea [55]. The main body includes three parts. Firstly, compress the data in $\mathcal{D}_{tr}$ into one dimension. Secondly, map the compressed data to some specific values. Finally, use such a value to get the label of input. Moreover, we will pay more attention to points outside the dataset. We use some skills to control the classification results of points that do not appear in the dataset $\mathcal{D}_{tr}$, so that the memorization network will give the wrong label to the points that are not in $\mathcal{D}_{tr}$ as much as possible to reduce its accuracy. The general approach is the following: (1) Find a set in which each point is not presented in $\mathcal{D}_{tr}$ and has the same label under distribution $\mathcal{D}$. Without loss of generality, let they have label $1$. (2) In the second step mentioned in the previous step, ensure that the mapped results of the points in the set mentioned in (1) are similar to the samples with label $-1$. This will cause the third step to output the label $-1$, leading to an erroneous classification result for the points in the set. The proof is given in Appendix E.*

*Remark* 5.4. Theorem 5.1 shows that the width of the generazable memorization network needs to increase with the increase of the data dimension. Theorem 5.3 shows that when $\mathrm{para}(\mathcal{F}) = \overline{O}(\sqrt{N})$, the memorization network may have poor generalizability for most distributions. The above two theorems indicate that no matter how large the dataset is, there always exist memorization networks with poor generalization. In terms of sample complexity, it means that for the hypotheses of neural networks with fixed width or with optimal number of parameters, the sample complexity is infinite, contrary to the uniform generalization bound for feedforward neural networks [63, Lemma D.16].

*Remark* 5.5. It is worth mentioning that the two theorems in this section cannot be obtained from the lower bound of the generalization gap [44], and more details are shown in Appendix E.

## 6 Sample complexity for memorization algorithm

As said in the preceding section, generalization of memorization inevitably requires certain conditions. In this section, we give the necessary and sufficient condition for generalization for the memorization algorithm in Section 4 in terms of sample complexity.

We first give a lower bound for the sample complexity for general memorization algorithms and then an upper bound for memorization algorithms which output networks with an optimal number of parameters. The lower and upper bounds are approximately the same, thus giving the exact sample complexity in this case.

## 6.1 Lower bound for sample complexity of memorization algorithm

Roughly speaking, the sample complexity of a learning algorithm is the number of samples required to achieve generalizability [44]. The following theorem gives a lower bound for the sample complexity of memorization algorithms based on $N_\mathcal{D}$, which has been defined in Theorem 4.3.

**Theorem 6.1.** *There exists* **no** *memorization algorithm $\mathcal{L}$ which satisfies that for any $n \in \mathbb{Z}_+, c \in \mathbb{R}_+, \epsilon, \delta \in (0,1)$, if $\mathcal{D} \in \mathcal{D}(n,c)$ and $N \geq v\frac{N_\mathcal{D}^2}{\ln^2(N_\mathcal{D})}(1 - 2\epsilon - \delta)$, it holds*

$$\mathbb{P}_{\mathcal{D}_{tr} \sim \mathcal{D}^N}(A(\mathcal{L}(\mathcal{D}_{tr})) \geq 1 - \epsilon) \geq 1 - \delta$$

*where $v$ is an absolute constant which does not depend on $N, n, c, \epsilon, \delta$.*

**Proof Idea.** *The mainly idea is that: for a dataset $\mathcal{D}_{tr} \subset [0,1]^n \times \{-1,1\}$ with $|\mathcal{D}_{tr}| = N$, we can find some distributions $\mathcal{D}_1, \mathcal{D}_2, \ldots$, such that if $\mathcal{D}_{tr,i} \sim (\mathcal{D}_i)^N$, then with a positive probability, it hold $\mathcal{D}_{tr,i} = \mathcal{D}_{tr}$. In addition, each distribution has a certain degree of difference from the others. It is easy to see that $\mathcal{L}(\mathcal{D}_{tr})$ is a fixed network for a given $\mathcal{L}$, so $\mathcal{L}(\mathcal{D}_{tr})$ cannot fit all $\mathcal{D}_i$ well because $\mathcal{D}_i$ are different to some degree. So, if a memorization algorithm $\mathcal{L}$ satisfies the condition in the theorem, we try to construct some distributions $\{\mathcal{D}_i\}_{i=1}^n$, and use the above idea to prove that $\mathcal{L}$ cannot fit one of the distributions in $\{\mathcal{D}_i\}_{i=1}^n$, and obtain contradictions. The proof of the theorem is given in Appendix F.*

*Remark* 6.2. In general, the sample complexity depends on the data distribution, hypothesis space, learning algorithms, and $\epsilon, \delta$. Since $N_\mathcal{D}$ is related to $n$ and $c$, the lower bound in Theorem 6.1 also depends on $n$ and $c$. Here, the hypothesis space is the memorization networks, which is implicitly reflected in $N_\mathcal{D}$.

*Remark* 6.3. Roughly strictly, if we consider interpolation learning, that is, training network under the constraint of memorizing the dataset, then Theorem 6.1 also provides a lower bound for the sample complexity.

This theorem shows that if we want memorization algorithms to have guaranteed generalization, then about $\overline{O}(N_\mathcal{D}^2)$ samples are required. As a consequence, we show that, for some data distribution, it need an exponential number of samples to achieve generalization. The proof is also in Appendix F.

**Corollary 6.4.** *For any memorization algorithm $\mathcal{L}$ and any $\epsilon, \delta \in (0,1)$, there exist $n \in \mathbb{Z}_+, c > 0$ and a distribution $\mathcal{D} \in \mathcal{D}(n,c)$, such that in order for $\mathcal{L}$ to have generalizability on $\mathcal{D}$, that is for all $N \geq N_0$, there is*

$$\mathbb{P}_{\mathcal{D}_{tr} \sim \mathcal{D}^N}(A(\mathcal{L}(\mathcal{D}_{tr})) \geq 1 - \epsilon) \geq 1 - \delta,$$

*$N_0$ must be more than $v(2^{2\lceil \frac{n}{\lceil c^2 \rceil}\rceil}c^4(1 - 2\epsilon - \delta)/n^2)$, where $v$ is an absolute constant not depending on $N, n, c, \epsilon, \delta$.*

## 6.2 Exact sample complexity of memorization algorithm with $N_\mathcal{D}$ parameters

In Theorem 6.1, it is shown that $\overline{\Omega}(N_\mathcal{D}^2)$ samples are necessary for generalizability of memorization. The following theorem shows that there exists a memorization algorithm that can reach generalization with $\overline{O}(N_\mathcal{D}^2)$ samples.

**Theorem 6.5.** *For all memorization algorithms $\mathcal{L}$ satisfies that $\mathcal{L}(\mathcal{D}_{tr})$ has at most $N_\mathcal{D}$ parameters, with probability 1 for $\mathcal{D}_{tr} \sim D^N$, we have*

**(1)** *For any $c \in \mathbb{R}, \epsilon, \delta \in (0,1)$, $n \in \mathbb{Z}_+$, if $\mathcal{D} \in \mathcal{D}(n,c)$ and $N \geq \frac{vN_\mathcal{D}^2 \ln(N_\mathcal{D}/(\epsilon^2\delta))}{\epsilon^2}$, then*

$$\mathbb{P}_{\mathcal{D}_{tr} \sim \mathcal{D}^N}(A(\mathcal{L}(\mathcal{D}_{tr})) \geq 1 - \epsilon) \geq 1 - \delta,$$

*where $v$ is an absolute constant which does not depend on $N, n, c, \epsilon, \delta$.*

**(2)** *If $P \neq NP$, then all such algorithms are not efficient.*

**Proof Idea.** *For the proof of (1), we need to use the $N_\mathcal{D}$ to calculate the VC-dimension [6], and take such a dimension in the generalization bound theorem [44] to obtain the result. For the proof of (2), we show that, if such algorithm is efficient, then we can solve the following reversible 6-SAT [43] problem, which is defined below and is an NPC problem. The proof of the theorem is given in Appendix G.*

**Definition 6.6.** Let $\varphi$ be a Boolean formula and $\overline{\varphi}$ the formula obtained from $\varphi$ by negating each variable. The Boolean formula $\varphi$ is called *reversible* if either both $\varphi$ and $\overline{\varphi}$ are satisfiable or both are not satisfiable. The *reversible satisfiability problem* is to recognize the satisfiability of reversible formulae in conjunctive normal form (CNF). By the *reversible 6-SAT*, we mean the reversible satisfiability problem for CNF formulae with six variables per clause. In [43], it is shown that the reversible 6-SAT is NPC.

Combining Theorems 6.1 and 6.5, we see that $N = \overline{O}(N_{\mathcal{D}}^2)$ is the necessary and sufficient condition for the memorization algorithm to generalize, and hence $\overline{O}(N_{\mathcal{D}}^2)$ is the exact sample complexity for memorization algorithms with $N_{\mathcal{D}}$ parameters over the distribution $\mathcal{D}(n, c)$.

Unfortunately, by (2) of Theorem 6.5, this memorization algorithm is not efficient when the memorization has no more than $N_{\mathcal{D}}$ parameters. Furthermore, we conjecture that there exist no efficient memorization algorithms that can use $\overline{O}(N_{\mathcal{D}}^2)$ samples to reach generalization in the general case, as shown in the following conjecture.

*Conjecture* 6.7. If P$\neq$ NP, there exist no efficient memorization algorithms that can reach generalization with $\overline{O}(N_{\mathcal{D}}^2)$ samples for all $\mathcal{D} \in \mathcal{D}(n, c)$.

*Remark* 6.8. This result also provides certain theoretical explanation for the over-parameterization mystery [45, 7, 4]: for memorization algorithms with $N_{\mathcal{D}}$ parameters, the exact sample complexity $\overline{O}(N_{\mathcal{D}}^2)$ is greater than the number of parameters. Thus, the networks is under-parameterized and for such a network, even if it is generalizable, it cannot be computed efficiently.

## 7 Efficient memorization algorithm with guaranteed generalization

In the preceding section, we show that there exist memorization algorithms that are generalizable when $N = \overline{O}(N_{\mathcal{D}}^2)$, but such an algorithm is not efficient. In this section, we give an efficient memorization algorithm with guaranteed generalization.

First, we define the efficient memorization sample complexity of $\mathcal{D}$.

**Definition 7.1.** For $(x, y) \sim \mathcal{D}$, let $L_{(x,y)} = \min_{(z,-y) \sim \mathcal{D}} \|x - z\|_2$ and $B((x, y)) = \mathbb{B}_2(x, L_{(x,y)}/3.1) = \{z \in \mathbb{R}^n : \|z - x\|_2 \leq L_{(x,y)}/3.1\}$. The nearby set $S$ of $\mathcal{D}$ is a subset of sample $(x, y)$ which is in distribution $\mathcal{D}$ and satisfies: (1) for any $(x, y) \sim \mathcal{D}$, $x \in \cup_{(z,w) \in S} B((z, w))$; (2) $|S|$ is minimum.

Evidently, for any $\mathcal{D} \in \mathcal{D}(n, c)$, its nearby set is finite, as shown by Proposition 7.7. $S_{\mathcal{D}} = |S|$ is called the *efficient memorization sample complexity* of $\mathcal{D}$, the meaning of which is given in Theorem 7.3.

*Remark* 7.2. In the above definition, we use $L_{(x,y)}/3.1$ to be the radius of $B((x, y))$. In fact, when 3.1 is replaced by any real number greater than 3, the following theorem is still valid.

**Theorem 7.3.** *There exists an efficient memorization algorithm $\mathcal{L}$ such that for any $c \in \mathbb{R}, \epsilon, \delta \in (0, 1)$, $n \in \mathbb{Z}_+$, and $\mathcal{D} \in \mathcal{D}(n, c)$, if $N \geq \frac{S_{\mathcal{D}} \ln(S_{\mathcal{D}}/\delta)}{\epsilon}$, then*

$$\mathbb{P}_{\mathcal{D}_{tr} \sim \mathcal{D}^N} (A(\mathcal{L}(\mathcal{D}_{tr})) \geq 1 - \epsilon) \geq 1 - \delta.$$

*Moreover, for any $\mathcal{D}_{tr} \sim \mathcal{D}^N$, $\mathcal{L}(\mathcal{D}_{tr})$ has at most $O(N^2 n)$ parameters.*

**Proof Idea.** *For a given dataset $\mathcal{D}_{tr} \subset [0, 1]^n \times \{-1, 1\}$, we use the following two steps to construct a memorization network.*

*Step 1. Find suitable convex sets $\{C_i\}$ in $[0, 1]^n$ such that each sample in $\mathcal{D}_{tr}$ is in at least one of these convex sets. Furthermore, if $x, z \in C_i$ and $(x, y_x), (z, y_z) \in \mathcal{D}_{tr}$, then $y_x = y_z$, and define $y(C_i) = y_x$.*

*Step 2. Construct a network $\mathcal{F}$ such that for any $x \in C_i$, $\text{Sgn}(\mathcal{F}(x)) = y(C_i)$. This network must be a memorization of $\mathcal{D}_{tr}$, because each sample in $\mathcal{D}_{tr}$ is in at least one of $\{C_i\}$. Hence, if $x \in C_i$ and $(x, y_x) \in \mathcal{D}_{tr}$, then $\text{Sgn}(\mathcal{F}(x)) = y(C_i) = y_x$. The proof of the theorem is given in Appendix H.*

*Remark* 7.4. Theorem 7.3 shows that there exists an efficient and generalizable memorization algorithm when $N = \overline{O}(S_{\mathcal{D}})$. Thus, $S_{\mathcal{D}}$ is an intrinsic complexity measure of $\mathcal{D}$ on whether it is

easy to learn and generalize. By Theorem 6.1, $S_{\mathcal{D}} \geq N_{\mathcal{D}}^2$ for some $\mathcal{D}$, but for some "nice" $\mathcal{D}$, $S_{\mathcal{D}}$ could be small. It is an interesting problem to estimate $S_{\mathcal{D}}$.

*Remark* 7.5. Theorem 7.3 uses $O(N^2 n)$ parameters, highlight the importance of over-parameterization [45, 7, 4]. Interestingly, Remark 6.8 shows that if the network has $O(\sqrt{N})$ parameters, even if it is generalizable, it cannot be computed efficiently.

The experimental results of the memorization algorithm mentioned in Theorem 7.3 are given in Appendix I. Unfortunately, for commonly used datasets such as CIFAR-10, this algorithm cannot surpass the network obtained by training with SGD, in terms of test accuracy. Thus, the main purpose of the algorithm is theoretical, that is, it provides a polynomial-time memorization algorithm that can achieve generalization when the training dataset contains $\overline{O}(S_{\mathcal{D}})$ samples. In comparison of theoretical works, training networks is NP-hard for small networks [32, 51, 39, 15, 3, 42, 16, 23, 21] and the guarantee of generalization needs strong assumptions on the loss function [46, 27, 34, 61, 60, 58].

Finally, we give an estimate for $S_{\mathcal{D}}$. From Corollary 6.4 and Theorem 7.3, we obtain a lower bound for $S_{\mathcal{D}}$.

**Corollary 7.6.** *There exists a distribution* $\mathcal{D} \in \mathcal{D}(n, c)$ *such that* $S_{\mathcal{D}} \ln(S_{\mathcal{D}}/\delta) \geq \overline{\Omega}(\frac{c^4}{n^2} 2^{2\lceil \frac{n}{\lceil c^2 \rceil} \rceil})$.

We will give an upper bound for $S_{\mathcal{D}}$ in the following proposition, and the proof is given in Appendix H.1. From the proposition, it is clear that $S_{\mathcal{D}}$ is finite.

**Proposition 7.7.** *For any* $\mathcal{D} \in \mathcal{D}(n, c)$, *we have* $S_{\mathcal{D}} \leq ([6.2n/c] + 1)^n$.

*Remark* 7.8. The above proposition gives an upper bound of $S_{\mathcal{D}}$ when $\mathcal{D} \in \mathcal{D}(n, c)$, and this does not mean that $S_{\mathcal{D}}$ is exponential for all $\mathcal{D} \in \mathcal{D}(n, c)$. Determining the conditions under which $S_{\mathcal{D}}$ is small for a given $\mathcal{D}$ is a compelling problem.

# 8 Conclusion

Memorization originally focuses on theoretical study of the expressive power of neural networks. Recently, memorization is believed to be a key reason why over-parameterized deep learning models have excellent generalizability and thus the more practical interpolation learning approach has been extensively studied. But the generalizability theory of memorization algorithms is not yet given, and this paper fills this theoretical gap in several aspects.

We first show how to construct memorization networks for dataset sampled i.i.d from a data distribution, which have the optimal number of parameters, and then show that some commonly used memorization networks do not have generalizability even if the dataset is drawn i.i.d. from a data distribution and contains a sufficiently large number of samples. Furthermore, we establish the sample complexity of memorization algorithm in several situations, including a lower bound for the memorization sample complexity and an upper bound for the efficient memorization sample complexity.

**Limitation and future work** Two numerical complexities $N_{\mathcal{D}}$ and $S_{\mathcal{D}}$ for a data distribution $\mathcal{D}$ are introduced in this paper, which are used to describe the size of the memorization networks and the efficient memorization sample complexity for any i.i.d. dataset of $\mathcal{D}$. $N_{\mathcal{D}}$ is also a lower bound for the sample complexity of memorization algorithms. However, we do not know how to compute $N_{\mathcal{D}}$ and $S_{\mathcal{D}}$, which is an interesting future work. Conjecture 6.7 tries to give a lower bound for the efficient memorization sample complexity. More generally, can we write $N_{\mathcal{D}}$ and $S_{\mathcal{D}}$ as functions of the probability density function $p(x, y)$ of $\mathcal{D}$?

Corollary 6.4 indicates that even for the "nice" data distributions $\mathcal{D}(n, c)$, to achieve generalization for some data distribution requires an exponential number of parameters. This indicates that there exists **"data curse of dimensionality"**, that is, to achieve generalizability for certain data distribution, neural networks with exponential number of parameters are needed. Considering the practical success of deep learning and the double descent phenomenon [45], the data distributions used in practice should have better properties than $\mathcal{D}(n, c)$, and finding data distributions with polynomial size efficient memorization sample complexity $E_{\mathcal{D}}$ is an important problem.

Finally, finding a memorization algorithm that can achieve SOTA results in solving practical image classification problems is also a challenge problem.

## Acknowledgments

This work is supported by CAS Project for Young Scientists in Basic Research, Grant No.YSBR-040, ISCAS New Cultivation Project ISCAS-PYFX-202201, and ISCAS Basic Research ISCAS-JCZD-202302. This work is also supported by NKRDP grant No.2018YFA0704705, grant GJ0090202, and NSFC grant No.12288201. The authors thank anonymous referees for their valuable comments.

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

# A Proof of Proposition 3.3

Using the following steps, we construct a distribution $\mathcal{D}$ in $[0,1] \times \{-1,1\}$. We use $(x,y) \sim \mathcal{D}$ to mean that

(1) Randomly select a number in $\{-1,1\}$ as the label $y$.

(2) If we get 1 as the label, then randomly select an irrational number in $[0,1]$ as samples $x$; if we get $-1$ as the label, then randomly select a rational number in $[0,1]$ as samples $x$.

Then Proposition 3.3 follows from the following lemma.

**Lemma A.1.** *For any neural network $\mathcal{F}$, we have $A_{\mathcal{D}}(\mathcal{F}) \leq 0.5$.*

*Proof.* Let $\mathcal{F}$ be a network. Firstly, we show that $\mathcal{F}$ can be written as

$$\mathcal{F} = \sum_{i=1}^{M} L_i(x) I(x \in A_i), \tag{1}$$

where $L_i$ are linear functions, $I(x) = 1$ if $x$ is true or $I(x) = 0$. In addition, $A_i$ is an interval and $A_j \cap A_i = \emptyset$ when $j \neq i$, and $L_i(x) I(x \in A_i)$ is a non-negative or non-positive function for any $i \in [M]$.

It is obvious that the network is a locally linear function with a finite number of linear regions, so we can write

$$\mathcal{F} = \sum_{i=1}^{M} L_i'(x) I(x \in A_i'), \tag{2}$$

where $L_i'$ are linear functions, $A_i'$ is an interval and $A_j' \cap A_i' = \emptyset$ when $j \neq i$.

Consider that $L_i'(x) I(x \in A_i') = L_i'(x) I(x \in A_i', L_i'(x) > 0) + L_i'(x) I(x \in A_i', L_i'(x) < 0)$, and $L_i'(x) I(x \in A_i', L_i'(x) > 0)$ is a non-negative function, $\{x \in A_i', L_i'(x) > 0\}$ is an interval which is disjoint with $\{x \in A_i', L_i'(x) < 0\}$. Similarly as $L_i'(x) I(x \in A_i', L_i'(x) < 0)$, so we use $L_i'(x) I(x \in A_i')$ in (2) instead of $L_i'(x) I(x \in A_i', L_i'(x) > 0) + L_i'(x) I(x \in A_i', L_i'(x) < 0)$. Then we get the equation (1).

By equation (2), we have that

$$
\begin{aligned}
& \mathbb{P}_{(x,y)\sim\mathcal{D}}(\mathrm{Sgn}(\mathcal{F}(x)) = y) \\
= & \mathbb{P}_{(x,y)\sim\mathcal{D}}(\mathrm{Sgn}(\sum_{i=1}^{M} L_i(x) I(x \in A_i)) = y) \\
= & \sum_{i=1}^{M} \mathbb{P}_{(x,y)\sim\mathcal{D}}(\mathrm{Sgn}(L_i(x) I(x \in A_i)) = y, x \in A_i) \\
= & \sum_{i=1}^{M} \mathbb{P}_{(x,y)\sim\mathcal{D}}(\mathrm{Sgn}(L_i(x) I(x \in A_i)) = y | x \in A_i) \mathbb{P}_{(x,y)\sim\mathcal{D}}(x \in A_i).
\end{aligned}
\tag{3}
$$

The second equation uses $A_i \cap A_j = \emptyset$.

For convenience, we use $x \in \mathbb{R}_r$ to mean that $x$ is an irrational number and $x \notin \mathbb{R}_r$ to mean that $x$ is a rational number. Then, if $L_i(x) I(x \in A_i)$ is a non-negative function, then we have $\mathbb{P}_{(x,y)\sim\mathcal{D}}(\mathrm{Sgn}(L_i(x) I(x \in A_i)) = y | x \in A_i) \leq \mathbb{P}_{(x,y)\sim\mathcal{D}}(x \in R_r | x \in A_i)$. Moreover, we have that

$$
\begin{aligned}
& \mathbb{P}_{(x,y)\sim\mathcal{D}}(x \in R_r | x \in A_i) \\
= & \frac{\mathbb{P}_{(x,y)\sim\mathcal{D}}(x \in R_r, x \in A_i)}{\mathbb{P}_{(x,y)\sim\mathcal{D}}(x \in A_i)} \\
= & \frac{0.5 \mathbb{P}_{(x,y)\sim\mathcal{D}}(x \in A_i | x \in R_r)}{\mathbb{P}_{(x,y)\sim\mathcal{D}}(x \in A_i)} \\
= & \frac{0.5 \mathbb{P}_{(x,y)\sim\mathcal{D}}(x \in A_i | x \in R_r)}{\mathbb{P}_{(x,y)\sim\mathcal{D}}(x \in R_r) \mathbb{P}_{(x,y)\sim\mathcal{D}}(x \in A_i | x \in R_r) + \mathbb{P}_{(x,y)\sim\mathcal{D}}(x \notin R_r) \mathbb{P}_{(x,y)\sim\mathcal{D}}(x \in A_i | x \notin R_r)} \\
= & \frac{\mathbb{P}_{(x,y)\sim\mathcal{D}}(x \in A_i | x \in R_r)}{\mathbb{P}_{(x,y)\sim\mathcal{D}}(x \in A_i | x \in R_r) + \mathbb{P}_{(x,y)\sim\mathcal{D}}(x \in A_i | x \notin R_r)}.
\end{aligned}
$$

By (2) in the definition of $\mathcal{D}$, we have $\mathbb{P}_{(x,y)\sim\mathcal{D}}(x \in A_i | x \in R_r) = \mathbb{P}_{(x,y)\sim\mathcal{D}}(x \in A_i | x \notin R_r)$. Substituting this in equation (3), we have that $\mathbb{P}_{(x,y)\sim\mathcal{D}}(\mathrm{Sgn}(L_i(x) I(x \in A_i)) = y | x \in A_i) \leq \mathbb{P}_{(x,y)\sim\mathcal{D}}(x \in R_r | x \in A_i) = \frac{\mathbb{P}_{(x,y)\sim\mathcal{D}}(x \in A_i | x \in R_r)}{\mathbb{P}_{(x,y)\sim\mathcal{D}}(x \in A_i | x \in R_r) + \mathbb{P}_{(x,y)\sim\mathcal{D}}(x \in A_i | x \notin R_r)} = 0.5$. Proof is similar when $L_i(x) I(x \in A_i)$ is a non-positive function.

Using this in equation (2), we have that

$$\mathbb{P}_{(x,y)\sim\mathcal{D}}(\text{Sgn}(\mathcal{F}(x)) = y)$$
$$= \sum_{i=1}^{M} \mathbb{P}_{(x,y)\sim\mathcal{D}}(\text{Sgn}(L_i(x)I(x \in A_i)) = y|x \in A_i)\mathbb{P}_{(x,y)\sim\mathcal{D}}(x \in A_i)$$
$$\leq \sum_{i=1}^{M} 0.5\mathbb{P}_{(x,y)\sim\mathcal{D}}(x \in A_i) \leq 0.5.$$

The lemma is proved. $\square$

## B Proof of Theorem 4.1

For the proof of this theorem, we mainly follow the constructive approach of the memorization network in [55]. Our proof is divided into four parts.

### B.1 Data Compression

The general method of constructing memorization networks will compress the data into a low dimensional space at first, and we follow this approach. We are trying to compress the data into a 1-dimensional space, and we require the compressed data to meet some conditions, as shown in the following lemma.

**Lemma B.1.** *Let $\mathcal{D}$ be a distribution in $[0,1]^n \times \{-1,1\}$ with separation bound $c$ and $\mathcal{D}_{tr} \sim \mathcal{D}^N$. Then, there exist $w \in \mathbb{R}^n$ and $b \in \mathbb{R}$ such that*
*(1): $O(nN^2/c) \geq wx + b \geq 1$ for all $x \in [0,1]^n$;*
*(2): $|wx - wz| \geq 4$ for all $(x,1), (z,-1) \in \mathcal{D}_{tr}$.*

To prove this lemma, we need the following lemma.

**Lemma B.2.** *For any $v \in \mathbb{R}^n$ and $T \geq 1$, let $u \in \mathbb{R}^n$ be uniformly randomly sampled from the hypersphere $S^{n-1}$. Then we have $P(|\langle u, v \rangle| < \frac{||v||_2}{T}\sqrt{\frac{8}{n\pi}}) < \frac{2}{T}$.*

This is Lemma 13 in [49]. Now, we prove the lemma B.1.

*Proof.* Let $c_0 = \min_{(x,-1),(z,1)\in\mathcal{D}_{tr}} ||x - z||_2$. Then, we prove the following result:

**Result R1:** Let $u \in \mathbb{R}^n$ be uniformly randomly sampled from the hypersphere $S^{n-1}$, then there are $P(|\langle u, (x-z) \rangle| \geq \frac{c_0}{4N^2}\sqrt{\frac{8}{n\pi}}, \forall(x,-1),(z,1) \in \mathcal{D}_{tr}) > 0.5$.

By lemma B.2, and take $T = 4N^2$, for any $x, z$ which satisfies $(x,-1), (z,1) \in \mathcal{D}_{tr}$, we have that: let $u \in \mathbb{R}^n$ be uniformly randomly sampled from the hypersphere $S^{n-1}$, then there are $P(|\langle u, (x-z) \rangle| < \frac{c_0}{4N^2}\sqrt{\frac{8}{n\pi}}) < \frac{2}{4N^2}$, using $||x - z||_2 \geq c_0$ here. So, it holds

$$P(|\langle u, (x-z) \rangle| \geq \tfrac{c_0}{4N^2}\sqrt{\tfrac{8}{n\pi}}, \forall(x,-1),(z,1) \in \mathcal{D}_{tr})$$
$$\geq 1 - \sum_{(x,-1),(z,1)\in\mathcal{D}_{tr}} P(|\langle u, (x-z) \rangle| < \tfrac{c_0}{4N^2}\sqrt{\tfrac{8}{n\pi}})$$
$$> 1 - \tfrac{2N^2}{4N^2}.$$
$$= 0.5$$

We proved Result R1.

In practice, to find such a vector, we can randomly select a vector $u$ in hypersphere $S^{n-1}$, and verify that if it satisfies $|\langle u, (x-z) \rangle| \geq \frac{c_0}{4N^2}\sqrt{\frac{8}{n\pi}}, \forall(x,-1),(z,1) \in \mathcal{D}_{tr}$. Verifying such a fact needs $\text{poly}(B(\mathcal{D}_{tr}))$ times. If such a $u$ is not what we want, randomly select a vector $u$ and verify it again.

In each selection, with probability 0.5, we can get a vector we need, so with $\ln 1/\epsilon$ times the selections, we can get a vector we need with probability $1 - \epsilon$.

**Construct $w, b$ and verify their rationality**

By the above result, we have that: there exists a $u \in \mathbb{R}^n$ such that $||u||_2 = 1$ and $|\langle u, (x-z) \rangle| \geq \frac{c_0}{4N^2}\sqrt{\frac{8}{n\pi}}, \forall(x,-1),(z,1) \in \mathcal{D}_{tr}$, and we can find such a $u$ in $\text{poly}(B(\mathcal{D}_{tr}), \ln(1/\epsilon))$ times.

Now, let $w = \frac{16\sqrt{n}N^2}{c_0}u$ and $b = ||w||_2\sqrt{n} + 1$, then we show that $w$ and $b$ are what we want:

(1): We have $O(nN^2/c) \geq wx + b \geq 1$ for all $x \in [0,1]^n$.

Firstly, because $\mathcal{D}$ is defined in $[0,1]^n \times \{-1,1\}$, so it holds $||x||_2 \leq \sqrt{n}$ for any $(x,y) \in \mathcal{D}_{tr}$, and consequently $wx + b \geq b - ||w||_2\sqrt{n} \geq 1$.

On the other hand, $|wx| \leq ||w||_2\sqrt{n} \leq O(\frac{nN^2}{c_0})$, so $wx + b \leq |wx| + b \leq O(nN^2/c_0) \leq O(nN^2/c)$.

(2): We have $|w(x-z)| \geq 4$ for all $(x,1),(z,-1) \in \mathcal{D}_{tr}$.

It is easy to see that $|w(x-z)| \geq |\frac{16\sqrt{n}N^2}{c_0}u(x-z)| = \frac{16\sqrt{n}N^2}{c_0}|u(x-z)|$. Because $|u(x-z)| \geq \frac{c_0}{4\sqrt{n}N^2}$, so $|w(x-z)| = \frac{16\sqrt{n}N^2}{c_0}|u(x-z)| \geq \frac{16\sqrt{n}N^2}{c_0}\frac{c_0}{4\sqrt{n}N^2} = 4$.

By Definition 3.1, we know that $c_0 \geq c$. So, $w$ and $b$ are what we want. The lemma is proved. $\square$

## B.2 Data Projection

The purpose of this part is to map the compressed data into appropriate values.

Let $w \in \mathbb{R}^n$ and $b \in \mathbb{R}$ be given and $\mathcal{D}_{tr} = \{(x_i, y_i)\}_{i=1}^N$. Without losing generality, we assume that $0 < wx_i < wx_{i+1}$.

In this section, we show that, after compressing the data into 1-dimension, we can use a network $\mathcal{F}$ to map $wx_i + b$ to $v_{[\frac{i}{[\sqrt{N}]}]}$, where $\{v_j\}_{j=0}^{[\frac{N}{[\sqrt{N}]}]} \in \mathbb{R}^+$ are given values. This network has $O(\sqrt{N})$ parameters and width 4, as shown in the following lemma.

**Lemma B.3.** *Let $\{x_i\}_{i=1}^N \subset \mathbb{R}^+$, $\{v_j\}_{j=0}^{[\frac{N}{[\sqrt{N}]}]} \subset \mathbb{R}^+$. Assume that $x_i < x_{i+1}$. Then a network $\mathcal{F}$ with width 4 and depth $O(\sqrt{N})$ (at most $O(\sqrt{N})$ parameters) can be obtained such that $\mathcal{F}(x_i) = v_{[\frac{i}{\sqrt{N}}]}$ for all $i \in [N]$.*

*Proof.* Let $\mathcal{F}^i(x)$ be the $i$-th hidden layer of network $\mathcal{F}$, $(\mathcal{F}^i)_j$ be the $j$-th nodes of $i$-th hidden layer of network $\mathcal{F}$.

Let $q_i = x_{i+1} - x_i$ and $t(i) = \text{argmax}_{j \in [N]}\{[j/\sqrt{N}] = i\}$. Consider the following network $\mathcal{F}$:

The $2i + 1$ hidden layer has width 4, and each node is:

$$(\mathcal{F}^{2i+1})_1(x) = \text{Relu}((\mathcal{F}^{2i})_2(x) - (x_{t(i)+1}) + 2q_{t(i)}/3);$$

$$(\mathcal{F}^{2i+1})_2(x) = \text{Relu}((\mathcal{F}^{2i})_2(x) - (x_{t(i)+1}) + q_{t(i)}/3);$$

$$(\mathcal{F}^{2i+1})_3(x) = \text{Relu}((\mathcal{F}^{2i})_1(x));$$

$$(\mathcal{F}^{2i+1})_4(x) = \text{Relu}((\mathcal{F}^{2i})_2(x)).$$

For the case $i = 0$, let $(\mathcal{F}^0)_2(x) = x$ and $(\mathcal{F}^1)_3(x) = v_0$.

The $(2i + 2)$-th hidden layer is:

$$(\mathcal{F}^{2i+2})_1(x) = \text{Relu}((\mathcal{F}^{2i+1})_3(x) + \frac{v_{i+1} - v_i}{q_{t(i)}/3}((\mathcal{F}^{2i+1})_1(x) - (\mathcal{F}^{2i+1})_2(x)));$$

$$(\mathcal{F}^{2i+2})_2(x) = \text{Relu}((\mathcal{F}^{2i+1})_4(x)).$$

The output is $\mathcal{F}(x) = (\mathcal{F}^{2[N/\sqrt{N}]})_1(x)$.

This network has width 4 and $O(\sqrt{N})$ hidden layers. We can verify that such a network is what we want as follows.

Firstly, it is easy to see that $(\mathcal{F}^{2i+2})_2(x) = \text{Relu}((\mathcal{F}^{2i+1})_4(x)) = \text{Relu}((\mathcal{F}^{2i})_2(x)) = \text{Relu}((\mathcal{F}^{2i-1})_4(x)) = \cdots = \text{Relu}((\mathcal{F}^1)_4(x)) = \text{Relu}(x) = x$.

Then, for $\frac{v_{i+1}-v_i}{q_{t(i)}/3}((\mathcal{F}^{2i+1})_1(x)-(\mathcal{F}^{2i+1})_2(x)) = \frac{v_i-v_{i-1}}{q_{t(i)}/3}(\text{Relu}(x-x_{t(i)+1}+2q_{t(i)}/3)-\text{Relu}(x-x_{t(i)+1}+q_{t(i)}/3)$, easy to verify that, when $x \leq x_{t(i)}$, it is 0; when $x \geq x_{t(i+1)}$, it is $v_{i+1}-v_i$.

By the above two results, we have that $(\mathcal{F}^{2i+2})_1(x) = \text{Relu}((\mathcal{F}^{2i+1})_3(x)+\frac{v_i-v_{i-1}}{q_{t(i)}/3}((\mathcal{F}^{2i+1})_1(x)-(\mathcal{F}^{2i+1})_2(x))) = \text{Relu}((\mathcal{F}^{2i})_1(x))$ when $x \leq x_{t(i)}$; and $(\mathcal{F}^{2i+2})_1(x) = \text{Relu}((\mathcal{F}^{2i})_3(x)+\frac{v_i-v_{i-1}}{q_{t(i)}/3}((\mathcal{F}^{2i+1})_1(x)-(\mathcal{F}^{2i+1})_2(x))) = \text{Relu}((\mathcal{F}^{2i})_1(x)+v_{i+1}-v_i)$ when $x \geq x_{t(i)+1}$.

So, we have that, if $t(i-1)+1 \leq j \leq t(i)$, there are $(\mathcal{F}^2)_1(x_j) = v_0$, $(\mathcal{F}^4)_1(x_j) = v_1-v_0+v_0 = v_1$, $(\mathcal{F}^6)_1(x_j) = v_2-v_1+v_1 = v_2$, ..., $(\mathcal{F}^{2i})_1(x_j) = v_i-v_{i-1}+v_{i-1} = v_i$; and $\mathcal{F}(x_j) = (\mathcal{F}^{2[N/\sqrt{N}]})_1(x_j) = (\mathcal{F}^{2[N/\sqrt{N}]-2})_1(x_j) = \cdots = (\mathcal{F}^{2i})_1(x_j) = v_i$.

So, by the definition of $t(i)$, we have that $\mathcal{F}(x_j) = v_{[\frac{j}{\sqrt{N}}]}$, such $\mathcal{F}$ is what we what and the lemma is proved. $\qquad\square$

## B.3 Label determination

The purpose of this part is to use the values to which the compressed data are mapped, mentioned in the above section, to determine the labels of the data.

Assuming $x_i$ is compressed to $c_i$ where $c_i \geq 1$ is given in section B.1. Value $v_i$ in section B.2 is designed as: $v_i = \overline{[c_{i[\sqrt{N}]+1}]\ldots[c_{(i+1)[\sqrt{N}]}]}$, where we treat $[c_j]$ as a $w$ digit number for all $j$ ($w$ is a given number). If there exist not enough digits for some $c_j$, we fill in 0 before it, and we use $\overline{ab}$ to denote the integer by putting $a$ and $b$ together.

First, prove a lemma.

**Lemma B.4.** *For a given N, there exists a network $f : \mathbb{R} \to \mathbb{R}^2$ with width 4 and at most $O(w)$ parameters such that, for any $w$ digit number $a_i > 0$, we have $f(\overline{a_1a_2\ldots a_N}) = (a_1, \overline{a_2\ldots a_N})$.*

*Proof.* Firstly, we show that, for any $a > b > 0$, there exists a network $F_{a,b}(x) : \mathbb{R}^+ \to \mathbb{R}^+$ with depth 2 and width 3, such that $F_{a,b}(x) = x$ when $x \in [0,a]$, and $F_{a,b}(x) = x - a$ when $x \in [a+b, 2a]$.

We just need to take $F_{a,b}(x) = \text{Relu}(x) - a/b\text{Relu}(x-a) + a/b\text{Relu}(x-(a+b))$.1 It is easy to verify that this is what we want.

Now, let $q \in N^+$ satisfy $2^q \leq 10^{w+1} - 1$ an $2^{q+1} > 10^{w+1}$ and $p < \frac{1}{10^{wN}}$. We consider the following network:
$$F = F_{2^0,p} \circ F_{2^1,p} \cdots \circ F_{2^{q-1},p} \circ F_{2^q,p},$$
and show that, $F(\overline{a_1a_2\ldots a_N}/10^{w(N-1)}) = \overline{a_2\ldots a_N}/10^{w(N-1)}$.

Firstly, we have $F_{2^q,p}(\overline{a_1a_2\ldots a_N}/10^{w(N-1)}) = \overline{a_1(q)a_2\ldots a_N}/10^{w(N-1)}$, where $a_1(q) = a_1$ if $\overline{a_1a_2\ldots a_N}/10^{w(N-1)} \leq 2^q$ and $a_1(q) = a_1 - 2^q$ if $\overline{a_1a_2\ldots a_N}/10^{w(N-1)} > 2^q + p$. Just by the definition of $q$, we know that there must be $\overline{a_1a_2\ldots a_N}/10^{w(N-1)} \leq 2^{q+1}$. Further by the definition of $p$, one of the following two inequalities is true:

$\overline{a_1a_2\ldots a_N}/10^{w(N-1)} < 2^q$ or $\overline{a_1a_2\ldots a_N}/10^{w(N-1)} > 2^q + p$.

So using the definition of $F_{2^q,p}$, we get the desired result.

Similarly as before, for $k = q-1, q-2, \ldots, 0$, we have $F_{2^k,p}(\overline{a_1(k+1)a_2\ldots a_N}/10^{w(N-1)}) = \overline{a_1(k)a_2\ldots a_N}/10^{w(N-1)}$, where $a_1(k) = a_1(k+1)$ if $\overline{a_1(k+1)a_2\ldots a_N}/10^{w(N-1)} \leq 2^k$ and $a_1(k) = a_1(k+1) - 2^k$ if $\overline{a_1(k+1)a_2\ldots a_N}/10^{w(N-1)} > 2^k + p$.

Then we have the following result: $a_1(k) < 2^k$ for any $k = 0, 1, \ldots, q$. By the definition, it is easy to see that $a_1 < 2^{q+1}$. If $a_1 < 2^q$, then $a_1(q) \leq a_1 < 2^q$; if $a_1 \geq 2^q$, then $\overline{a_1a_2\ldots a_N}/10^{w(N-1)} > 2^q + p$, so $a_1(q) = a_1 - 2^q < 2^{q+1} - 2^q = 2^q$. Thus $a_1(q) < 2^q$. When $a_1(t) < 2^t$ for a $t \in [q]$, similar as before, we have $a_1(t-1) < 2^{t-1}$. And $t = q$ is proved, so we get the desired result.

It is easy to see that, $a_1(k)$ are non negative integers, so there must be $F(\overline{a_1a_2\ldots a_N}/10^{w(N-1)}) = \overline{a_1(0)a_2\ldots a_N}/10^{w(N-1)} = \overline{a_2\ldots a_N}/10^{w(N-1)}$, by $a_1(0) < 2^0 = 1$, which implies $a_1(0) = 0$.

Now we construct a network $F_b$ as follows:

$F_b(x) = F_{b1} \circ F_{b1}(x)$ such that:

$F_{b1}(x) : \mathbb{R} \to \mathbb{R}^2$ and $F_{b1}(x) = (F(x/10^{w(N-1)}), x)$ where $x$ is defined as before.

$F_{b2}(x) : \mathbb{R}^2 \to \mathbb{R}^2$ and $F_{b2}((x_1, x_2)) = (x_2/10^{w(N-1)} - x_1, x_1 * 10^{w(N-1)})$.

Now we verify that $\mathcal{F}_b$ is what we want.

By the structure of $F$, $\mathcal{F}_b$ has width 4 and depth $O(w)$, so there are at most $O(w)$ parameters.

It is easy to see that $F_{b1}(\overline{a_1 a_2 \ldots a_N}) = (\overline{a_2 \ldots a_N}/10^{w(N-1)}, \overline{a_1 a_2 \ldots a_N})$. Then by the definition of $F_{b2}(x)$, we have $F_b(x) = (a_1, \overline{a_2 \ldots a_N})$, this is what we want. The lemma is proved. $\qquad \square$

By the preceding lemma, we have the following lemma.

**Lemma B.5.** *There is a network $\mathbb{R}^2 \to \mathbb{R}$ with at most $O(Nw)$ parameters and width 6, and for any $\{a_i\}_{i=1}^N$ where $a_j$ is a $w$ digit number and $a_j \geq 1$, which satisfies $f(x, \overline{a_1 a_2 \ldots a_N}) > 0.1$ if $|x - a_k| < 1$ for some $k \in [N]$, and $f(x, \overline{a_1 a_2 \ldots a_N}) = 0$ if $|x - a_k| \geq 1.1$ for all $k \in [N]$.*

*Proof.* The proof idea is as follows: First, we use $x$ and $\overline{a_1 a_2 \ldots a_N}$ to judge if $|x - a_1| < 1$ as follows: Using lemma B.4, we calculate $a_1$ and $\overline{a_2 \ldots a_N}$ and then calculate $|x - a_1|$.

If $|x - a_1| < 1$, then we let the network output a positive number; if $|x - a_1| \geq 1$, then calculate $\overline{a_2 \ldots a_N}$, and use $x$ and $\overline{a_2 \ldots a_N}$ to repeat the above process until all $|x - a_i|$ have been calculated.

The specific structure of the network is as follows:

**step 1:** Firstly, for a given $N$, we introduce a sub-network $f_s : \mathbb{R}^2 \to \mathbb{R}^2$, which satisfies $(f_s)_1(x, \overline{a_1 a_2 \ldots a_N}) > 0.1$ if $|x - a_1| < 1$, and $f_s(x, \overline{a_1 a_2 \ldots a_N}) = 0$ if $|x - a_1| \geq 1.1$, and $(f_s)_2(x, \overline{a_1 a_2 \ldots a_N}) = \overline{a_2 \ldots a_N}$. And $f_s$ has $O(w)$ parameters and width 5.

The first part of $f_s$ is to calculate $a_1$ and $\overline{a_2 \ldots a_N}$ by lemma B.4. We also need to keep $x$, and the network has width 5. The second part of $f_s$ is to calculate $|x - a_1|$ and keep $\overline{a_2 \ldots a_N}$ by using $|x| = \text{Relu}(x) + \text{Relu}(-x)$, which has width 4. The output of $f_s$ is $\text{Relu}(1.1 - |x - a_1|)$. Easy to check that this is what we want.

**step 2:** Now we build the $f$ mentioned in the lemma.

Let $f = g \circ f_N \circ f_{N-1} \cdots \circ f_1$.

For each $i \in [N]$, we will let the input of $f_i$ which is also the output of $f_{i-1}$ when $i > 1$ be the form $(x, \overline{a_i a_{i+1} \ldots a_N}, q_i)$, where $q_1 = 0$. The detail is as follows:

For $i \in [N]$, in $f_i$, construct $f_s(x, \overline{a_i a_{i+1} \ldots a_N})$ at first, and then let $q_{i+1} = q_i + (f_s)_1(x, \overline{a_i a_{i+1} \ldots a_N})$, to keep $q_i$ in each layer, where we need one more width than $f_s$. Then, output $(x, \overline{a_{i+1} a_{i+2} \ldots a_N}, q_{i+1})$, which is also the input of $(i + 1)$-th part.

The output of $f$ is $q_{N+1}$, that is, $g(x, 0, q_{N+1}) = q_{N+1}$. Now, we show that, $f$ is what we want.

(1): $f$ has at most $O(Nw)$ parameters and width 6, which is obvious, because each part $f_i$, $f_i$ has $O(w)$ parameters by lemma B.4, and $f$ has at most $N$ parts, so we get the result.

(2): $f(x, \overline{a_1 a_2 \ldots a_N}) > 0.1$ if $|x - a_k| < 1$ for some $k$.

This is because when $|x - a_k| < 1$, the $k$-th part will make $q_{k+1} = q_k + f_s(x, \overline{a_k a_{k+1} \ldots a_N}) > 0.1$, because $(f_s)_1(x, \overline{a_k a_{k+1} \ldots a_N}) > 0.1$ as said in step 1. Since $q_{j+1} = q_j + (f_s)_1 \geq q_j$, we have $f(x, \overline{a_1 a_2 \ldots a_N}) = q_{N+1} \geq q_{k+1} > 0.1$.

(3): $f(x, \overline{a_1 a_2 \ldots a_N}) = 0$ if $|x - a_k| \geq 1.1$ for all $k$.

This is because when $|x - a_k| \geq 1.1$, the $k$-th part will make $q_{k+1} = q_k + f_s(x, \overline{a_k a_{k+1} \ldots a_N}) = q_k$, because $f_s(x, \overline{a_k a_{k+1} \ldots a_N}) = 0$ as said in step 1. Since $f_s(x, \overline{a_k a_{k+1} \ldots a_N}) = 0$ for all $k$, we have $f(x, \overline{a_1 a_2 \ldots a_N}) = q_{N+1} = q_N + f_s(x, \overline{a_N}) = q_N = \cdots = q_0 = 0$. $\qquad \square$

## B.4 The proof of Theorem 4.1

Now, we will prove Theorem 4.1. As we mentioned before, three steps are required: data compression, data projection, and label determination. The proof is as follows.

*Proof.* Assume that $\mathcal{D}_{tr} = \{x_i\}_{i=1}^N$, without loss of generality, let $x_i \neq x_j$. Now, we show that there is a memorization network $\mathcal{F}$ of $\mathcal{D}_{tr}$ with $\overline{O}(\sqrt{N})$ parameters.

**Part One, data compression.**

The part is to compress the data in $\mathcal{D}_{tr}$ into $\mathbb{R}$. Let $w, b$ satisfy (1) and (2) in lemma B.1. Then, the first part of $\mathcal{F}$ is $f_1(x) = \text{Relu}(wx + b)$.

**Part two, data projection.**

Let $c_i = f_1(x_i)$, without loss of generality, we assume $c_i \leq c_{i+1}$ and $y_1 = 1$. We define $c_i'$ as: $c_i' = c_i$ if $x_i$ has label 1; otherwise $c_i' = c_1$.

Let $t(i) = \text{argmax}_{j \in [N]}\{[j/\sqrt{N}] = i\}$ and $v_k = \overline{[c_{t(k-1)+1}'][c_{t(k-1)+2}']\cdots[c_{t(k)}']}$.

In this part, the second part of $\mathcal{F}(x)$, named as $f_2(x) : \mathbb{R} \to \mathbb{R}^2$, need to satisfy $f_2(c_i) = (v_{[\frac{i}{\sqrt{N_0}}]}, c_i)$ for any $i \in [N]$.

By lemma B.3, a network with $O(\sqrt{N})$ parameters and width 4 is enough to map $x_i$ to $v_{[\frac{i}{\sqrt{N}}]}$ and for keeping the input, and one node is needed at each layer. So $f_2$ just need $O(\sqrt{N})$ parameters and width 5.

**Part Three, Label determination.**

In this part, we will use the $v_k$ mentioned in part two to output the label of input. The third part, nameed as $f_3(v, c)$, should satisfy that:

For $f_3(v_k, c)$, where $v_k = \overline{[c_{t(k-1)+1}'][c_{t(k-1)+2}']\cdots[c_{t(k)}']}$ is defined above, if $|c - c_q'| < 1$ for some $q \in [t(k-1) + 1, t(k)]$, then $f_3(v_k, c) > 0.1$; and $f_3(v_k, c) = 0$ if $|c - c_q'| \geq 1.1$ for all $q \in [t(k-1) + 1, t(k)]$.

Because the number of digits for $c_i$ is $O(\ln(nN/c))$ by (1) in lemma B.1 and lemma B.5, we know that such a network need $O(\sqrt{N}\ln(Nn/c))$ parameters.

**Construction of $\mathcal{F}$ and verify it:**

Let $\mathcal{F}(x) = f_3(f_2(f_1(x))) - 0.05$. We show that $\mathcal{F}$ is what we want.

(1): By parts one, two, three, it is easy to see that $\mathcal{F}$ has at most $O(\sqrt{N}\ln(Nn/c))$ parameters and width 6.

(2): $\mathcal{F}(x)$ is a memorization of $\mathcal{D}_{tr}$. For any $(x_i, y_i) \in \mathcal{D}_{tr}$, consider two sub-cases:

(1.1: if $y_i = 1$): Using the symbols in Part Two, $f_2(f_1(x_i))$ will output $(v_{[\frac{i}{\sqrt{N}}]}, f_1(x_i))$. Since $c_i' = c_i$ because $y_i = 1$, by part three, we have $f_3(f_2(f_1(x))) - 0.05 \geq 0.1 - 0.05 > 0$.

(1.2 if $y_i = -1$): By (2) in lemma B.1, for $\forall (z, 1) \in \mathcal{D}_{tr}$, we know that $|f_1(x_i) - [f_1(x_1)]| \geq |f_1(x_i) - f_1(x_1)| - |f_1(x_1) - [f_1(x_1)]| \geq 4 - 1 = 3$. So, by part three, we have $f_3(f_2(f_1(x_i))) = 0 - 0.05 < 0$.

**The Running Time:** In Part One, it takes $\text{poly}(B(\mathcal{D}_{tr}), \ln \epsilon)$ times to find such $w$ and $b$ with probability $1 - \epsilon$, as said in lemma B.1. In other parts, the parameters are calculated deterministically. We proved the theorem. $\square$

## C Proof of Theorem 4.3

*Proof.* It suffices to show that there exists a memorization algorithm $L$, such that if $\mathcal{D} \in \mathcal{D}(n, c)$ and $\mathcal{D}_{tr} \sim \mathcal{D}^N$, then the network $L(\mathcal{D}_{tr})$ has a constant number of parameters (independent of $N$). The construction has four steps.

**Step One:** Calculate the $\min_{(x,y_x),(z,y_z)\in\mathcal{D}_s}\|x-z\|_2$, name it as $c_0$.

**Step Two:** There is a $\mathcal{D}_{tr}\subset\mathcal{D}_{tr}$, such that:

(c1): For any $(x,y_x),(z,y_z)\in\mathcal{D}_s$, it holds $\|x-z\|_2>c_0/3$;

(c2): For any $(x,y_x)\in\mathcal{D}_{tr}$, it holds $\|x-z\|_2\le c_0/3$ for some $(z,y_z)\in\mathcal{D}_s$.

It is obvious that such $\mathcal{D}_s$ exists.

**Step Three:** We prove that $|\mathcal{D}_s|\le\frac{(1+2c_0/3)^n}{C_n(c_0/3)^n}$, where $C_n$ is the volume of unit ball in $\mathbb{R}^n$. Let $Q=\frac{(1+2c/3)^n}{C_n(c/3)^n}$, consider that $c_0\ge c$, so there are $|\mathcal{D}_s|\le Q$.

Let $B_2(x,r)=\{z:\|z-x\|_2\le r\}$, and $V(A)$ the volume of $A$.

Due to $\mathcal{D}_s\subset\mathcal{D}_{tr}\subset[0,1]^n\times\{-1,1\}$, so $\cup_{(x,y)\in\mathcal{D}_s}B_2(x,c_0/3)\in[-c_0/3,1+c_0/3]^n$. By condition (c1), we have $B_2(x,c_0/3)\cap B_2(z,c_0/3)=\emptyset$ for any $(x,y_x),(z,y_z)\in\mathcal{D}_s$, so we have $\sum_{(x,y)\in\mathcal{D}_s}V(B_2(x,c_0/3))\le(1+2c_0/3)^n$, which means $|\mathcal{D}_s|\le\frac{(1+2c_0/3)^n}{C_n(c_0/3)^n}<Q$.

**Step Four:** There is a robust memorization network [62] with at most $O(Qn)$ parameters for $\mathcal{D}_s$ with robust radius $c_0/3$, and this memorization network is a memorization of $\mathcal{D}_{tr}$.

By condition (c1), there is a robust memorization network $\mathcal{F}_{rm}$ with $O(|\mathcal{D}_s|n)$ parameters for $\mathcal{D}_s$ with radius $c_0/3$ [62]. By step three, we have $|\mathcal{D}_s|\le Q$, so that such a network has at most $O(Qn)$ parameters.

By condition (c2), for any $(x,y_x)\in\mathcal{D}_{tr}$, there is a $(z,y_z)\in\mathcal{D}_s$ satisfying $\|x-z\|_2\le c_0/3$. Firstly, there must be $y_x=y_z$, because the distribution $\mathcal{D}$ has separation bound $c_0$, and if $y_x\ne y_z$ then $\|x-z\|_2\ge c_0>c_0/3$. Then, since robust memorization $\mathcal{F}_{rm}$ has robust radius $c_0/3$, we have $\mathrm{Sgn}(\mathcal{F}_{rm}(x))=\mathrm{Sgn}(\mathcal{F}_{rm}(z))=y_z=y_x$, so $\mathcal{F}_{rm}$ is a memorization network of $\mathcal{D}_{tr}$. The theorem is proved. $\square$

# D Proof for Theorem 5.1

In this section, we will prove that networks with small width cannot have a good generalization for some distributions. For a given width $w$, we will construct a distribution on which any network with width $w$ will have poor generalization. The proof consists of the following parts.

## D.1 Disadvantages of network with small width

In this section, we demonstrate that a network with a small width may have some unfavorable properties. We have the following simple fact.

**Lemma D.1.** *Let the first transition weight matrix of network $\mathcal{F}$ be $W$. Then if $Wx=Wz$, we have $\mathcal{F}(x)=\mathcal{F}(z)$.*

If $W$ is not full-rank, then there exist $x$ and $z$ satisfying $Wx=Wz$. Moreover, if $x$ and $z$ have different labels, according to lemma D.1, we have $\mathcal{F}(x)=\mathcal{F}(z)$, so there must be an incorrect result given between $\mathcal{F}(x)$ and $\mathcal{F}(z)$.

According to the theorem of matrices decomposition, we also have the following fact.

**Lemma D.2.** *Let the first transition weight matrix of network $\mathcal{F}:\mathbb{R}^n\to\mathbb{R}$ be $W$. If $W$ has width $w<n$, then exists a $W_1\in\mathbb{R}^{w\times n}$, whose rows are orthogonal and unit such that $W_1x=W_1z$ implies $\mathcal{F}(x)=\mathcal{F}(z)$.*

*Proof.* Using matrix decomposition theory, we can write $W=NW_1$, where $N\in\mathbb{R}^{w\times w}$ and $W_1\in\mathbb{R}^{w\times n}$ and the rows of $W_1$ are orthogonal to each other and unit.

Next, we only need to consider $W_1$ as the first transition matrix of the network $\mathcal{F}$ and use lemma D.1. $\square$

At this point, we can try to construct a distribution where any network with small width will have poor generalization.

## D.2 Some useful lemmas

In this section, we introduce some lemmas which are used in the proof in section D.3.

**Lemma D.3.** *Let $B(r)$ be the ball with radius $r$ in $\mathbb{R}^n$. For any given $\delta > 0$, let $\epsilon = 2\delta/n$. Then we have $\frac{V(B(\sqrt{1-\epsilon}r))}{V(B(r))} > 1 - \delta$.*

*Proof.* we have $\frac{V(B(\sqrt{1-\epsilon}r))}{V(B(r))} = (1 - \epsilon)^{n/2} \geq 1 - n\epsilon/2 = 1 - \delta$. $\qquad\square$

For $w \in \mathbb{R}^{a,b}$ and $q \in \mathbb{R}^a$, let $q \circ w = \sum_{i=1}^{a} q_i w_i$, where $q_i$ is the $i$-th weight of $q$, $w_i$ is the $i$-th row of $w_i$. Then we have

**Lemma D.4.** *Let $W \in \mathbb{R}^{w \times n}$, and its rows are unit and orthogonal.*

*(1): For any $q_1 \neq q_2 \in \mathbb{R}^w$, we have*
$$\{x \in \mathbb{R}^n : Wx = W(q_1 \circ W)\} \cap \{x \in \mathbb{R}^n : Wx = W(q_2 \circ W)\} = \emptyset.$$

*(2): If $S$ is the unit ball in $\mathbb{R}^n$, then $S = \cup_{q \in \mathbb{R}^w, ||q||_2 \leq 1}\{x \in \mathbb{R}^n : Wx = W(q \circ W), x \in S\}$.*

*(3): For any $q \in \mathbb{R}^w$, $\{x \in \mathbb{R}^n : Wx = W(q \circ W), x \in S\}$ is a ball in $\mathbb{R}^{n-w}$ with volume $(1 - ||q||_2^2)^{(n-w)/2} C_{n-w}$, where $C_i$ is the volume of the unit ball in $\mathbb{R}^i$.*

*Proof.* First, we define an orthogonal coordinate system $\{W_i\}_{i=1}^{n}$ in $\mathbb{R}^n$. Let $W_i$ be the $i$-th row of $W$ when $i \leq w$. When $i > w$, let $W_i$ be a unit vector orthogonal with all $W_j$ where $j < i$.

Then for all $x \in \mathbb{R}^n$, we say $\widetilde{x}_i$ is the $i$-th weight of $x$ under such coordinate system. Then, $Wx = Wz$ if and only $\widetilde{x}_i = \widetilde{z}_i$ for $i \in [w]$.

Now, we can prove the lemma.

(1): The first weight $w$ of $q_1 \circ W$ under orthogonal coordinate system $\{W_i\}_{i=1}^{n}$ is $q_1$, so if $x \in \{x \in \mathbb{R}^n : Wx = W(q_1 \circ W)\}$, we have $\widetilde{x}_i = (q_1)_i$ for $i \in [w]$.

The first $w$ weight of $q_2 \circ W$ under orthogonal coordinate system $\{W_i\}_{i=1}^{n}$ is $q_2$, so if $x \in \{x \in \mathbb{R}^n : Wx = W(q_2 \circ W)\}$, we have $\widetilde{x}_i = (q_2)_i$ for $i \in [w]$. Because $q_1 \neq q_2 \in \mathbb{R}^w$, we get the result.

(2): For any $x \in \mathbb{R}^n$, let $q(x) = (\widetilde{x_1}, \widetilde{x_2}, \ldots, \widetilde{x_w}) \in \mathbb{R}^w$. It is easy to see that $||x||_2 = \sqrt{\sum_{i=1}^{n} \widetilde{x}_i^2}$, so $||q(x)||_2 \leq 1$ when $||x||_2 \leq 1$.

Now we verify that: for any $s \in S$, we have $s \in \{x \in \mathbb{R}^n : Wx = W(q(s) \circ W), x \in S\}$.

Firstly, we have $Ws = \sum_{i=1}^{w} < w_i, \sum_{i=1}^{N} \widetilde{s}_i w_i >= \sum_{i=1}^{w} \widetilde{s}_i$.

Secondly, we have $W(q(s) \circ W) = \sum_{i=1}^{w} < w_i, \sum_{i=1}^{w} \widetilde{s}_i w_i >= \sum_{i=1}^{w} \widetilde{s}_i$. So $Ws = W(q(s) \circ W)$, resulting in $s \in \{x \in \mathbb{R}^n : Wx = W(q(s) \circ W), x \in S\}$, which implies that $S = \cup_{q \in \mathbb{R}^w, ||q||_2 \leq 1}\{x \in \mathbb{R}^n : Wx = W(q \circ W), x \in S\}$.

(3): By the proof of (2), we know that if $x$ satisfies $\widetilde{x}_i = q_i$ for $i \in [w]$, then $x \in \{x \in \mathbb{R}^n : Wx = W(q \circ W)\}$. By (1), $\{x \in \mathbb{R}^n : Wx = W(q \circ W)\}$ will not intersect for different $q$. Therefore, $x \in \{x \in \mathbb{R}^n : Wx = W(q \circ W)\}$ equals $\widetilde{x}_i = q_i$ for $i \in [w]$.

Since $||x||_2 = \sqrt{\sum_{i=1}^{n} \widetilde{x}_i^2}$, when $x \in \{x \in \mathbb{R}^n : Wx = W(q \circ W)\}$, we have $\widetilde{x}_i = q_i$ for $i \in [w]$, so $\sum_{i=w+1}^{n} \widetilde{x}_i^2 = ||x||_2^2 - ||q||_2^2$, and such $n - w$ weight is optional.

Therefore, $\{x \in \mathbb{R}^n : Wx = W(q \circ W), x \in S\}$ is a ball in $\mathbb{R}^{n-w}$ with radius $\sqrt{1 - ||q||_2^2}$, so we get the result. $\qquad\square$

**Lemma D.5.** *Let $r_3 > r_2 > r_1$, $n \geq 1$ and $x \leq r_1$, then $\frac{(r_3-x)^n - (r_2-x)^n}{(r_1-x)^n} \geq \frac{r_3^n - r_2^n}{r_1^n}$.*

*Proof.* Let $f(x) = \frac{(r_3-x)^n - (r_2-x)^n}{(r_1-x)^n}$. We just need to prove $f(x) \geq f(0)$ when $x \leq r_1$. We calculate the derivative $f(x)$ at first:
$$f'(x) = \frac{((r_3-x)^n - (r_2-x)^n)'(r_1-x)^n - ((r_3-x)^n - (r_2-x)^n)((r_1-x)^n)'}{(r_1-x)^{2n}}.$$

It is easy to calculate that $((r_3 - x)^n - (r_2 - x)^n)' = -n((r_3 - x)^{n-1} - (r_2 - x)^{n-1})$ and $((r_1 - x)^n)' = -n(r_1 - x)^{n-1}$. Putting this into the above equation, we have

$$f'(x) = -P(x)(((r_3 - x)^{n-1} - (r_2 - x)^{n-1})(r_1 - x) - ((r_3 - x)^n - (r_2 - x)^n))$$

Where $P(x)$ is a positive value about $x$. Since

$$
\begin{aligned}
& ((r_3 - x)^{n-1} - (r_2 - x)^{n-1})(r_1 - x) - ((r_3 - x)^n - (r_2 - x)^n) \\
= \ & -(r_3 - x)^{n-1}(r_3 - r_1) + (r_2 - x)^{n-1}(r_2 - r_1) \\
\leq \ & 0
\end{aligned}
$$

we have $f'(x) \geq 0$, resulting in $f(x) \geq f(0)$. The lemma is proved. $\qquad\square$

**Lemma D.6.** *Let $a > b > 1$, $n > m \geq 1$. If $a^n - b^n = 1$. Then $a^m - b^m \leq 1$.*

*Proof.* We have $1 = a^n - b^n \geq b^{n-m}(a^m - b^m) > a^m - b^m$. $\qquad\square$

**Lemma D.7.** *Let $a > qb$ where $q < 1$ and $a, b > 0$. Then $\min\{a, b\} \geq qb$.*

*Proof.* When $\min\{a, b\} = b$, by $q < 1$, the result is obvious. When $\min\{a, b\} = a$, by $a > qb$, the result is obvious. $\qquad\square$

**Lemma D.8.** *For any $w > 0$, there exist $r_1, r_2, r_3$ and $n$ such that*

*(1): $r_3^n - r_2^n = r_1^n$;*

*(2): $r_3^{n-w} - r_2^{n-w} \geq 0.99 r_1^{n-w}$.*

*Proof.* Because the equations are all homogeneous, without loss of generality, we assume that $r_1 = 1$. We take $\alpha = 2^{1/n} - 1$, $\beta + \alpha = 3^{1/n} - 1$, and $n$ to satisfy $3^{w/n} < 1.001$. Let $r_2 = 1 + \alpha$, $r_3 = 1 + \alpha + \beta$. We show that this is what we want.

At first, we have $r_3^n - r_2^n = (1 + \alpha + \beta)^n - (1 + \alpha)^n = 3 - 2 = 1 = r_1^n$. We also have $(1 + \alpha + \beta)^w < 1.001$, named (k1). So we have

$$
\begin{aligned}
& r_3^{n-w} - r_2^{n-w} \\
= \ & (1 + \alpha + \beta)^{n-w} - (1 + \alpha)^{n-w} \\
= \ & \frac{(1+\alpha+\beta)^{n-w}(1+\alpha)^w - (1+\alpha)^n}{(1+\alpha)^w} \\
\geq \ & \frac{(1+\alpha+\beta)^{n-w}(1+\alpha)^w - (1+\alpha)^n}{1.001} \ (by \ (k1)) \\
= \ & \frac{(1+\alpha+\beta)^n - (1+\alpha+\beta)^{n-w}((1+\alpha+\beta)^w - (1+\alpha)^w) - (1+\alpha)^n}{1.001} \\
\geq \ & \frac{(1+\alpha+\beta)^n - (1+\alpha)^n - 0.001(1+\alpha+\beta)^n}{1.001} \ (by \ (k1)) \\
= \ & \frac{(1+\alpha+\beta)^n - (1+\alpha)^n - 0.003}{1.001} \\
= \ & \frac{1 - 0.003}{1.001} \\
\geq \ & 0.99.
\end{aligned}
$$

The lemma is proved. $\qquad\square$

### D.3 Construct the distribution

In this section, we construct the distribution in Theorem 5.1.

**Definition D.9.** Let $q$ be a point in $[0, 1]^n$, $0 < r_1 < r_2 < r_3$, and we define $B_2^k(z, t) = \{x \in \mathbb{R}^k : ||x - z||_2 \leq t\}$, where $k \in N^+$, $z \in \mathbb{R}^k$ and $t \geq 0$.

The distribution $\mathcal{D}(n, q, r_1, r_2, r_3)$ is defined as:

(1): This is a distirbution on $\mathbb{R}^n \times \{-1, 1\}$.

(2): A point has label 1 if and only if it is in $B_2^n(q, r_1)$. A point has label -1 if and only if it is in $B_2^n(q, r_3)/B_2^n(q, r_2)$.

(3): The points with label 1 or -1 satisfy the uniform distribution, and let the density function be $f(x) = \lambda = \frac{1}{V(B_2^n(q, r_3)) - V(B_2^n(q, r_2)) + V(B_2^n(q, r_1))}$.

We now prove Theorem 5.1.

*Proof.* Use the notations in Definition D.9.

Now, we let $r_i, q, n, w$ satisfy:
(c1): $B_2^n(q, r_3) \in [0,1]^n$;
(c2): $r_3^n - r_2^n = r_1^n$;
(c3): $r_3^{n-w} - r_2^{n-w} \geq 0.99 r_1^{n-w}$.
Lemma D.8 ensures that such $r_i, q, n$ exist.

Let distribution $\mathcal{D} = \mathcal{D}(n, q, r_1, r_2, r_3)$, where $\mathcal{D}(n, q, r_1, r_2, r_3)$ is given in Definition D.9. Now, we show that $\mathcal{D}$ is what we want. We prove that for any given $\mathcal{F}$ with width $w$, we have $A_{\mathcal{D}}(\mathcal{F}) < 0.51$.

Firstly, we define some symbols. Using lemma D.2, let $W \in \mathbb{R}^{w \times n}$ whose rows are unit and orthogonal and satisfy that $Wx = Wz$ implying $\mathcal{F}(x) = \mathcal{F}(z)$.

Then define $S_{1,x} = \{z : Wz = Wx, z \in B_2^n(q, r_1)\}$ and $S_{2,x} = \{z : Wz = Wx, z \in B_2^n(q, r_3)/B_2^n(q, r_2)\}$.

By lemma D.2, we know that, for any given $x$, the points in $S_{1,x} \cup S_{2,x}$ have the same output after inputting to $\mathcal{F}$, but the points in $S_{1,x}$ have label 1 and the points in $S_{2,x}$ have label -1. So $\mathcal{F}$ must give the wrong label to the point in $S_{1,x}$ or $S_{2,x}$.

The proof is then divided into two parts.

**Part One:** Let $h \in B_2^w(0, r_1)$, and $x(h) = q + h \circ W \in \mathbb{R}^n$, where $\circ$ is defined in section D.2. Consider that for any given $h$, $\mathcal{F}$ must give the wrong label to the point in $S1_{x(h)}$ or $S2_{x(h)}$, we have that $\mathcal{F}$ will give the wrong label with probability at least $\min\{\mathbb{P}_{(x,y)\sim\mathcal{D}}(x \in S1_{x(h)}), \mathbb{P}_{(x,y)\sim\mathcal{D}}(x \in S2_{x(h)})\}$. So, now we only need to sum these values about $h$.

For any different $h_1, h_2 \in B_2^w(0, r_1)$, we have $S1_{x(h_1)} \cap S1_{x(h_2)} = \emptyset$, $S2_{x(h_1)} \cap S2_{x(h_2)} = \emptyset$, and $\cup_{h \in B_2^w(0,r_1)} S1_{x(h)} = B_2^n(q, r_1)$. By (1) and (2) in lemma D.4. Proof is similar for $S2_{x(h)}$. Then, by the volume of $S1_{x(h)}, S2_{x(h)}$ calculated in lemma D.4, we know that, the probability of $\mathcal{F}$ producing an error on distribution $\mathcal{D}$ is at least

$$\int_{h \in B_2^w(0,r_1)} \min\{\mathbb{P}_{(x,y)\sim\mathcal{D}}(x \in S1_{x(h)}), \mathbb{P}_{(x,y)\sim\mathcal{D}}(x \in S2_{x(h)})\}$$
$$= \lambda C_{n-w} \int_{x \in B_2^w(0,r_1)} \min\{(r_1^2 - ||x||_2^2)^{(n-w)/2},$$
$$(r_3^2 - ||x||_2^2)^{(n-w)/2} - (r_2^2 - ||x||_2)^{(n-w)/2}\}dx$$

where $C_{n-w}$ is the volume of the unit ball in $\mathbb{R}^{n-w}$ as mentioned in lemma D.4. Next, we will estimate the lower bound of this value

**Part Two:** Firstly, by lemma D.5, we know that $\frac{(r_3^2 - ||x||_2^2)^{(n-w)/2} - (r_2^2 - ||x||_2^2)^{(n-w)/2}}{(r_1^2 - ||x||_2^2)^{(n-w)/2}} \geq \frac{(r_3^2)^{(n-w)/2} - (r_2^2)^{(n-w)/2}}{(r_1^2)^{(n-w)/2}}$.

Then, by lemma D.6 and (c2), we know that $\frac{(r_3^2)^{(n-w)/2} - (r_2^2)^{(n-w)/2}}{(r_1^2)^{(n-w)/2}} \leq 1$. Thus by lemma D.7, we have

$$\lambda C_{n-w} \int_{x \in B_2^w(0,r_1)} \min\{(r_1^2 - ||x||_2^2)^{(n-w)/2}, (r_3^2 - ||x||_2^2)^{(n-w)/2} - (r_2^2 - ||x||_2^2)^{(n-w)/2}\}dx$$
$$= \lambda C_{n-w} \int_{x \in B_2^w(0,r_1)} \min\{(r_1^2 - ||x||_2^2)^{(n-w)/2},$$
$$\frac{(r_3^2 - ||x||_2^2)^{(n-w)/2} - (r_2^2 - ||x||_2^2)^{(n-w)/2}}{(r_1^2 - ||x||_2^2)^{(n-w)/2}}(r_1^2 - ||x||_2^2)^{(n-w)/2}\}dx$$
$$\geq \lambda C_{n-w} \int_{x \in B_2^w(0,r_1)} \min\{(r_1^2 - ||x||_2^2)^{(n-w)/2},$$
$$\frac{(r_3^2)^{(n-w)/2} - (r_2^2)^{(n-w)/2}}{(r_1^2)^{(n-w)/2}}(r_1^2 - ||x||_2^2)^{(n-w)/2}\}dx$$
$$\geq \lambda C_{n-w} \frac{(r_3^2)^{(n-w)/2} - (r_2^2)^{(n-w)/2}}{(r_1^2)^{(n-w)/2}} \int_{x \in B_2^w(0,r_1)} (r_1^2 - ||x||_2^2)^{(n-w)/2}dx$$
$$= \frac{(r_3^2)^{(n-w)/2} - (r_2^2)^{(n-w)/2}}{(r_1^2)^{(n-w)/2}} \mathbb{P}_{(x,y)\sim\mathcal{D}}(y = 1).$$

From $r_3^n - r_2^n = r_1^n$, we know that $\lambda V(B_2^n(q, r_1)) = \lambda(V(B_2^n(q, r_3)) - V(B_2^n(q, r_2))) = 0.5$, so $\mathbb{P}_{(x,y)\sim\mathcal{D}}(y = -1) = \mathbb{P}_{(x,y)\sim\mathcal{D}}(y = -1) = 0.5$, and further consider the (c3), we have

$$\frac{(r_3^2)^{(n-w)/2} - (r_2^2)^{(n-w)/2}}{(r_1^2)^{(n-w)/2}}\mathbb{P}_{(x,y)\sim\mathcal{D}}(y = 1)$$
$$\geq \quad 0.5\frac{(r_3^2)^{(n-w)/2} - (r_2^2)^{(n-w)/2}}{(r_1^2)^{(n-w)/2}}$$
$$\geq \quad 0.49.$$

The theorem is proved. □

# E    Proof of Theorem 5.3

Firstly, note that Theorem 5.3 cannot be proved by the following classic result.

**Theorem E.1** ([57]). *Let $\mathcal{D}$ be any joint distribution over $\mathbb{R}^n \times \{-1, 1\}$, $\mathcal{D}_{tr}$ a dataset of size $N$ selected i.i.d. from $\mathcal{D}$, and $\mathbf{H} = \{h : \mathbb{R}^n \to \mathbb{R}\}$ the hypothesis space. Then with probability at least $1 - \delta$,*

$$\sup_{h\in\mathbf{H}} |\mathcal{R}(h, \mathcal{D}) - \mathcal{R}(h, \mathcal{D}_{tr})| \geq \frac{\mathrm{Rad}_N(\mathbf{H})}{2} - O(\sqrt{\frac{\ln 1/\delta}{N}}),$$

*where $\mathcal{R}(h, \mathcal{D})$ is the population risk, $\mathcal{R}(h, \mathcal{D}_{tr})$ is the empirical risk, and $\mathrm{Rad}_N(\mathbf{H})$ is the Radermecher complexity of $\mathbf{H}$.*

Theorem E.1 is the classical conclusion about the lower bound of generalization error, and theorem 5.3 and Theorem E.1 are different. Firstly, Theorem E.1 is established on the basis of probability, whereas Theorem 5.3 is not. Secondly, Theorem E.1 highlights the existence of a gap between the empirical error and the generalization error for certain functions within the hypothesis space, and does not impose any constraints on the value of empirical error. However, memorization networks, which perfectly fit the training set, will inherently have a zero empirical error, so Theorem E.1 cannot directly address Theorem 5.3. Lastly, Theorem E.1 relies on Radermacher complexity, which can be challenging to calculate, while Theorem 5.3 does not have such a requirement.

For the proof of Theorem 5.3, we mainly follow the constructive approach of memorization network in [55], but during the construction process, we will also consider the accuracy of the memorization network. Our proof is divided into four parts.

## E.1    Data Compression

The general method of constructing memorization networks compresses the data into a low dimensional space at first, and we adopt this approach. We are trying to compress the data into 1-dimension space. However, we require the compressed data to meet some conditions, as stated in the following lemma.

**Lemma E.2.** *Let $\mathcal{D}$ be a distribution in $[0, 1]^n \times \{-1, 1\}$ with separation bound $c$ and density $r$, and $\mathcal{D}_{tr} \sim \mathcal{D}^N$. Then, there are $w \in \mathbb{R}^n$ and $b \in \mathbb{R}$ that satisfy:*
*(1): $O(nN^3r/c) \geq wx + b \geq 1$ for all $x \in [0, 1]^n$;*
*(2): $|wx - wz| \geq 4$ for all $(x, 1), (z, -1) \in \mathcal{D}_{tr}$;*
*(3): $\mathbb{P}_{(x,y)\sim\mathcal{D}}(\exists(z, y_z) \in \mathcal{D}_{tr}, |wx - wz| \leq 3) < 0.01$.*

*Proof.* Since distribution $\mathcal{D}$ is definition on $[0, 1]^n$, we have $c \leq 1$ and $r \geq 1$.

Because the density function of $\mathcal{D}$ is $r$, we have $\mathbb{P}_{(x,y)\sim\mathcal{D}}(x \in B_2(z, r_1)) \leq rV(B_2(z, r_1)) < r(2r_1)^n = \frac{1}{400N^2}$ for all $z \in \mathbb{R}^n$, where $r_1 = \frac{1}{2(400rN^2)^{1/n}}$. It is easy to see that $r_1 \leq 1$ because $r \geq 1$.

Then, we have the following two results:

**Result one:** Let $u \in \mathbb{R}^n$ be uniformly randomly sampled from the hypersphere $S^{n-1}$. Then we have $P(|\langle u, (x - z)\rangle| \geq \frac{c}{4N^2}\sqrt{\frac{8}{n\pi}}, \forall(x, -1), (z, 1) \in \mathcal{D}_{tr}) > 0.5$. The proof is similar to that of lemma B.1.

**Result Two:** Let $u \in \mathbb{R}^n$ be uniformly randomly sampled from the hypersphere $S^{n-1}$. Then
$\mathbb{P}_u(\mathbb{P}_{(x,y)\sim\mathcal{D}}(\exists(x_i, y_i) \in \mathcal{D}_{tr}, |\langle u, (x - x_i)\rangle| < \frac{r}{800N^2}\sqrt{\frac{8}{n\pi}}) < 0.01) > 0.5$.

Firstly, by lemma B.2, and take $T = 800N^2$, we can get that: for any given $v \in \mathbb{R}^n$, if $u \in \mathbb{R}^n$ be uniformly randomly sampled from the hypersphere $S^{n-1}$, then $P(|\langle u, v\rangle| < \frac{||v||_2}{800N^2}\sqrt{\frac{8}{n\pi}}) < \frac{1}{400N^2}$. Thus, by such inequality, the density of $\mathcal{D}$ and the definition of $r_1$, we have that:

$$
\begin{aligned}
& \mathbb{P}_{u,(x,y)\sim\mathcal{D}}(|\langle u, (x - v)\rangle| < \frac{r_1}{800N^2}\sqrt{\frac{8}{n\pi}}) \\
= \ & \mathbb{P}_{u,(x,y)\sim\mathcal{D}}(|\langle u, (x - v)\rangle| < \frac{r_1}{800N^2}\sqrt{\frac{8}{n\pi}}| \, ||x - v||_2 \geq r_1)\mathbb{P}_{(x,y)\sim\mathcal{D}}(||x - v||_2 \geq r_1) \\
& + \mathbb{P}_{u,(x,y)\sim\mathcal{D}}(|\langle u, (x - v)\rangle| < \frac{r_1}{800N^2}\sqrt{\frac{8}{n\pi}}| \, ||x - v||_2 < r_1)\mathbb{P}_{(x,y)\sim\mathcal{D}}(||x - v||_2 < r_1) \\
< \ & \mathbb{P}_{u,(x,y)\sim\mathcal{D}}(|\langle u, (x - v)\rangle| < \frac{||x-v||_2}{800N^2}\sqrt{\frac{8}{n\pi}}| \, ||x - v||_2 \geq r_1) + \mathbb{P}_{(x,y)\sim\mathcal{D}}(||x - v||_2 < r_1) \\
\leq \ & \mathbb{P}_u(|\langle u, (x - v)\rangle| < \frac{||x-v||_2}{800N^2}\sqrt{\frac{8}{n\pi}}) + \mathbb{P}_{(x,y)\sim\mathcal{D}}(||x - v||_2 < r_1) \\
< \ & \frac{1}{400N^2} + \frac{1}{400N^2} = 1/(200N^2).
\end{aligned}
$$

On the other hand, we have

$$
\begin{aligned}
& \mathbb{P}_{u,(x,y)\sim\mathcal{D}}(|\langle u, (x - v)\rangle| < \frac{r_1}{800N^2}\sqrt{\frac{8}{n\pi}}) \\
\geq \ & \mathbb{P}_u(\mathbb{P}_{(x,y)\sim\mathcal{D}}(|\langle u, (x - v)\rangle| < \frac{r_1}{800N^2}\sqrt{\frac{8}{n\pi}}) \geq 0.01/N) * 0.01/N.
\end{aligned}
$$

So, we have $\mathbb{P}_u(\mathbb{P}_{(x,y)\sim\mathcal{D}}(|\langle u, (x-v)\rangle| < \frac{r}{800N^2}\sqrt{\frac{8}{n\pi}}) \geq 0.01/N) < \frac{1}{200N^2}/(0.01/N) = 1/(2N)$. Name this inequality as (*).

On the other hand, we have

$$
\begin{aligned}
& \mathbb{P}_u(\mathbb{P}_{(x,y)\sim\mathcal{D}}(\exists(x_i, y_i) \in \mathcal{D}_{tr}, |\langle u, (x - x_i)\rangle| < \frac{r}{800N^2}\sqrt{\frac{8}{n\pi}}) < 0.01) \\
= \ & 1 - \mathbb{P}_u(\mathbb{P}_{(x,y)\sim\mathcal{D}}(\exists(x_i, y_i) \in \mathcal{D}_{tr}, |\langle u, (x - x_i)\rangle| < \frac{r}{800N^2}\sqrt{\frac{8}{n\pi}}) \geq 0.01)
\end{aligned}
$$

Then, if a $u \in \mathbb{R}^n$ satisfies $\mathbb{P}_{(x,y)\sim\mathcal{D}}(\exists(x_i, y_i) \in \mathcal{D}_{tr}, |\langle u, (x - x_i)\rangle| < \frac{r}{800N^2}\sqrt{\frac{8}{n\pi}}) \geq 0.01$, then we have $\mathbb{P}_{(x,y)\sim\mathcal{D}}(|u(x - x_i)| < \frac{r}{800N^2}\sqrt{\frac{8}{n\pi}}) \geq 0.01/N$ for some $(x_i, y_i) \in \mathcal{D}_{tr}$.

So taking $v$ as $x_i$ in inequality (*) and using the above result, we have

$$
\begin{aligned}
& \mathbb{P}_u(\mathbb{P}_{(x,y)\sim\mathcal{D}}(\exists(x_i, y_i) \in \mathcal{D}_{tr}, |\langle u, (x - x_i)\rangle| < \frac{r}{800N^2}\sqrt{\frac{8}{n\pi}}) < 0.01) \\
= \ & 1 - \mathbb{P}_u(\mathbb{P}_{(x,y)\sim\mathcal{D}}(\exists(x_i, y_i) \in \mathcal{D}_{tr}, |\langle u, (x - x_i)\rangle| < \frac{r}{800N^2}\sqrt{\frac{8}{n\pi}}) \geq 0.01) \\
\geq \ & 1 - \sum_{(x_i, y_i)\in\mathcal{D}_{tr}} \mathbb{P}_u(\mathbb{P}_{(x,y)\sim\mathcal{D}}(|\langle u, (x - x_i)\rangle| < \frac{r}{800N^2}\sqrt{\frac{8}{n\pi}}) \geq 0.01/N) \\
> \ & 1 - N\frac{1}{2N} = 0.5.
\end{aligned}
$$

So we get the result. This is what we want.

**Construct $w, b$ and verify their property**

Consider the fact: if $A(u), B(u)$ are two events about random variable $u$, and $\mathbb{P}_u(A(u) = True) > 0.5, \mathbb{P}_u(B(u) = True) > 0.5$, then there is a $u$, which makes events $A(u)$ and $B(u)$ occurring simultaneously. By the above fact and Results one and two, we have that there exist $||u||_2 = 1$ and $u \in \mathbb{R}^n$ such that $|\langle u, (x - z)\rangle| \geq \frac{c}{4N^2}\sqrt{\frac{8}{n\pi}}, \forall(x, -1), (z, 1) \in \mathcal{D}_{tr}$ and $\mathbb{P}_{(x,y)\sim\mathcal{D}}(\exists(x_i, y_i) \in \mathcal{D}_{tr}, |\langle u, (x - x_i)\rangle\rangle| < \frac{r}{800N^2}\sqrt{\frac{8}{n\pi}}) < 0.01$.

Now, let $w = max\{\frac{2400\sqrt{n}N^2}{r_1}, \frac{16\sqrt{n}N^2}{c}\}u$ and $b = ||w||_2\sqrt{n} + 1$, then we show that $w$ and $b$ are what we want:

(1): we have $O(nN^3) \geq wx + b \geq 1$ for all $x \in [0, 1]^n$.

Firstly, because $\mathcal{D}$ is defined in $[0, 1]^n \times \{-1, 1\}$, we have $||x||_2 \leq \sqrt{n}$, resulting in and $wx + b \geq b - ||w||_2\sqrt{n} \geq 1$.

On the other hand, using $c \leq 1$ and $r_1 \leq 1$, we have $|wx| \leq ||w||_2 \sqrt{n} \leq O(\frac{nN^2}{r_1 c})$, so $wx + b \leq |wx| + b \leq O(nN^3 r^{1/n}/c)$.

(2): We have $|w(x - z)| \geq 4$ for all $(x, 1), (z, -1) \in \mathcal{D}_{tr}$.

It is easy to see that $|w(x - z)| \geq |\frac{16\sqrt{n}N^2}{c} u(x - z)| = \frac{16\sqrt{n}N^2}{c}|u(x - z)|$. Because $|u(x - z)| \geq \frac{c}{4\sqrt{n}N^2}$, so $|w(x - z)| = \frac{16\sqrt{n}N^2}{c}|u(x - z)| \geq \frac{16\sqrt{n}N^2}{c} \frac{c}{4\sqrt{n}N^2} = 4$.

(3): we have $\mathbb{P}_{(x,y)\sim\mathcal{D}}(\exists(z, y_z) \in \mathcal{D}_{tr}, |wx - wz| \leq 3) < 0.01$.

Because $|w(x - z)| \geq \frac{2400\sqrt{n}N^2}{r_1}|u(x - z)| \geq |u(x - z)|$, and consider that $\mathbb{P}_{(x,y)\sim\mathcal{D}}(\exists(z, y_z) \in \mathcal{D}_{tr}, |u(x - z)| < \frac{r_1}{800N^2}\sqrt{\frac{8}{n\pi}}) < 0.01$, we get the result. So, $w$ and $b$ are what we want. and the lemma is proved. $\qquad\square$

### E.2 Data Projection

The purpose of this part is to map the compressed data to appropriate values. Let $w \in \mathbb{R}^n$ and $b \in \mathbb{R}$ be given, and $\mathcal{D}_{tr} = \{(x_i, y_i)\}_{i=1}^N$. Without losing generality, we assume that $wx_i < wx_{i+1}$.

In this section, we show that, after compressing the data into 1-dimension, we can use a network $\mathcal{F}$ to map $wx_i + b$ to $v_{[\frac{i}{[\sqrt{N}]}]}$, where $\{v_j\}_{j=0}^{[\frac{N}{[\sqrt{N}]}]}$ are the given values. Furthermore, $\mathcal{F}$ should also satisfy $\mathcal{F}(wx + b) \in \{v_j\}_{j=0}^{[\frac{N}{[\sqrt{N}]}]+1}$ for all $x \in [0, 1]^n$ except for a small portion.

This network has $O(\sqrt{N})$ parameters, as shown below.

**Lemma E.3.** *Let $w \in \mathbb{R}^n$ and $b \in \mathbb{R}$ be given, $\{v_j\}_{j=0}^{[\frac{N}{[\sqrt{N}]}]} \subset \mathbb{R}$ and $1 > \epsilon > 0$ be given.*

*Let $\mathcal{D}_{tr} = \{(x_i, y_i)\}_{i=1}^N$ and $\mathcal{D}_{tr} \sim \mathcal{D}^N$ where $\mathcal{D}$ is a distribution, and assume that $wx_i + b < wx_{i+1} + b$.*

*Then a network $\mathcal{F}$ with width $O(\sqrt{N})$, depth 2, and at most $O(\sqrt{N})$ parameters, can satisfy that:*

*(1): $\mathcal{F}(wx_i + b) = v_{[\frac{i}{\sqrt{N}}]}$ for all $i \in [N]$;*

*(2): $\mathbb{P}_{(x,y)\sim\mathcal{D}}(\mathcal{F}(wx + b) \in \{v_j\}_{j=0}^{[\frac{N}{[\sqrt{N}]}]}) \geq 1 - \epsilon$.*

*Proof.* Let $q_i = (wx_{i+1} + b) - (wx_i + b)$ and $q = \min_i\{q_i\}$. Then we consider the set of points $S_i = \{wx_i + b + \frac{q\epsilon}{2N} * j\}_{j=1}^{[N/\epsilon]+1}$, for any $i$. We have that:

$$\begin{aligned}
&\sum_{s \in S_i} \mathbb{P}_{(x,y)\sim\mathcal{D}}(wx + b \in (s - \frac{q\epsilon}{2N}/2, s + \frac{q\epsilon}{2N}/2)) \\
= \ &\mathbb{P}_{(x,y)\sim\mathcal{D}}(\exists s \in S_i, wx + b \in (s - \frac{q\epsilon}{2N}/2, s + \frac{q\epsilon}{2N}/2)) \\
\leq \ &1
\end{aligned}$$

Consider that $|S_i| \geq N/\epsilon$, so for any $i$, there is a $s_i \in S_i$, makes that $\mathbb{P}_{(x,y)\sim\mathcal{D}}(wx + b \in (s_i - \frac{q\epsilon}{2N}/2, s_i + \frac{q\epsilon}{2N}/2)) \leq \frac{\epsilon}{N}$.

And it is easy to see that $S_i$ satisfies the following result: if $z \in S_i$, then:

$$wx_i + b < wx_i + b + \frac{q\epsilon}{2N} \leq z \leq wx_i + b + \frac{q\epsilon}{2N}([N/\epsilon] + 1) < wx_i + b + q_i = wx_{i+1} + b.$$

So we have $(s_i - \frac{q\epsilon}{2N}/2, s_i + \frac{q\epsilon}{2N}/2) \in (wx_i + b, wx_{i+1} + b)$, Name this inequality as $(*)$.

Let $k = [\frac{N}{[\sqrt{N}]}]$ and $t(i) = \text{argmax}_{j\in[N]}\{[j/\sqrt{N}] = i\}$. Now, we define such a network:

$$\mathcal{F}(x) = \sum_{i=1}^k \frac{v_i - v_{i-1}}{\frac{q\epsilon}{2N}}(\text{Relu}(x - s_{t(i)} + \frac{q\epsilon}{2N}/2) - \text{Relu}(x - s_{t(i)} - \frac{q\epsilon}{2N}/2)) + v_0.$$

This network has width $2k$, depth 2 and $O(\sqrt{N})$ parameters. We can verify that such networks satisfy (1) and (2).

Verify (1): For a given $i \in [N]$, let $c(i) = [\frac{i}{\sqrt{N}}]$. Then, when $j < c(i)$, we have $t(j) < i$, so $s_{t(j)} + \frac{q\epsilon}{2N}/2 \le wx_{t(j)+1} + b \le wx_i + b$ (this has been shown in $(*)$), resulting in: $\frac{v_j - v_{j-1}}{\frac{q\epsilon}{2N}}(\text{Relu}(wx_i + b - s_{t(j)} + \frac{q\epsilon}{2N}/2) - \text{Relu}(wx_i + b - s_{t(j)} - \frac{q\epsilon}{2N}/2) = v_j - v_{j-1}$. When $j \ge c(i)$, similar to before, we have $s_{t(j)} - \frac{q\epsilon}{2N}/2 > wx_i + b$, resulting in $\frac{v_j - v_{j-1}}{\frac{q\epsilon}{2N}}(\text{Relu}(wx_i + b - s_{t(j)} + \frac{q\epsilon}{2N}/2) - \text{Relu}(wx_i + b - s_{t(j)} - \frac{q\epsilon}{2N}/2) = 0$. So $\mathcal{F}(x_i) = v_0 + (v_1 - v_0) + \cdots + (v_c(i) - v_{c(i)-1}) = v_{c(i)}$, this is what we want.

Verify (2): At first, we show that for any $x \in [0,1]^n$ satisying $wx + b \notin \cup_{i=1}^k(s_i - \frac{q\epsilon}{2N}/2, s_i + \frac{q\epsilon}{2N}/2)$, we have $\mathcal{F}(x) \in \{v_i\}$.

This is because: for any $x$ satisfies $wx + b \notin \cup_{i=1}^k(s_i - \frac{q\epsilon}{2N}/2, s_i + \frac{q\epsilon}{2N}/2)$, we have $\mathcal{F}(wx + b) = v_0 + (v_1 - v_0) + \cdots + (v_k - v_{k-1}) = v_k$, where $k$ satisfies $s_{t(k)} < wx + b$ and $k$ is the maximum. The proof is similar as above.

Second, we show that the probability of such $x$ is at least $1 - \epsilon$.

By $\mathbb{P}_{(x,y)\sim\mathcal{D}}(wx + b \in (s_i - \frac{q\epsilon}{2N}/2, s_i + \frac{q\epsilon}{2N}/2)) \le \frac{\epsilon}{N}$ for any $i$, we have $\mathbb{P}_{(x,y)\sim\mathcal{D}}(\exists i, wx + b \in (s_i - \frac{q\epsilon}{2N}/2, s_i + \frac{q\epsilon}{2N}/2)) \le \sum_{i=1}^k \mathbb{P}_{(x,y)\sim\mathcal{D}}(wx + b \in (s_i - \frac{q\epsilon}{2N}/2, s_i + \frac{q\epsilon}{2N}/2)) \le \epsilon/N * N = \epsilon$, this is what we want. So $\mathcal{F}$ is what we want. The lemma is proved. $\qquad\square$

### E.3 Label determination

This is the same as in section B.3.

### E.4 The proof of Theorem 5.3

Three steps are required: data compression, data projection, label determination. The specific proof is as follows.

*Proof.* Assume that $\mathcal{D}_{tr} = \{x_i\}_{i=1}^N$, without loss of generality, let $x_i \ne x_j$. Now, we show that there is a memorization network $\mathcal{F}$ of $\mathcal{D}_{tr}$ with $\overline{O}(\sqrt{N})$ parameters but with poor generalization.

**Part One, data compression.** The first part is to compress the data in $\mathcal{D}_{tr}$ into $\mathbb{R}$, let $w, b$ satisfy (1),(2),(3) in lemma E.2. Then, the first part of $\mathcal{F}$ is $f_1(x) = \text{Relu}(wx + b)$.

On the other hand, not just samples in $\mathcal{D}_{tr}$, all the data in $\mathbb{R}^n$ have been compressed into $\mathbb{R}$ by $f_1(x)$. By (3) in lemma E.2, we have $\mathbb{P}_{(x,y)\sim\mathcal{D}}(\exists (z, y_z) \in \mathcal{D}_{tr}, |wx - wz| \le 3) < 0.01$, resulting in, we have $\mathbb{P}_{(x,y)\sim\mathcal{D}}(|wx - wz| > 3 \ for \ \forall(z, y_z) \in \mathcal{D}_{tr}) > 0.99$. By the probability theory, we have

$$
\begin{aligned}
&\mathbb{P}_{(x,y)\sim\mathcal{D}}(|wx - wz| > 3 \ for \ \forall(z, y_z) \in \mathcal{D}_{tr} > 0.99) \\
= \ &\mathbb{P}_{(x,y)\sim\mathcal{D}}(|wx - wz| > 3 \ for \ \forall(z, y_z) \in \mathcal{D}_{tr} > 0.99, y = -1) + \\
&\mathbb{P}_{(x,y)\sim\mathcal{D}}(|wx - wz| > 3 \ for \ \forall(z, y_z) \in \mathcal{D}_{tr} > 0.99, y = 1) \\
> \ &0.99.
\end{aligned}
$$

Without losing generality, we assume that $\mathbb{P}_{(x,y)\sim\mathcal{D}}(\forall(z, y_z) \in \mathcal{D}_{tr}, |wx - wz| > 3, y = 1) > 0.99/2$, which represents the following fact. Define $S = \{x : x \ has \ label \ 1 \ and \ |wx - wz| > 3 \ for \ \forall(z, y_z) \in \mathcal{D}_{tr}\}$. Then the probability of points in $S$ is at least $0.99/2$. In the following proof, in order to make the network having bad generalization, we will make the network giving these points (the points in $S$) incorrect labels.

**Part two, data projection.**

Let $c_i = f_1(x_i)/$ Without losing generality, we will assume $c_i \le c_{i+1}$.

Now, assume that we have $N_0$ samples in $\mathcal{D}_{tr}$ with label 1, and $\{i_j\}_{j=1}^{N_0} \subset [N]$ such that $x_{i_j}$ has label 1, and $i_j < i_{j+1}$. Let $t(i) = \text{argmax}_{j\in[N]}\{[j/\sqrt{N_0}] = i\}$ and $v_k = [c_{i_{t(k-1)+1}}][c_{i_{t(k-1)+2}}] \ldots [c_{i_{t(k)}}]$. In this part, the second part of $\mathcal{F}(x)$, named as $f_2(x)$, need to satisfy $f_2(c_{i_j}) = (v_{[\frac{j}{\sqrt{N}}]}, c_{i_j})$. Furthermore, we also hope that $\mathbb{P}_{(x,y)\sim\mathcal{D}}(f_2(f_1(x))[1] \in \{v_i\}) \ge 0.999$, where $f_2(f_1(x))[i]$ is the $i$-th weight of $f_2(f_1(x))$, and $\mathbb{P}_{(x,y)\sim\mathcal{D}}(f_2(f_1(x))[2]) = f_1(x)$.

By lemma B.3, a network with $O(\sqrt{N})$ parameters and depth 2 is enough to calculate $v_{[\frac{j}{\sqrt{N}}]}$ by $c_{i_j}$, and the output in $\{v_i\}$ has probability 0.999. Retaining $c_i$ just need one node. So $f_2$ need $O(\sqrt{N})$ parameters.

**Part Three, Label determination.** In this part, we will use the $v_k$ mentioned in part two to output the label of inputs. The third part, named as $f_3(v, c)$, should satisfy that for $f_3(v_k, c)$, where $v_k = \overline{[c_{i_{t(k-1)+1}}][c_{i_{t(k-1)+2}}]\ldots[c_{i_{t(k)}}]}$ as mentioned above, if $|c - c_{i_q}| < 1$ for some $q \in [t(k-1)+1, t(k)]$, then $f_3(v_k, c) > 0.1$, and $f_3(v_k, c) = 0$ when $|c - c_{i_q}| \geq 1.1$ for all $q \in [t(k-1)+1, t(k)]$.

This network need $O(\sqrt{N_0}\ln(N_0 nr/c))$ parameters, by (1) in lemma E.2 and lemma B.5.

**Construction of $\mathcal{F}$ and verify it:**

Let $\mathcal{F}(x) = f_3(f_2(f_1(x))) - 0.05$. We show that $\mathcal{F}$ is what we want.

(1): By parts one, two, three, and the fact $N_0 \leq N$, it is easy to see that $\mathcal{F}$ has at most $O(n + \sqrt{N}\ln(Nnr/c))$ parameters.

(2): $\mathcal{F}(x)$ is a memorization of $\mathcal{D}_{tr}$. For any $(x, y) \in \mathcal{D}_{tr}$, two cases are consided.

(1.1, if $y = 1$): using the symbols in Part two, because $y = 1$, so $x = x_{i_k}$ for some $k$. As mentioned in part two, $f_2(f_1(x))$ will output $(v_{i_{[\frac{k}{\sqrt{N_0}}]}}, f_1(x))$. Then, by part three, because $|f_1(x) - [f_1(x)]| < 1$, so we have $f_3(f_2(f_1(x))) - 0.05 \geq 0.1 - 0.05 > 0$.

(1.2 if $y = -1$): By (2) in lemma E.2, for $\forall (z, 1) \in \mathcal{D}_{tr}$, we know that $|f_1(x) - [f_1(z)]| \geq |f_1(x) - f_1(z)| - |f_1(z) - [f_1(z)]| \geq 4 - 1 = 3$. So, by part three, we have $f_3(f_2(f_1(x))) = 0 - 0.05 < 0$.

(3): $A_{\mathcal{D}}(\mathcal{F}) < 0.51$. We show that, almost all $x \in S$ ($S$ is mentioned in part one) will be given wrong label.

For $x \in S$, we have $|wx - wx_i| \geq 3$, so $|wx + b - [wx_i + b]| \geq 2$ for all $(x_i, y_i) \in \mathcal{D}_{tr}$. Then for any $v_i$, by part three and the definition of $v_i$, we have $f_3(v_i, wx + b) = 0$ when $x \in S$. So, when $f_2(f_1(x))[1] \in \{v_i\}$ and $x \in S$, we have $f_3(f_2(f_1(x))) - 0.05 = 0 - 0.05 < 0$.

Consider that for any $x \in S$, the label of $x$ is 1 in distribution $\mathcal{D}$. So when $x \in S$ satisfies $f_2(f_1(x))[1] \in \{v_i\}$, we find that $f(x)$ gives the wrong label to $x$. Since $P(x \in S) \geq 0.99/2$ and $P(f_2(f_1(x))[1] \in \{v_i\}) > 0.999$, we have $P(x \in S, f_2(f_1(x))[1] \in \{v_i\}) \geq 0.99/2 - 0.001 > 0.49$.

By the above result, we have that, with probability at least 0.49, $\text{Sgn}(f(x)) \neq y$, so $A_{\mathcal{D}}(f) < 0.51$. So, we prove the theorem. $\qquad\square$

# F  Proof of Theorem 6.1

We first give three simple lemmas.

**Lemma F.1.** *We can find $2^{[\frac{n}{\lceil c^2\rceil}]}$ points in $[0, 1]^n$, and the distance between any two points shall not be less than $c$.*

*Proof.* Let $t = [\frac{n}{\lceil c^2\rceil}]$. We just need to consider following points in $[0, 1]^n$:

For any given $i_1, i_2, i_3, \ldots, i_t \in \{0, 1\}$, let $x_{i_1, i_2, i_3, \ldots, i_t}$ be the vector in $[0, 1]^n$ satisfying: for any $j \in [t]$, the $(j-1)\lceil c^2\rceil + 1$ to $j\lceil c^2\rceil$ weights of $x_{i_1, i_2, i_3, \ldots, i_t}$ is $i_j$; other weights are 0.

We will show that, if $\{i_1, i_2, i_3, \ldots, i_t\} \neq \{j_1, j_2, j_3, \ldots, j_t\}$, then it holds $||x_{i_1, i_2, i_3, \ldots, i_t} - x_{j_1, j_2, j_3, \ldots, j_t}||_2 \geq c$. Without losing generality, let $i_1 \neq j_1$. Then the first $\lceil c^2\rceil$ weights of $x_{i_1, i_2, i_3, \ldots, i_t}$ and $x_{j_1, j_2, j_3, \ldots, j_t}$ are different: one is all 1, and the other is all 0. So, the distance between such two points is at least $\sqrt{\lceil c^2\rceil} \geq c$.

Then $\{x_{i_1, i_2, i_3, \ldots, i_t}\}_{i_j \in [0,1]}$ is the $2^t$ point we want, so we prove the lemma. $\qquad\square$

**Lemma F.2.** *If $\epsilon, \delta \in (0, 1)$ and $k, x \in \mathbb{Z}_+$ satisfy that: $x \leq k(1 - 2\epsilon - \delta)$, then $2^x(\sum_{j=0}^{[k\epsilon]} \binom{k-x}{j}) < 2^k(1 - \delta)$.*

*Proof.* We have

$$2^x(\textstyle\sum_{j=0}^{[k\epsilon]} \binom{k-x}{j}) \leq 2^x 2^{k-x} \frac{[k\epsilon]}{k-x} \leq 2^k \frac{k\epsilon}{k-x} < 2^k(1 - \delta).$$

The first inequality sign uses $\sum_{j=0}^{m} \binom{n}{m} \leq m2^n/n$ where $m \leq n/2$, and by $x \leq k(1 - 2\epsilon - \delta)$, so $[k\epsilon] \leq (k - x)/2$. The third inequality sign uses the fact $x \leq k(1 - 2\epsilon - \delta)$. $\qquad\square$

**Lemma F.3.** *If $k, v \in \mathbb{R}^+$ such that $kv > 3$, and $a = [kv]$ and $3 \leq b \leq \sqrt{k}\ln(\sqrt{k})$, then $a \geq (b/\ln(b))^2 v/2$.*

*Proof.* If $\sqrt{k} \leq b/\ln(b)$, then $b \leq \sqrt{k}\ln(\sqrt{k}) < \sqrt{k}\ln(b) \leq b$, which is impossible. So $b \leq \sqrt{k}\ln(\sqrt{k})$, and then $\sqrt{k} \geq b/\ln(b)$. Resulting in $a \geq kv - 1 \geq kv/2 \geq (b/\ln(b))^2 v/2$. $\qquad\square$

Now, we prove Theorem 6.1

*Proof.* By Theorem 4.1, we know that there is a $v_1 > 1$, when $\sqrt{N} \geq n$, for any distribution $\mathcal{D} \in \mathcal{D}(n, c)$ and $\mathcal{D}_{tr} \sim \mathcal{D}^N$, $\mathcal{D}_{tr}$ has a memorization with $v_1\sqrt{N}\ln(Nn/c)$ parameters. We will show that Theorem 6.1 is true for $v = \frac{1}{32v_1^2}$.

Assume Theorem 6.1 is wrong, then there exists a memorization algorithm $\mathcal{L}$ such that for any $n \in \mathbb{Z}_+, c, \epsilon, \delta \in (0, 1)$, if $\mathcal{D} \in \mathcal{D}(n, c)$ and $N \geq \frac{1}{32v_1^2} * \frac{N_{\mathcal{D}}^2}{\ln^2(N_{\mathcal{D}})}(1 - 2\epsilon - \delta)$, we have

$$\mathbb{P}_{\mathcal{D}_{tr} \sim \mathcal{D}^N}(A(\mathcal{L}(\mathcal{D}_{tr})) \geq 1 - \epsilon) \geq 1 - \delta.$$

We will derive contradictions based on this $\mathcal{L}$.

**Part 1: Find some points and values.**

We can find $k, n, c, \delta, \epsilon$ satisfying

(1): we have $n, k \in \mathbb{Z}_+$ and $12v_1 \leq n \leq \sqrt{k}$. Let $c = 1$, and we can find $k$ points in $[0, 1]^n$ and the distance between any pair of these points is greater than $c$;

(2): $\delta, \epsilon \in (0, 1)$ and $q = [k(1 - 2\epsilon - \delta)] \geq 3$.

By lemma F.1, to make (1) valid, we just need $n^2 < k \leq 2^n$, and (2) is easy to satisfy.

**Part 2: Construct some distribution**

Let $\{u_i\}_{i=1}^k$ satisfy $u_i \in [0, 1]^n$ and $||u_i - u_j||_2 \geq c$. By (1) mentioned in (1) in Part 1, such $\{u_i\}_{i=1}^k$ must exist. Now, we consider the following types of distribution $\mathcal{D}$:

(c1): $\mathcal{D}$ is a distribution in $\mathcal{D}(n, c)$ and $\mathbb{P}_{(x,y)\sim\mathcal{D}}(x \in \{u_i\}_{i=1}^k) = 1$.

(c2): $\mathbb{P}_{(x,y)\sim\mathcal{D}}(x = u_i) = \mathbb{P}_{(x,y)\sim\mathcal{D}}(x = u_j) = 1/k$ for any $i, j \in [k]$.

It is obvious that, by $||u_i - u_j||_2 \geq c$, such a distribution exists. Let $S$ be the set that contains all such distributions. We will show that for $\mathcal{D} \in S$, it holds $N_{\mathcal{D}} \leq v_1\sqrt{k}\ln(kn/c)$.

By Theorem 4.1 and definition of $v_1$, we know that for any distribution $\mathcal{D} \in S$, let $y_i$ be the label of $u_i$ in distribution $\mathcal{D} \in S$. Then there is a memorization $\mathcal{F}$ of $\{(u_i, y_i)\}_{i=1}^k$ with at most $v_1\sqrt{k}\ln(kn/c)$ parameters. Then by (c1), the above result implies $A_{\mathcal{D}}(\mathcal{F}(x)) = 1$, so we know that $N_{\mathcal{D}} \leq v_1\sqrt{k}\ln(kn/c)$ for any $\mathcal{D} \in S$. Moreover, by $k \geq n \geq 3$, $c = 1$ and it is easy to see that $N_{\mathcal{D}} \geq n$. We thus have $3 \leq N_{\mathcal{D}} \leq 4v_1\sqrt{k}\ln(\sqrt{k})$.

**Part 3: A definition.**

Moreover, for $\mathcal{D} \in S$, we define $S(\mathcal{D})$ as the following set:

$Z \in S(\mathcal{D})$ if and only if $Z \in [k]^q$ is a vector satisfying: Define $D(Z)$ as $D(Z) = \{(u_{z_i}, y_{z_i})\}_{i=1}^q$, then $A_{\mathcal{D}}(\mathcal{L}(D(Z))) \geq 1 - \epsilon$, where $z_i$ is the $i$-th weight of $Z$ and $y_{z_i}$ is the label of $u_{z_i}$ in distribution $\mathcal{D}$.

It is easy to see that, if we i.i.d select $q$ samples in distribution $\mathcal{D}$ to form a dataset $\mathcal{D}_{tr}$, then

(1): By $c2$, with probability 1, $\mathcal{D}_{tr}$ only contains the samples $(u_j, y_j)$ where $j \in [k]$;

(2): Let $\mathcal{D}_{tr}$ has the form shown in (1). Then every time a sample is selected, it is in $\{(u_i, y_i)\}_{i=1}^k$. Now we construct a vector in $[k]^q$ as follows: the index of $i$-th selected samples as the $i$-th component of the vector. Then each selection situation corresponds to a vector in $[k]^q$ which is constructed as before. Then by the definition of $S(\mathcal{D})$, we have $A_{\mathcal{D}}(\mathcal{L}(\mathcal{D}_{tr})) \geq 1 - \epsilon$ if and only if the corresponding vector of $\mathcal{D}_{tr}$ is in $S(\mathcal{D})$.

Putting $N_{\mathcal{D}} \leq 4v_1 \sqrt{k} \ln(\sqrt{k})$ and $q = [k(1 - 2\epsilon - \delta)]$ in lemma F.3, we have $q \geq (\frac{N_{\mathcal{D}}/(4v_1)}{\ln(N_{\mathcal{D}}/(4v_1))})^2 (1 - 2\epsilon - \delta)/2 \geq \frac{N_{\mathcal{D}}^2(1 - 2\epsilon - \delta)}{32v_1^2 \ln^2(N_{\mathcal{D}})}$.

By the above result and the by the assumption of $\mathcal{L}$ at the beginning of the proof, so that for any $\mathcal{D} \in S$ we have t

$$\mathbb{P}_{\mathcal{D}_{tr} \sim \mathcal{D}^q}(A(\mathcal{L}(\mathcal{D}_{tr})) \geq 1 - \epsilon) = \frac{|S(\mathcal{D})|}{k^q} \geq 1 - \delta. \tag{4}$$

**Part 4: Prove the Theorem.**

Let $S_s$ be a subset of $S$, and $S_s = \{\mathcal{D}_{i_1, i_2, \ldots, i_k}\}_{i_j \in \{-1, 1\}, j \in [k]} \subset S$, where distribution $\mathcal{D}_{i_1, i_2, \ldots, i_k}$ satisfies the label of $u_j$ is $i_j$, where $j \in [k]$.

We will show that there exists at least one $\mathcal{D} \subset S_s$, such that $|S(\mathcal{D})| < (1 - \delta)k^q$, which is contrary to equation 4. To prove that, we just need to prove that $\sum_{\mathcal{D} \in S_s} |S(\mathcal{D})| < (1 - \delta)2^k k^q$, use $|S_s| = 2^k$ here.

To prove that, for any vector $Z \in [k]^q$, we estimate how many $\mathcal{D} \in S_s$ which makes $Z$ to be included in $S(\mathcal{D})$.

**Part 4.1, situation of a given vector $Z$ and a given distribution $\mathcal{D}$.**

For a $Z = (z_i)_{i=1}^q$ and $\mathcal{D}$ such that $Z \in S(\mathcal{D})$, let $\text{len}(Z) = \{c \in [k] : \exists i, c = z_i\}$. We consider the distributions in $S_s$ that satisfy the following condition: for $i \in \text{len}(Z)$, the label of $u_i$ is equal to the label of $u_i$ in $\mathcal{D}$.

Obviously, we have $2^{k - |\text{len}(Z)|}$ distributions that can satisfy the above condition in $S_s$. Let such distributions make up a set $S_{ss}(\mathcal{D}, Z)$. Now, we estimate how many distributions $\mathcal{D}_s$ in $S_{ss}(\mathcal{D}, Z)$ satisfy $Z \in S(\mathcal{D}_s)$.

For any distribution $G \in S_s$, let $y(G)_i$ be the label of $u_i$ in distribution $G$, and define the dataset $\mathcal{D}_{tr} = \{(u_{z_i}, y(\mathcal{D})_{z_i})\}_{i=1}^q$. Then $Z \in S(\mathcal{D}_s)$ if and only if: for at least $k - [k\epsilon]$ of $i \in [k]$, $\mathcal{L}(\mathcal{D}_{tr})$ gives the label $y(\mathcal{D}_s)_i$ to $u_i$.

Firstly, consider that when $i \in \text{len}(Z)$. For any $\mathcal{D}_s \in S_{ss}(\mathcal{D}, Z)$, we have $y(\mathcal{D}_s)_i = y(\mathcal{D})_i$ and $\mathcal{L}(\mathcal{D}_{tr})$ must give the label $y(\mathcal{D})_i$ to $u_i$, so when $i \in \text{len}(Z)$, $\mathcal{L}(\mathcal{D}_{tr})$ gives the label $y(\mathcal{D}_s)_i$ to $u_i$.

Then, consider $i \notin \text{len}(Z)$. Because $Z$ is a given vector, so if $Z \in S(\mathcal{D}_s)$, the label $y(\mathcal{D}_s)_i$ where $i \notin \text{len}(Z)$ are at most $[k\epsilon]$ different from the label of $u_i$ given by $\mathcal{L}(\mathcal{D}_{tr})$.

So, by the above two results, this kind of $\mathcal{D}_s$ is at most $\sum_{i=0}^{[k\epsilon]} \binom{k - |\text{len}(Z)|}{i}$. So, we have $\sum_{i=0}^{[k\epsilon]} \binom{k - |\text{len}(Z)|}{i}$ number of distributions $\mathcal{D}_s$ in $S_{ss}(\mathcal{D}, Z)$ satisfy $Z \in S(\mathcal{D}_s)$.

**Part 4.2, for any vector $Z$ and distribution $\mathcal{D}$.**

Firstly, for a given $Z$, we have at most $2^{|len(Z)|}$ different $S_{ss}(\mathcal{D}, Z)$ for $\mathcal{D} \in \mathcal{D}_S$.

Because when $\mathcal{D}_1$ and $\mathcal{D}_2$ satisfy $y(\mathcal{D}_1)_i = y(\mathcal{D}_2)_i$ for any $i \in \text{len}(Z)$, we have $\mathcal{D}_{ss}(\mathcal{D}_1, Z) = \mathcal{D}_{ss}(\mathcal{D}_2, Z)$, and $2^{|\text{len}(Z)|}$ situations of label of $u_i$ where $i \in \text{len}(Z)$, so there exist at most $2^{|\text{len}(Z)|}$ different $\mathcal{S}_{ss}(\mathcal{D}, Z)$.

By part 4.1, for a $\mathcal{S}_{ss}(\mathcal{D}, Z)$, at most $\sum_{i=0}^{[k\epsilon]} \binom{k-|\text{len}(Z)|}{i}$ of $\mathcal{D}_s \in S_{ss}(\mathcal{D}, Z)$ satsify $Z \in S(\mathcal{D}_s)$. So by the above result and consider that $\mathcal{D}_s = \cup_{\mathcal{D} \in \mathcal{D}_s} \mathcal{S}_{ss}(\mathcal{D}, Z)$, at most $2^{|\text{len}(Z)|} \sum_{i=0}^{[k\epsilon]} \binom{k-|\text{len}(Z)|}{i}$ number of $\mathcal{D}_s \in S_s$ such that $Z \in S(\mathcal{D}_s)$.

And there exist $k^q$ different $Z$, so $\sum_{\mathcal{D} \in S_s} |S(\mathcal{D})| \leq \sum_Z 2^{|\text{len}(Z)|} \sum_{i=0}^{[k\epsilon]} \binom{k-|\text{len}(Z)|}{i} \leq \sum_Z 2^k (1 - \delta) = k^q 2^k (1 - \delta)$. For the last inequality, we use $2^{|\text{len}(Z)|} \sum_{i=0}^{[k\epsilon]} \binom{k-|\text{len}(Z)|}{i} < 2^k (1 - \delta)$, which can be shown by $|\text{len}(Z)| \leq q$ and lemma F.2.

This is what we want. we proved the theorem. $\qquad \square$

We now prove Corollary 6.4.

*Proof.* Using lemma F.1, we can find $2^{\left[\frac{n}{\lceil c^2 \rceil}\right]}$ points in $[0, 1]^n$ and the distance between any two points shall not be less than $c$. So we take a $\epsilon, \delta$ such that $1 - 2\epsilon - \delta > 0$, $n = 3[12v_1/(1 - 2\epsilon - \delta)] + 3$, $c = 1$ and $k = 2^{\left[\frac{n}{\lceil c^2 \rceil}\right]}$ in the (1) in the part 1 of the proof of Theorem 6.1, then similar as the proof of Theorem 6.1, and we get this corollary. $\qquad \square$

# G  Proof of Theorem 6.5

## G.1  The Existence

Firstly, it is easy to show that there exists a memorization algorithm which satisfies $\mathcal{L}(\mathcal{D}_{tr}) \leq N_{\mathcal{D}}$ when $\mathcal{D}_{tr} \sim \mathcal{D}^N$ with probability 1. We just consider the following memorization algorithm:

For a given dataset $D$, let $\mathcal{L}(D)$ be the memorization of $\mathcal{D}$ with minimum parameters, as shown in Theorem 4.1. Then $\text{para}(\mathcal{L}(D)) \leq \overline{O}(\sqrt{|D|})$.

And if $D$ is i.i.d selected from distribution $\mathcal{D}$, where $\mathcal{D} \in \mathcal{D}(n, c)$, then by the definition of $\mathcal{L}$ and $N_{\mathcal{D}}$ in Theorem 4.3, we have $\text{para}(\mathcal{L}(D)) \leq N_{\mathcal{D}}$ with probability 1. So $\mathcal{L}$ is what we want.

## G.2  The Sample Complexity of Generalization

To prove (1) in the theorem, we need three lemmas.

**Lemma G.1** ([44]). *Let $H$ be a hypothesis space with VCdim $h$ and $\mathcal{D}$ is distribution of data, if $N \geq h$, then with probability $1 - \delta$ of $\mathcal{D}_{tr} \sim \mathcal{D}^N$, we have*

$$|E_{\mathcal{D}}(\mathcal{F}) - E_{\mathcal{D}_{tr}}(\mathcal{F})| \leq \sqrt{\frac{8h \ln \frac{2eN}{h} + 8 \ln \frac{4}{\delta}}{N}}$$

*for any $\mathcal{F} \in H$. Here, $E_{\mathcal{D}}(\mathcal{F}) = E_{(x,y) \sim \mathcal{D}}[I(\mathcal{F}(x) = y)]$, $E_{\mathcal{D}_{tr}}(\mathcal{F}) = \sum_{(x,y \in \mathcal{D}_{tr})}[I(\mathcal{F}(x) = y)]$ and $I(x) = 1$ if x is true or $I(x) = 0$.*

*Moreover, when $h \geq 1$, we have*

$$|E_{\mathcal{D}}(\mathcal{F}) - E_{\mathcal{D}_{tr}}(\mathcal{F})| \leq \sqrt{\frac{8h \ln \frac{8eN}{\delta h}}{N}}.$$

**Lemma G.2.** *If $e \leq ba/c$, then we have $a \ln(bu) \leq cu$ when $u \geq 2a \ln(ba/c)/c$.*

*Proof.* Firstly, we have $\frac{a \ln(bu)}{cu} = \frac{\ln(ba/c(cu/a))}{cu/a}$, and we just need to show $\frac{\ln(ba/c(cu/a))}{cu/a} \leq 1$.

Then, we show that there are $2 \ln(ba/c) \leq ba/c$. Just consider the function $g(x) = x - 2 \ln x$, by $g'(x) = 1 - 2/x$, so $g'(x) \geq 0$ when $x \geq 2$, so $g(ba/c) \geq g(e) = e - 2 > 0$, this is what we want.

Now we consider the function $f(x) = \ln((ba/c)x)/x$, by the above result, we have that $1 \leq 2 \ln(ba/c) \leq ba/c$, we have that

$$
\begin{aligned}
&f(2 \ln(ba/c)) \\
=\ &\ln(2(ba/c) \ln(ba/c))/(2 \ln(ba/c)) \\
\leq\ &\ln((ba/c) * (ba/c))/(2 \ln(ba/c)) \\
=\ &1.
\end{aligned}
$$

And consider that $f'(x) = \frac{1 - \ln((ba/c)x)}{x^2} \leq 0$ when $x \geq 1$, so, when $x \geq 2\ln(ba/c)$, we have $f(x) \leq f(2\ln(ba/c)) \leq 1$, which means that when $cu/a \geq 2\ln(ba/c)$, it holds $\frac{\ln(ba/c(cu/a))}{cu/a} \leq 1$. The lemma is proved. $\qquad \square$

**Lemma G.3** ([6]). *Let $H_m$ be the hypothesis space composed of the networks with at most $m$ parameters. Then the VCdim of $H_m$ is not more than $qm^2 \ln(m)$, where $q$ is a constant not dependent on $m$.*

Then we can prove (1) in the theorem.

*Proof.* Let $\mathcal{D}_{tr} \sim \mathcal{D}^N$. Because the algorithm satisfies the condition in theorem, then $\mathcal{L}(\mathcal{D}_{tr}) \in H_{N_\mathcal{D}}$, where $H_{N_\mathcal{D}}$ is defined in lemma G.3. By lemma G.3, the VCdim of $H_{\mathcal{D}_{tr}}$ is not more than $qN_\mathcal{D}^2 \ln(N_\mathcal{D})$ for some $q \geq 1$. Using lemma G.1 to this fact, we have $N \geq \frac{16qN_\mathcal{D}^2 \ln(N_\mathcal{D}) \ln(\frac{64qeN_\mathcal{D}^2 \ln(N_\mathcal{D})}{\delta \epsilon^2})}{\epsilon^2}$. Take these values in lemma G.1, and considering that the memorization algorithm $\mathcal{L}$ must satisfy that $E_{\mathcal{D}_{tr}}(\mathcal{L}(\mathcal{D}_{tr})) = 1$, using lemma G.2 (just take $a = 8qN_\mathcal{D}^2 \ln(N_\mathcal{D})$, $b = 8e/\delta$ and $c = \epsilon^2$ in lemma G.2), we have

$$1 - E_\mathcal{D}(\mathcal{L}(\mathcal{D}_{tr})) \leq \sqrt{\frac{8qN_\mathcal{D}^2 \ln(N_\mathcal{D}) \ln \frac{8eN}{\delta}}{N}} \leq \epsilon$$

which implies $1 - \epsilon \leq E_\mathcal{D}(\mathcal{L}(\mathcal{D}_{tr}))$. The theorem is proved. $\qquad \square$

### G.3 More Lemmas

We need three more lemmas to prove Theorem 6.5.

**Lemma G.4.** *Let $\mathcal{D} \subset [0,1]^n \times \{-1, 1\}$. Then $D$ has a memorization with width 1 if and only if $\mathcal{D}$ is linearly separable.*

*Proof.* If $D$ is linearly separable, then it obviously has a memorization with width 1.

If $D$ has a memorization with width 1, we show that $D$ is linearly separable. Let $\mathcal{F}$ be the memorization network of $\mathcal{D}$ with width 1, and $\mathcal{F}_1$ the first layer of $\mathcal{F}$.

**Part 1:** We show that it is impossible to find any $(x_1, 1), (x_2, -1), (x_3, 1) \in D$ such that $\mathcal{F}_1(x_1) < \mathcal{F}_1(x_2) < \mathcal{F}_1(x_3)$. If we can, then contradiction will be obtained.

Assume $(x_1, 1), (x_2, -1), (x_3, 1) \in D$ such that $\mathcal{F}_1(x_1) < \mathcal{F}_1(x_2) < \mathcal{F}_1(x_3)$.

It is easy to see that, for any linear function $wx + b$ and $u \leq v \leq k$, we have $wu + b \leq wv + b \leq wk + b$ or $wu + b \geq wv + b \geq wk + b$, which implies $\text{Relu}(wu + b) \leq \text{Relu}(wv + b) \leq \text{Relu}(wk + b)$ or $\text{Relu}(wu + b) \geq \text{Relu}(wv + b) \geq \text{Relu}(wk + b)$.

Because $(x_1, 1), (x_2, -1), (x_3, 1) \in D$ satisfy that $\mathcal{F}_1(x_1) < \mathcal{F}_1(x_2) < \mathcal{F}_1(x_3)$, and each layer of $\mathcal{F}$ is a linear function composite Relu, so after each layer, the order of $\mathcal{F}_1(x_1), \mathcal{F}_1(x_2), \mathcal{F}_1(x_3)$ is not changed or contrary. So there must be $\mathcal{F}(x_1) \leq \mathcal{F}(x_2) \leq \mathcal{F}(x_3)$ or $\mathcal{F}(x_1) \geq \mathcal{F}(x_2) \geq \mathcal{F}(x_3)$. Then $\mathcal{F}$ cannot classify $x_1, x_2, x_3$ correctly, which contradicts to the fact that $\mathcal{F}$ is a memorization of $D$.

**Part 2:** We show that, it is impossible to find any $(x_1, -1), (x_2, 1), (x_3, -1) \in D$ such that $\mathcal{F}_1(x_1) < \mathcal{F}_1(x_2) < \mathcal{F}_1(x_3)$.

This is similar to Part 1.

By parts 1 and 2, without losing generalization, we know that for any $(x_1, 1), (x_2, -1) \in D$, it holds $\mathcal{F}_1(x_1) > \mathcal{F}_1(x_2)$. Since $\mathcal{F}_1$ is a linear function composite Relu, $D$ is linear separable. $\qquad \square$

**Lemma G.5.** *Let $D = \{(x_i, y_i)\} \subset [0, 1] \times \{-1, 1\}$. Then $D$ has a memorization with width 2 and depth 2 if and only if at least one of the following conditions is valid.*

*(c1): There is a closed interval $I$ such that: if $(x, 1) \in D$ then $x \in I$ and if $(x, -1) \in D$ then $x \notin I$.*

*(c2): There is a closed interval $I$ such that: if $(x, 1) \in D$ then $x \notin I$ and if $(x, -1) \in D$ then $x \in I$.*

*Proof.* **Part 1:** We show that if condition (c1) is valid, then $D$ has a memorization with width 2 and depth 2. It is similar for (c2) to be valid.

Let $I = [a, b]$. If for all $(x, -1) \in D$, we have $x < a$, then $D$ is linear separable, and the result is valid. If for all $(x, -1) \in D$, we have $x > b$, then $D$ is linear separable, and the result is valid. Now we consider the situation where $x > a$ for some $(x, -1) \in D$ and $x < b$ for some $(x, -1) \in D$.

Let $x_{-1} = max_{(x,-1)\in D}\{x < a\}$. Then for $\mathcal{F}_1(x) = x - (x_{-1} + a)/2$, it is easy to verify that $F_1(x) > \frac{a - x_{-1}}{2}$ for all $x \geq a$ and $F_1(x) < 0$ for all $(x_0, -1) \in D$ such that $x_0 < a$.

Let $x_1 = \min_{(x,-1)\in D}\{x > b\}$. Then for $\mathcal{F}_2(x) = x - (x_1 + b)/2$, it is easy to verify that $F_2(x) < 0$ for all $x \leq b$ and $F_2(x) > (x_1 - b)/2$ for all $(x_0, -1) \in D$ such that $x_0 > b$.

Let the network $F$ be defined by $F = \mathrm{Relu}(F_1(x)) - T\mathrm{Relu}(F_2(x)) - t$, where $T = \frac{8}{x_1 - b}$ is a positive real number, and $t = \frac{a - x_{-1}}{4} > 0$.

Now we prove $F$ is what we want. It is easy to see that, $F$ is a depth 2 width 2 network. When $x \in [a, b]$, then $\mathcal{F}_1(x) > \frac{a - x_{-1}}{2}$ and $F_2(x) \leq 0$, so $F(x) > 0$. For $(x, -1) \in D$ such that $x < a$, we have $\mathcal{F}_1(x) < 0$ and $F_2(x) < 0$, so $F(x) < 0$; for $(x, -1) \in D$ such that $x > b$, we have $\mathcal{F}_1(x) < 1$ and $F_2(x) > \frac{x_1 - b}{4}$, so $F(x) \leq 1 - 2 - \frac{a - x_{-1}}{4} < 0$, this is what we want.

**Part 2:** If $D$ has a memorization with width 2 and depth 2, then we show that $D$ satisfies conditions (c1) or (c2).

If $D$ is linear separable, (c1) and (c2) are valid. If not, without losing generality, assume that $(x_1, 1), (x_2, -1), (x_3, 1) \in D$ such that $x_1 < x_2 < x_3$ (for the situation that $(x_1, -1), (x_2, 1), (x_3, -1) \in D$ such that $x_1 < x_2 < x_3$, the proof is similar). Then we show that if $(x, -1) \in D$, we have $x_1 < x < x_3$. Assume $(x_0, -1) \in D$ such that $x_0 < x_1$, then we have that $x_0 < x_1 < x_2 < x_3$, then we can deduce the contradiction.

Let $F = a\mathrm{Relu}(F_1(x)) + b\mathrm{Relu}(F_2(x)) + c$ be the memorization network of $D$, where $F_i(x)$ is a linear function. Let $u, v \in \mathbb{R}$ such that $F_1(u) = F_2(v) = 0$, without loss generality, let $u \leq v$.

Then we know that $F$ is linear in such three regions: $(-\infty, u]$, $[u, v]$ and $[v, \infty)$. We call the three regions as linear regions of $F$. We prove the following three results at first.

(1): The slope of $F$ on $(-\infty, u]$ is positive.

Firstly, we show that $x_0 \in (-\infty, u]$. If not, since $(x_0, -1), (x_1, 1), (x_2, -1)$ are not linear separable, and $(x_1, 1), (x_2, -1), (x_3, 1)$ are not linear separable, we have $(x_0, -1), (x_1, 1) \in [u, v]$ and $(x_2, -1), (x_3, 1) \in [v, \infty)$. Then, because $x_1 > x_0$ and $F(x_1) > F(x_0)$, and $F$ is linear in $[u, v]$, we have that $F(v) \geq F(x_1) > 0$. Now we consider the points $(v, 1), (x_2, -1), (x_3, 1)$. It is easy to see that $F$ memorizes such three points and they are in the linear region of $F$, so $(v, 1), (x_2, -1), (x_3, 1)$ is linear separable, which is impossible because $v \leq x_2 \leq x_3$ and resulting in contradiction, so $x_0 \in (-\infty, u]$.

If the slope of $F$ on $(-\infty, u]$ is not positive, since $u \geq x_0$, we have $F(u) < 0$. Now we consider the points $(u, -1), (x_1, 1), (x_2, -1), (x_3, 1)$. Just similar to above to get the contradiction. So the slope of $F$ on $(-\infty, u]$ is positive.

(2): The slope of $F$ on $[v, \infty)$ is positive. Similar to (1).

(3): The slope of $F$ on $(-\infty, u]$ is negative. If not, $F$ must be a non-decreasing function, which is impossible.

Using (1),(2),(3), we can get a contradiction, which means that there is a $(x_0, -1) \in D$ such that $x_0 < x_1$ is not possible.

Consider that, in a linear region of $F$, if the activation states of $F_1$ and $F_2$ are both not activated, then on such linear regions, the slope of $F$ is 0. But due to (1),(2),(3), all linear regions have non-zeros slope of $F$, so on each linear regions, at least one of $F_1$ and $F_2$ is activated. So, the activation states of $F_1$ and $F_2$ at $(-\infty, u]$, $[u, v]$ and $[v, \infty)$ is $(-, +), (+, +), (+, -)$ (+ means activated, - means not activated).

Then the slope of $F$ on $[u, v]$ is equal to the sum of the slopes of $F$ on $(-\infty, u]$ and the slope of $F$ on $[v, \infty)$. But by (1),(2),(3), that means a negative number is equal to the sum of two positive numbers, which is impossible. So we get a contradiction.

So if $(x_0, -1) \in D$, we have $x_0 > x_1$. Similar to before, we have $x_0 < x_3$. So we get the result.

By the above result, all the samples $(x_0, -1) \in D$ satisfies $x \in (x_1, x_3)$, so there is a close interval in $(x_1, x_3)$ such that: if $(x_0, -1) \in D$, then $x_0$ is in such interval, then (c2) is vald, and we prove the lemma. □

### G.4 The algorithm is no-efficient.

Now we prove (2) of theorem 6.5, that is, all such algorithm is not efficient if $P \neq NP$. We need the reversible 6-SAT problem defined in definition 6.6.

*Proof.* We will show that, if there is an efficient memorization algorithm which satisfies the conditions of the theorem (has at most $N_\mathcal{D}$ parameters with probability 1), then we can solve the reversible 6-SAT in polynomial time, which implies $P = NP$.

Firstly, for the 6-SAT problem, we write it as the following form.

Let $\varphi = \wedge_{i=1}^m \varphi_i(n, m)$ be a 6-SAT for $n$ variables, where $\varphi_i(n, m) = \vee_{j=1}^6 \tilde{x}_{i,j}$ and $\tilde{x}_{i,j}$ is either $x_s$ or $\neg x_s$ for some $s \in [n]$ (see Definition 6.6). Then, we define some vectors in $\mathbb{R}^n$ based on $\varphi_i(n, m)$.

For $i \in [m]$, define $Q_i^\varphi \in \mathbb{R}^n$ as follows: $Q_i^\varphi[j] = 1$ if $x_j$ occurs in $\varphi_i(n, m)$; $Q_i^\varphi[j] = -1$ if $\neg x_j$ occurs in $\varphi_i(n, m)$; $Q_i^\varphi[j] = 0$ otherwise. $Q_i^\varphi[j]$ is the $j$-th entry in $Q_i^\varphi$, then six entries in $Q_i^\varphi$ are 1 or $-1$ and all other entries are zero.

Now, we define a binary classification dataset $\mathcal{D}(\varphi) = \{(x_i, y_i)\}_{i=1}^{m+4n} \subset [0, 1]^n \times [2]$ as follows.

(1) For $i \in [n]$, $x_i = \mathbf{1}_i/3 + 1.1\mathbf{1}/3$, $y_i = 1$.

(2) For $i \in \{n+1, n+2, \ldots, 2n\}$, $x_i = 1.1\mathbf{1}_{i-n}/3 + 1.1\mathbf{1}/3$, $y_i = -1$.

(3) For $i \in \{2n+1, 2n+2, \ldots, 3n\}$, $x_i = -\mathbf{1}_{i-2n}/3 + 1.1\mathbf{1}/3$, $y_i = 1$.

(4) For $i \in \{3n+1, 3n+2, \ldots, 4n\}$, $x_i = -1.1\mathbf{1}_{i-3n}/3 + 1.1\mathbf{1}/3$, $y_i = -1$.

(5) For $i \in \{4n+1, 4n+2, \ldots, 4n+m\}$, $x_i = 1/12 Q_{i-4n}^\varphi + 1.1\mathbf{1}/3$, $y_i = 1$.

Here, $\mathbf{1}_i$ is the vector whose $i$-th weight is 1 and other weights are 0, $\mathbf{1}$ is the vector whose weights are all 1.

Let $\mathcal{L}$ be an efficient memorization algorithm which satisfies the condition in the theorem. Then we prove the following result: If $n \geq 4$ and $\varphi$ is a reversible 6-SAT problem, then $\text{para}(\mathcal{L}(\mathcal{D}(\varphi))) = n+8$ if and only if $\varphi$ has a solution, which means $P = NP$ and leads to that $\mathcal{L}$ does not exist when $P \neq NP$. The proof is divided into two parts.

**Part 1:** If $\varphi$ is a reversible 6-SAT problem that has a solution, then $\text{para}(\mathcal{L}(\mathcal{D}(\varphi))) = n+8$.

To prove this part, we only need to prove that $\text{para}(\mathcal{L}(\mathcal{D}(\varphi))) \geq n+8$ and $\text{para}(\mathcal{L}(\mathcal{D}(\varphi))) \leq n+8$.

**Part 1.1:** we have $\text{para}(\mathcal{L}(\mathcal{D}(\varphi))) \geq n+8$.

Firstly, we show that $\{(x_1, 1), (x_{n+1}, -1), (x_{2n+1}, 1), (x_{3n+1}, -1)\} \subset \mathcal{D}(\varphi)$ are not linearly separable. This is because $\{x_1, x_{n+1}, x_{2n+1}, x_{3n+1}\}$ is a linear transformation of $\{\mathbf{1}_1, 1.1\mathbf{1}_1, -\mathbf{1}_1, -1.1\mathbf{1}_1\}$, so $\{(x_1, 1), (x_{n+1}, -1), (x_{2n+1}, 1), (x_{3n+1}, -1)\} \subset \mathcal{D}(\varphi)$ are not linearly separable if and only if $\{(\mathbf{1}_1, 1), (1.1\mathbf{1}_1, -1), (-\mathbf{1}_1, 1), (-1.1\mathbf{1}_1, -1)\}$ are not linearly separable, by the definition of $\mathbf{1}_1$, easy to see that $\{(\mathbf{1}_1, 1), (1.1\mathbf{1}_1, -1), (-\mathbf{1}_1, 1), (-1.1\mathbf{1}_1, -1)\}$ are not linearly separable, so we get the result.

By the above result, a subset of $\mathcal{D}(\varphi)$ is not linearly separable, so we have that $\mathcal{D}(\varphi)$ is not linearly separable. So, by lemma G.4, $\mathcal{L}(\mathcal{D}(\varphi))$ must have width more than 1. For a network with width at least 2, when it has depth 2, it has at least $2n+5$ parameters; when it has depth 3, it has at least $n+8$ parameters; when it has depth more than 3, it has at least $n+10$ parameters. So when $n \geq 4$, we have $\text{para}(\mathcal{L}(\mathcal{D}(\varphi))) \geq n+8$.

**Part 1.2:** If $\varphi$ is a reversible 6-SAT problem that has a solution, then $\text{para}(\mathcal{L}(\mathcal{D}(\varphi))) \leq n+8$.

We define a distribution $\mathcal{D}$ at first. $\mathcal{D}$ is defined on $\mathcal{D}(\varphi)$, and each point has the same probability. It is easy to see that, $\mathcal{D} \in \mathcal{D}(n, 1/30)$.

Since when $N \geq m + 4n$, we have $P_{\mathcal{D}_{tr} \sim \mathcal{D}^N}(\mathcal{D}_{tr} = \mathcal{D}(\varphi)) > 0$, so by the definition of $N_\mathcal{D}$ and the fact $\mathcal{L}$ satisfies the conditions in the theorem, we have $\text{para}(\mathcal{L}(\mathcal{D}(\varphi))) \leq N_\mathcal{D}$. Moreover, because $\mathcal{D}$ is defined on $\mathcal{D}(\varphi)$, we will construct a network with $n + 8$ parameters to memorize $\mathcal{D}(\varphi)$ to show that $N_\mathcal{D} \leq n + 8$, which implies $\text{para}(\mathcal{L}(\mathcal{D}(\varphi))) \leq N_\mathcal{D} \leq n + 8$ because $\mathcal{L}$ satisfies the condition in the theorem.

This network has three layers, the first layer has width 1; the second layer has width 2; the third output layer has width 1.

Let $s = (s_1, s_2, \ldots, s_n) \in \{-1, 1\}^n$ be a solution of $\varphi$. Then the first layer is $\mathcal{F}_1(x) = \text{Relu}(3s(x - 1.1\mathbf{1}/3) + 3)$. Then we have the following results:

(1): $\mathcal{F}_1(x) = 4.1$ or $\mathcal{F}_1(x) = 1.9$ for all $(x, -1) \in \mathcal{D}(\varphi)$;

(2): $2 \leq |\mathcal{F}_1(x)| \leq 4$ for all $(x, 1) \in \mathcal{D}(\varphi)$.

(1) is very easy to validate. We just prove (2).

For $i \in [n]$ and $i \in \{2n + 1, \ldots, 3n\}$, because $s \in \{-1, 1\}^n$, so $3s(x - 1.1\mathbf{1}/3) = 1$ or $3s(x - 1.1\mathbf{1}/3) = -1$, which implies $2 \leq |\mathcal{F}_1(x_i)| \leq 4$.

For $i \in \{4n + 1, \ldots, 4n + m\}$, $x_i - 1.1\mathbf{1}/3$ has only six components that are not 0. Because $s$ is the solution of $\varphi$, which indicates that at least one of the six non-zero components of $x_i - 1.1\mathbf{1}/3$ has the same positive or negative shape as the corresponding component of $s$. Consider that such six non-zero components of $x_i - 1.1\mathbf{1}/3$ are in $\{-1/12, 1/12\}$, so $3s(x_i - 1.1\mathbf{1}/3) \geq 1/4 - 5/4 = -1$.

Moreover, because $\varphi$ is a reversible problem, so $\varphi_i(n, m)$ and $\overline{\varphi_i(n, m)}$ are both in the $\varphi$, which indicates that the positive and negative forms of the six non-zero components of $x_i - 1.1\mathbf{1}/3$ cannot be exactly the same as the positive and negative forms of the corresponding components in $s$, or there must be $\overline{\varphi_i(n, m)} = 0$, which contradicts to $s$ is the solution of $\varphi$. So, we have $3s(x_i - 1.1\mathbf{1}/3) \leq 5/4 - 1/4 = 1$.

Then we have that, for $i \in \{4n + 1, \ldots, 4n + m\}$, it holds $3s(x_i - 1.1\mathbf{1}/3) \in [-1, 1]$, resulting in $2 \leq |\mathcal{F}_1(x_i)| \leq 4$. We proved (2).

By (1) and (2), and using lemma G.5, there is a network $\mathcal{F}_2 : \mathbb{R} \to \mathbb{R}$ with width 2, depth 2 and 7 parameters that can classify the $\{(\mathcal{F}_1(x_i), y_i)\}_{i=1}^{4n+m}$, so $\mathcal{F}_2 \circ \mathcal{F}_1$ is the network we want.

By such a network, we have that $N_\mathcal{D} \leq n + 8$, and then, we have $\text{para}(\mathcal{L}(\mathcal{D}(\varphi))) \leq N_\mathcal{D} \leq n + 8$. We proved the result.

**Part Two:** If $\varphi$ is a reversible 6-SAT problem and $\text{para}(\mathcal{L}(\mathcal{D}(\varphi))) = n + 8$, then $\varphi$ has a solution.

If $\mathcal{L}(\mathcal{D}(\varphi))$ has width 2 of the first layer, then $\text{para}(\mathcal{L}(\mathcal{D}(\varphi))) \geq 2n + 5 > n + 8$, so when $\text{para}(\mathcal{L}(\mathcal{D}(\varphi))) = n + 8$, the first layer has width 1.

Write $\mathcal{L}(\mathcal{D}(\varphi)) = \mathcal{F}_2(\mathcal{F}_1(x))$, and write $\mathcal{F}_1$ as $\mathcal{F}_1(x) = \text{Relu}(3s(x - 1.1\mathbf{1}/3) + b)$, and let $s = (s_1, s_2, \ldots, s_n)$.

We will prove that $\text{Sgn}(s) = (\text{Sgn}(s_1), \text{Sgn}(s_2), \text{Sgn}(s_3), \ldots, \text{Sgn}(s_n))$ is a solution of $\varphi$. The proof is given in two parts.

**Part 2.1** we have $1.1|s_i| \geq |s_j|$ for any $i, j \in [n]$. Firstly, we have $s_i \neq 0$ for any $i \in [n]$. Because if $s_i = 0$, it holds $\mathcal{F}_1(x_i) = \mathcal{F}_1(x_{n+i})$, which implies that $\mathcal{L}(\mathcal{D}(\varphi))$ gives the same label to $x_i$ and $x_{n+i}$, but $x_i$ and $x_{n+i}$ have the different labels in dataset $\mathcal{D}(\varphi)$, so it contradicts $\mathcal{L}(\mathcal{D}(\varphi))$ is the memorization of $\mathcal{D}(\varphi)$.

Without losing generality, let $|s_1| \geq |s_2| \geq \cdots \geq |s_n|$. Then we just need to prove that $1.1|s_n| \geq |s_1|$.

Because $\mathcal{D}(\varphi)$ is not linear separable, so by lemma G.4, $\mathcal{L}(\mathcal{D}(\varphi))$ has width more than 1. Because $\mathcal{F}_1$ has width 1, so $\mathcal{F}_2$ has width 2 and 7 parameters, resulting in that $\mathcal{F}_2$ is a network with width 2 and depth 2. And $\mathcal{F}_2$ can classify such six points: $\{(\mathcal{F}_1(x_i), y_i)\}_{i \in \{1, n+1, 2n+1, 3n+1, 2n, 4n\}}$.

If $s_1 > 0$, taking the values of $x_1, x_{n+1}, x_{2n+1}, x_{3n+1}$ in $\mathcal{F}_1$, we have $1.1s_1 + b = \mathcal{F}_1(x_{n+1}) \geq s_1 + b = \mathcal{F}_1(x_1) \geq -s_1 + b = \mathcal{F}_1(x_{2n+1}) \geq -1.1s_1 + b = \mathcal{F}_1(x_{3n+1})$, which implies $\mathcal{F}_1(x_{n+1}) \geq$

$\mathcal{F}_1(x_1) \geq \mathcal{F}_1(x_{2n+1}) \geq \mathcal{F}_1(x_{3n+1})$; if $s_1 < 0$. Similarly as before, we have $\mathcal{F}_1(x_{n+1}) \leq \mathcal{F}_1(x_1) \leq \mathcal{F}_1(x_{2n+1}) \leq \mathcal{F}_1(x_{3n+1})$. So, $\mathcal{F}_1(x_1)$ and $\mathcal{F}_1(x_{2n+1})$ are always in the interval from $\mathcal{F}_1(x_{n+1})$ to $\mathcal{F}_1(x_{3n+1})$.

Consider that $x_{n+1}$ and $x_{3n+1}$ have label $-1$, $x_1$ and $x_{2n+1}$ have label 1, so by Lemma G.5, if $\{(\mathcal{F}_1(x_i), y_i)\}_{i \in \{1, n+1, 2n+1, 3n+1, 2n, 4n\}}$ can be memorized by a depth 2 width 2 network, then $\mathcal{F}_1(x_{2n})$ and $\mathcal{F}_1(x_{4n})$ must be not in the interval from $\mathcal{F}_1(x_1)$ to $\mathcal{F}_1(x_{2n+1})$, or we cannot find a interval satisfies the conditions of lemma G.5.

Since $\max\{\mathcal{F}_1(x_{2n}), \mathcal{F}_1(x_{4n})\} = 1.1|s_n| + b$, to ensure that $\mathcal{F}_1(x_{2n})$ and $\mathcal{F}_1(x_{4n})$ are not in the interval from $\mathcal{F}_1(x_1)$ to $\mathcal{F}_1(x_{2n+1})$, we have $\max\{\mathcal{F}_1(x_{2n}), \mathcal{F}_1(x_{4n})\} = 1.1|s_n| + b \geq \max\{\mathcal{F}_1(x_1), \mathcal{F}_1(x_{2n+1})\} = |s_1| + b$ or $\max\{\mathcal{F}_1(x_{2n}), \mathcal{F}_1(x_{4n})\} = 1.1|s_n| + b \leq \min\{\mathcal{F}_1(x_1), \mathcal{F}_1(x_{2n+1})\} = -|s_1| + b$. The second case is impossible, so we have $1.1|s_n| \geq |s_1|$. This is what we want in this part.

**Part 2.2** We show that $\mathrm{Sgn}(s)$ is the solution of $\varphi$. Assume that $\mathrm{Sgn}(s)$ is not the solution of $\varphi$. There is a $i \in \{4n+1, \ldots, 4n+m\}$, such that the positive and negative forms of the six non-zero components of $x_i$ are exactly the same as the positive and negative forms of the corresponding components in $s$. Then $sx_i + b \geq 6/4|s_n| + b \geq 6/4.4|s_1| + b \geq 1.1|s_1| + b$. So, by $\max\{\mathcal{F}_1(x_{1+n}), \mathcal{F}_1(x_{3n+1})\} = 1.1|s_1| + b$ and $\min\{\mathcal{F}_1(x_{1+n}), \mathcal{F}_1(x_{3n+1})\} = -1.1|s_1| + b$, we know that $\mathcal{F}_1(x_i)$ is not in the interval from $\mathcal{F}_1(x_{1+n})$ to $\mathcal{F}_1(x_{3n+1})$.

Then similar to part 2.1, consider the point $\{(\mathcal{F}_1(x_i), y_i)\}_{i \in \{1, n+1, 2n+1, 3n+1, i\}}$, we have that $\mathcal{F}_1(x_1)$ and $\mathcal{F}_1(x_{2n+1})$ are always in the interval from $\mathcal{F}_1(x_{n+1})$ to $\mathcal{F}_1(x_{3n+1})$, but $\mathcal{F}_1(x_i)$ is not in the interval from $\mathcal{F}_1(x_{1+n})$ to $\mathcal{F}_1(x_{3n+1})$. By lemma G.5 and the fact that the label of $\mathcal{F}_1(x_{n+1})$ and $\mathcal{F}_1(x_{3n+1})$ is different from that of other three samples, we cannot find an interval satisfying the condition in lemma G.5, so $\mathcal{F}_2(x)$ cannot classify such five points: $\{(\mathcal{F}_1(x_i), y_i)\}_{i=1, n+1, 2n+1, 3n+1, i}$. This is contradictory, as $\mathcal{L}(\mathcal{D}(\varphi))$ is the memorization of $\mathcal{D}(\varphi)$. So, the assumption is wrong, we prove the theorem. $\qquad\square$

# H  Proof of Theorem 7.3

## H.1  Proof of Proposition 7.7

*Proof.* It suffices to prove that we can find an $S_c(\mathcal{D}) \subset \{(x, y) \| (x, y) \sim \mathcal{D}\}$ such that for any $(x, y) \sim \mathcal{D}$, we have $x \in \cup_{(z,w) \in S_c(\mathcal{D})} B((z, w))$.

Let $S_c = \{(i_1 c/(6.2n), i_2 c/(6.2n), \ldots, i_n c/(6.2n)) \| i_j \in \{0, 1, \ldots, [6.2n/c] + 1\}\}$, and define $S_c(\mathcal{D})$ as: for any $(i_1 c/(6.2n), i_2 c/(6.2n), \ldots, i_n c/(6.2n)) \in S_c$, randomly take a $(x, y) \sim \mathcal{D}$ satisfying $\|x - (i_1 c/(6.2n), i_2 c/(6.2n), \ldots, i_n c/(6.2n))\|_\infty \leq c/(6.2n)$ (if we have such a $x$), and put $(x, y)$ into $S_c(\mathcal{D})$.

Then, we have that, for any $(x, y) \sim \mathcal{D}$, there is a point $z \in S_c$ such that $\|z - x\|_\infty \leq c/(6.2n)$, and there is a $(x_z, y_z) \in S_c(\mathcal{D})$ such that $\|z - x_z\|_\infty \leq c/(6.2n)$, so $\|x_z - x\|_\infty \leq c/(3.1n)$, which implies $\|x - x_z\|_\infty \leq c/3.1$.

Since the radius of $B((z, w))$ is more than $c/3.1$, for any $(x, y) \sim \mathcal{D}$, we have $x \in \cup_{(z,w) \in S_c(\mathcal{D})} B((z, w))$, we prove the lemma. $\qquad\square$

## H.2  Main idea

For a given dataset $\mathcal{D}_{tr} \subset [0, 1]^n \times \{-1, 1\}$, we use the following two steps to construct a memorization network:

(c1): Find suitable convex sets $\{C_i\}$ in $[0, 1]^n$, ensuring that: each sample in $\mathcal{D}_{tr}$ is in at least one of these convex sets. Furthermore, if $x, z \in C_i$ and $(x, y_x), (z, y_z) \in \mathcal{D}_{tr}$, then $y_x = y_z$, and define $y(C_i) = y_x$.

(c2): Construct a network $\mathcal{F}$ satisfying that for any $x \in C_i$, $\mathrm{Sgn}(\mathcal{F}(x)) = y(C_i)$. Such a network must be a memorization of $\mathcal{D}_{tr}$, because each sample in $\mathcal{D}_{tr}$ is in at least one of $\{C_i\}$, so if $x \in C_i$ and $(x, y_x) \in \mathcal{D}_{tr}$, then $\mathrm{Sgn}(\mathcal{F}(x)) = y(C_i) = y_x$, which is the network we want.

## H.3 Finding convex sets

For a given dataset $\mathcal{D}_{tr} \subset [0,1]^n \times \{-1,1\}$, let $\mathcal{D}_{tr} = \{(x_i, y_i)\}_{i=1}^N$, and for $i \in [n]$, the convex sets $C_i$ are constructed as follows:

(1): For any $i,j \in [N]$, define $S_{i,j}(x) = (x_i - x_j)(x - (0.51 * x_i + 0.49 * x_j))$, it is easy to see that $S_{i,j}$ is a vertical between $x_i$ and $x_j$;

(2): The convex sets $C_i$ are defined as $C_i = \cap_{j \in [N], y_i \neq y_j} \{x \in [0,1]^n \| S_{i,j}(x) \geq 0\}$.

Now, we have the following lemma, which implies that $C_i$ satisfies conditions (c1) mentioned in above.

**Lemma H.1.** *If $C_i$ are constructed as above, then*

*(1): $x_i \in C_i$;*

*(2): If $z \in C_i$ and $(z, y_z) \in \mathcal{D}_{tr}$, then $y_z = y_i$;*

*(3): $C_i$ is a convex set.*

*Proof.* Firstly, we show that $x_i \in C_i$. For any $i,j \in [N]$, taking $x_i$ into $S_{i,j}(x)$, we have $S_{i,j}(x_i) = 0.49 \|x_i - x_j\|_2^2 > 0$, so $x_i \in \{x \in [0,1]^n \| S_{i,j}(x) \geq 0\}$. Thus $x_i \in \cap_{j \in [N], y_i \neq y_j} \{x \in [0,1]^n \| S_{i,j}(x) \geq 0\} = C_i$.

Then, we show that: if $y_j \neq y_i$, then $x_j \notin C_i$, which implies (2) of lemma is valid.

For any $i,j \in [N]$, taking $x_j$ into $S_{i,j}(x)$, we have $S_{i,j}(x_j) = -0.51 \|x_i - x_j\|_2^2 < 0$, so $x_j \notin \{x \in [0,1]^n \| S_{i,j}(x) \geq 0\}$. Thus $x_j \notin \cap_{k \in [N], y_i \neq y_k} \{x \in [0,1]^n \| S_{i,k}(x) \geq 0\} = C_i$.

Finally, we show $C_i$ is a convex set. Because for any $i,j \in [N]$, $\{x \in [0,1]^n \| S_{i,j}(x) \geq 0\}$ is a convex set, and the combination of convex sets is also convex set, so $C_i$ is a convex set. $\qquad\square$

## H.4 Construct the Network

We show how to construct a network $\mathcal{F}$, such that $\mathrm{Sgn}(\mathcal{F}(x)) = y(C_i)$ for any $x \in C_i$, where $C_i$ is defined in section H.3.

For a given dataset $\mathcal{D}_{tr} = \{(x_i, y_i)\}_{i=1}^N$, we construct a network $\mathcal{F}_{mem}$ which has three layers as following.

(1): Let $r = 0.01 * \min_{i,j \in [N], y_i \neq y_j} \|x_i - x_j\|_2^2$. For any $i,j \in [N]$, $S_{i,j}$ defined in section H.3, let $u_i(x) = \sum_{j \in [N], y_j \neq y_i} \mathrm{Relu}(-S_{i,j}(x)) - r$. It is easy to see that $u_i$ is a depth 2 network.

(2): The first two layers are $\mathcal{F}_1 : \mathbb{R}^n \to \mathbb{R}^N$. Let $\mathcal{F}_1(x)[i]$ be the $i$-th output of $\mathcal{F}_1(x)$, then let $\mathcal{F}_1(x)[i]$ equal to $\mathrm{Relu}(-u_i(x))$. It is easy to see that, $\mathcal{F}_1(x)$ requires $O(N^2 n)$ parameters.

(3): The third layer is $\mathcal{F}_2 : \mathbb{R}^N \to \mathbb{R}$, and $\mathcal{F}_2(v) = \sum_{i=1}^N y_i v_i$, where $v_i$ is the $i$-th weight of $v_i$.

Now, we prove that $\mathrm{Sgn}(\mathcal{F}_{mem}(x)) = y(C_i)$ for any $x \in C_i$. We need the following lemma.

**Lemma H.2.** *For any $x \in C_i$, we have $u_i(x) < 0$ and $u_j(x) > 0$ when $y_i \neq y_j$.*

*Proof.* Assume that $x \in C_i$. We prove the following two properties, and hence the lemma.

**P1.** $u_i(x) < 0$.

By the definition of $C_i$, we have $S_{i,j}(x) \geq 0$ for all $j \in [N]$ staisfying $y_i \neq y_j$, so $u_i(x) = \sum_{j \in [N], y_j \neq y_i} \mathrm{Relu}(-S_{i,j}(x)) - r = \sum_{j \in [N], y_j \neq y_i} 0 - r = -r < 0$.

**P2.** $u_j(x) > 0$ **when** $y_i \neq y_j$.

For any $j$ such that $y_i \neq y_j$, we show $S_{j,i}(x) \leq -0.02 \|x_i - x_j\|_2^2$ at first. Because $x \in C_i$, so $S_{i,j}(x) \geq 0$, that is $(x_i - x_j)(x - (0.51 * x_i + 0.49 * x_j)) \geq 0$, so

$$
\begin{aligned}
&(x_i - x_j)(x - (0.51 * x_i + 0.49 * x_j)) \\
=\ & (x_i - x_j)(x - (0.49 * x_i + 0.51 * x_j)) - 0.02 \|x_i - x_j\|_2^2 \\
=\ & -S_{j,i}(x) - 0.02 \|x_i - x_j\|_2^2 \\
\geq\ & 0.
\end{aligned}
$$

Thus $S_{j,i}(x) \leq -0.02\|x_i - x_j\|_2^2$. Then, by the above result, taking the value of $r$ in it, we have $u_j(x) \geq \text{Relu}(-S_{i,j}(x)) - r \geq 0.02\|x_i - x_j\|_2^2 - r > 0$. $\qquad \square$

By the above lemma, we can prove the result.

**Lemma H.3.** *we have* $\text{Sgn}(\mathcal{F}_{mem}(x)) = y_i$ *for any* $x \in C_i$.

*Proof.* Let $x \in C_i$. By lemma H.2, we have $\mathcal{F}_1(x)[i] > 0$, and $\mathcal{F}_1(x)[j] = 0$ when $j$ satisfies $y_j \neq y_i$, so $\mathcal{F}(x) = \sum_{j \in [N]} y_j \mathcal{F}_1(x)[j] = y_i \sum_{j \in [N], y_j = y_i} \mathcal{F}_1(x)[j]$, by $\mathcal{F}_1(x)[i] > 0$, and we thus have $\text{Sgn}(\mathcal{F}(x)) = y_i$. $\qquad \square$

## H.5 Effective and Generalization Guarantee

In this section, we prove that the above algorithm is an effective memorization algorithm with guaranteed generalization. We give a lemma.

**Lemma H.4.** *For any* $a, b, c \in \mathbb{R}^n$ *such that* $\|b - a\|_2 \geq 3.1\|a - c\|_2$, *let* $V$ *be the plane* $(b - c)(x - (0.51c + 0.49b))$. *Then the distance of* $a$ *to the plane* $V$ *is greater than* $\|b - a\|/3.1$.

*Proof.* Let $\|a - b\|_2 = L_{ab}$, $\|a - c\|_2 = L_{ac}$, $\|c - b\|_2 = L_{bc}$. Let the angle $\angle abc = \theta$. Then the distance between $a$ and the plane $V$ is $L_{ab} \cos\theta - 0.51 L_{bc}$.

Using cosine theorem, we have $\cos\theta = \frac{L_{bc}^2 + L_{ab}^2 - L_{ac}^2}{2L_{bc}L_{ab}}$, so we just need to prove that $\frac{L_{bc}^2 + L_{ab}^2 - L_{ac}^2}{2L_{bc}} - 0.51L_{bc} \geq L_{ab}/3.1$, that is $\frac{0.5L_{ab}^2 - 0.5L_{ac}^2 - L_{ab}L_{bc}/3.1}{L_{bc}^2} \geq 0.01$. It is easy to see that such value is inversely proportional to $L_{ac}$ and $L_{bc}$. By $L_{ac} \leq L_{ab}/3.1$ and $L_{bc} \leq L_{ac} + L_{ab} \leq 4.1L_{ab}/3.1$, we have $\frac{0.5L_{ab}^2 - 0.5L_{ac}^2 - L_{ab}L_{bc}/3.1}{L_{bc}^2} \geq \frac{0.5 - 0.5/(3.1)^2 - 4.1/(3.1)^2}{(4.1/3.1)^2} > 0.01$. The lemma is proved. $\qquad \square$

We now show that the algorithm is effective and has generalization guarantee.

*Proof.* Let $\mathcal{F}_{mem}$ be the memorization network of $\mathcal{D}_{tr}$ constructed by the above algorithm.

**Effective.** We show that $\mathcal{F}_{mem}$ is a memorization of $\mathcal{D}_{tr}$ can be constructed in polynomial time.

It is easy to see that, $u_i$ has width at most $N$, and each value of parameters can be calculated by $\mathcal{D}_{tr}$ in polynomial time. So $\mathcal{F}_1$ defined in (1) in section H.4 can be calculated in polynomial time. It is easy to see that the $\mathcal{F}_2$ defined in (1) in section H.4 can be calculated in polynomial time. This, $\mathcal{F}$ can be calculated in polynomial time.

**Generalization Guarantee.** Let $S = \{(v_i, y_{v_i})\}_{i=1}^{S_\mathcal{D}}$ be the nearby set defined in Definition 7.1. Then, we show the result in two parts.

**Part One**, we show that: for a $(x_i, y_i) \in \mathcal{D}_{tr}$, if $x_i \in B((v_j, y_{v_j}))$ for a $j \in [S_\mathcal{D}]$, then $\text{Sgn}(\mathcal{F}(x)) = y_i$ for any $x \in B((v_j, y_{v_j}))$.

Firstly, we show that it holds $B((v_j, y_{v_j})) \in C_i$. For any $k \in [N]$ such that $y_k \neq y_i$, we have $\|v_j - x_k\|_2 \geq 3.1r \geq 3.1\|v_j - x_i\|$, where $r$ is the radius of $B((v_j, y_{v_j}))$ so by lemma H.4, the distance from $v_j$ to $S_{ik}(x)$ is greater than $r$, which means that the points in $B((v_j, y_{v_j}))$ are on the same side of the plane $S_{ik}(x)$, by $x_i \in B((v_j, y_{v_j}))$ and $S_{ik}(x_i) > 0$ as said in lemma H.1. Thus, for any $x \in B((v_j, y_{v_j}))$, we have $S_{ik}(x) \geq 0$. By $C_i = \cap_{j \in [N], y_i \neq y_j}\{x \in [0,1]^n \| S_{i,j}(x) \geq 0\}$, we know that $B((v_j, y_{v_j})) \in C_i$.

By the above result, if $x \in B((v_j, y_{v_j}))$, then $x \in C_i$; so by lemma H.3, we have $\text{Sgn}(\mathcal{F}(x)) = y_i$ for all $x \in B((v_j, y_{v_j}))$.

**Part Two**, we show that if $\mathcal{D}_{tr} \sim \mathcal{D}^N$ and $N \geq S_\mathcal{D}/\epsilon \ln(S_\mathcal{D}/\delta)$, then $\mathbb{P}_{\mathcal{D}_{tr} \sim \mathcal{D}^N}(A_\mathcal{D}(\mathcal{F}_{mem}) \geq 1 - \epsilon) \geq 1 - \delta$.

Let $Q_i = \mathbb{P}_{(x,y) \sim \mathcal{D}}(x \in B((v_i, y_{v_i})))$, then without losing generality, we assume that $Q_1 \leq Q_2 \leq \cdots \leq Q_{S_\mathcal{D}}$. Then, for the dataset $\mathcal{D}_{tr} = \{(x_i, y_i)\}_{i=1}^N$, let $Z(\mathcal{D}_{tr}) = \{j \in [S_\mathcal{D}] \| \exists i \in [N], x_i \in B((v_j, y_{v_j}))\}$. The proof is given in three parts.

**part 2.1.** Firstly, we show that $A_{\mathcal{D}}(\mathcal{F}_{mem}) \geq 1 - \sum_{i \notin Z(\mathcal{D}_{tr})} Q_i$.

If $i \in Z(\mathcal{D}_{tr})$, then by the definition of $Z(\mathcal{D}_{tr})$, we know that there is a $j \in [N]$ such that $x_j \in B((v_i, y_{v_i}))$, so by part one, we have $\text{Sgn}(\mathcal{F}_{mem}(x)) = y_j$ for any $x \in B((v_i, y_{v_i}))$.

Moreover, for any $(x, y) \sim \mathcal{D}$ and $x \in B((v_i, y_{v_i}))$, by lemma H.1 and $B((v_i, y_{v_i})) \in C_j$ which has been shown in part one, we know that $y = y_j$.

So $\text{Sgn}(\mathcal{F}_{mem}(x)) = y_j = y$ for any $(x, y) \sim \mathcal{D}$ and $x \in B((v_i, y_{v_i}))$, which means that $\mathcal{F}_{mem}$ gives the correct label to all $x \in B((v_i, y_{v_i}))$ when $i \in Z(\mathcal{D}_{tr}, S)$. So $A_{\mathcal{D}}(\mathcal{F}_{mem}) \geq \sum_{i \in Z(\mathcal{D}_{tr}, S)} Q_i \geq 1 - \sum_{i \notin Z(\mathcal{D}_{tr}, S)} Q_i$.

**part 2.2.** Now, we show that $\mathbb{P}_{\mathcal{D}_{tr} \sim \mathcal{D}^N}(\sum_{i \notin Z(\mathcal{D}_{tr})} Q_i \leq \epsilon) \geq 1 - \delta$.

Let $Cc_i = \{\mathcal{D}_{tr} \| \mathcal{D}_{tr} \sim \mathcal{D}^N, i \notin Z(\mathcal{D}_{tr}) \ and \ j \in Z(\mathcal{D}_{tr}) \ for \ \forall j > i\}$, easy to see that $Cc_j \cap Cc_i = \emptyset$ when $i \neq j$ and $\sum_{i=0}^{N} \mathbb{P}_{\mathcal{D}_{tr} \sim \mathcal{D}^N}(\mathcal{D}_{tr} \in Cc_i) = 1$. It is easy to see that, $\mathbb{P}_{\mathcal{D}_{tr} \sim \mathcal{D}^N}(\mathcal{D}_{tr} \in Cc_i) \leq (1 - Q_i)^N$ when $i \geq 1$.

Firstly we have that, if some $i \in [S_{\mathcal{D}}]$ makes that $Q_i < \epsilon/i$, then for any $\mathcal{D}_{tr} \in Cc_j$ where $j \leq i$, we have $\sum_{k \notin Z(\mathcal{D}_{tr})} Q_k \leq \sum_{k=1}^{j} Q_k \leq jQ_j \leq iQ_i < \epsilon$.

So that, we consider two situations.

**Situation 1: There is a $i \in [S_{\mathcal{D}}]$ such that $Q_i < \epsilon/i$.**

Let $N_0$ be the biggest number in $[S_{\mathcal{D}}]$ such that $Q_{N_0} < \epsilon/N_0$. Then we have that:

$$
\begin{aligned}
& \mathbb{P}_{\mathcal{D}_{tr} \sim \mathcal{D}^N}(\sum_{i \notin Z(\mathcal{D}_{tr})} Q_i \leq \epsilon) \\
= \ & \mathbb{P}_{\mathcal{D}_{tr} \sim \mathcal{D}^N}(\sum_{i \notin Z(\mathcal{D}_{tr})} Q_i \leq \epsilon \| \mathcal{D}_{tr} \in \cup_{k=0}^{N_0} Cc_k) \mathbb{P}_{\mathcal{D}_{tr} \sim \mathcal{D}^N}(\mathcal{D}_{tr} \in \cup_{k=0}^{N_0} Cc_k) \\
& + \mathbb{P}_{\mathcal{D}_{tr} \sim \mathcal{D}^N}(\sum_{i \notin Z(\mathcal{D}_{tr})} Q_i \leq \epsilon \| \mathcal{D}_{tr} \in \cup_{k=N_0+1}^{[S_{\mathcal{D}}]} Cc_k) \mathbb{P}_{\mathcal{D}_{tr} \sim \mathcal{D}^N}(\mathcal{D}_{tr} \in \cup_{k=N_0+1}^{[S_{\mathcal{D}}]} Cc_k) \quad (5) \\
= \ & \mathbb{P}_{\mathcal{D}_{tr} \sim \mathcal{D}^N}(\mathcal{D}_{tr} \in \cup_{k=0}^{N_0} Cc_k) + \mathbb{P}_{\mathcal{D}_{tr} \sim \mathcal{D}^N}(\sum_{i \notin Z(\mathcal{D}_{tr})} Q_i \leq \epsilon \| \mathcal{D}_{tr} \in \cup_{k=N_0+1}^{[S_{\mathcal{D}}]} Cc_k) \\
& \mathbb{P}_{\mathcal{D}_{tr} \sim \mathcal{D}^N}(\mathcal{D}_{tr} \in \cup_{k=N_0+1}^{[S_{\mathcal{D}}]} Cc_k).
\end{aligned}
$$

Hence, we have

$$
\begin{aligned}
& \mathbb{P}_{\mathcal{D}_{tr} \sim \mathcal{D}^N}(\mathcal{D}_{tr} \in \cup_{k=N_0+1}^{[S_{\mathcal{D}}]} Cc_k) \\
\leq \ & \sum_{i=N_0+1}^{S_{\mathcal{D}}} \mathbb{P}_{\mathcal{D}_{tr} \sim \mathcal{D}^N}(\mathcal{D}_{tr} \in Cc_i) \\
\leq \ & \sum_{i=N_0+1}^{S_{\mathcal{D}}} (1 - Q_i)^N \\
\leq \ & \sum_{i=N_0+1}^{S_{\mathcal{D}}} e^{-NQ_i} \\
\leq \ & \sum_{i=N_0+1}^{S_{\mathcal{D}}} e^{-N\epsilon/i} \\
\leq \ & \sum_{i=1}^{S_{\mathcal{D}}} e^{-N\epsilon/i} \\
\leq \ & S_{\mathcal{D}} e^{-N\epsilon/S_{\mathcal{D}}} \\
\leq \ & \delta.
\end{aligned}
$$

The last step is to take $N \geq S_{\mathcal{D}}/\epsilon \ln(S_{\mathcal{D}}/\delta)$ in. So, taking the above result in equation 5, we have

$$
\begin{aligned}
& \mathbb{P}_{\mathcal{D}_{tr} \sim \mathcal{D}^N}(\sum_{i \notin Z(\mathcal{D}_{tr})} Q_i \leq \epsilon) \\
\geq \ & 1 - \delta + \mathbb{P}_{\mathcal{D}_{tr} \sim \mathcal{D}^N}(\sum_{i \notin Z(\mathcal{D}_{tr})} Q_i \leq \epsilon \| \mathcal{D}_{tr} \in \cup_{k=N_0+1}^{[S_{\mathcal{D}}]} Cc_k)\delta \\
\geq \ & 1 - \delta
\end{aligned}
$$

which is what we want.

**Situation 2: There is no $i \in [S_{\mathcal{D}}]$ such that $Q_i < \epsilon/i$.**

Then, we have

$$
\begin{aligned}
& \mathbb{P}_{\mathcal{D}_{tr} \sim \mathcal{D}^N}(\mathcal{D}_{tr} \in \cup_{k=1}^{[S_{\mathcal{D}}]} Cc_k) \\
\leq \; & \sum_{i=1}^{S_{\mathcal{D}}} \mathbb{P}_{\mathcal{D}_{tr} \sim \mathcal{D}^N}(\mathcal{D}_{tr} \in Cc_i) \\
\leq \; & \sum_{i=1}^{S_{\mathcal{D}}} (1 - Q_i)^N \\
\leq \; & \sum_{i=1}^{S_{\mathcal{D}}} e^{-NQ_i} \\
\leq \; & \sum_{i=1}^{S_{\mathcal{D}}} e^{-N\epsilon/i} \\
\leq \; & S_{\mathcal{D}} e^{-N\epsilon/S_{\mathcal{D}}} \\
\leq \; & \delta.
\end{aligned}
$$

So with probability $1 - \delta$, we have $\mathcal{D}_{tr} \in Cc_0$. When $\mathcal{D}_{tr} \in Cc_0$, we have $Z(\mathcal{D}_{tr}) = [S_{\mathcal{D}}]$, so that $\sum_{i \notin Z(\mathcal{D}_{tr})} Q_i = 0$. Hence, $\mathbb{P}_{\mathcal{D}_{tr} \sim \mathcal{D}^N}(\sum_{i \notin Z(\mathcal{D}_{tr})} Q_i \leq \epsilon) \geq 1 - \delta$.

**part 2.3** Now we can prove the part 2, by part 2.1 and part 2.2, we have that $\mathbb{P}_{\mathcal{D}_{tr} \sim \mathcal{D}^N}(A_{\mathcal{D}}(\mathcal{F}_{mem}) \geq 1 - \epsilon) \geq \mathbb{P}_{\mathcal{D}_{tr} \sim \mathcal{D}^N}(1 - \sum_{i \notin Z(\mathcal{D}_{tr},S)} Q_i \geq 1 - \epsilon) \geq 1 - \delta$. The theorem is proved. $\square$

# I   Experiments

We try to verify Theorem 7.3 on MNIST and CIFAR10 [33].

## I.1   Experiment on MNIST

For MNIST, we tested all binary classification problems with different label compositions. For each pair of labels, we use 500 corresponding samples with each label in the original dataset to form a new dataset $\mathcal{D}_{tr}$, and then construct memorization network for $\mathcal{D}_{tr}$ by Theorem 7.3. For each binary classification problem, Table 1 shows the accuracy on the samples with such two labels in testset.

Table 1: On MNIST, accuracy for all binary classification problems with different label compositions, use memorization algorithm by theorem 7.3. The result in row $i$ and column $j$ is the result for classifying classes $i$ and $j$.

| category | 0 | 1 | 2 | 3 | 4 | 5 | 6 | 7 | 8 | 9 |
|---|---|---|---|---|---|---|---|---|---|---|
| 0 | - | 0.99 | 0.96 | 0.99 | 0.99 | 0.97 | 0.96 | 0.98 | 0.98 | 0.97 |
| 1 | 0.99 | - | 0.97 | 0.99 | 0.98 | 0.99 | 0.98 | 0.98 | 0.98 | 0.99 |
| 2 | 0.96 | 0.97 | - | 0.96 | 0.97 | 0.96 | 0.96 | 0.97 | 0.93 | 0.97 |
| 3 | 0.99 | 0.99 | 0.96 | - | 0.98 | 0.95 | 0.98 | 0.95 | 0.92 | 0.96 |
| 4 | 0.99 | 0.98 | 0.97 | 0.98 | - | 0.98 | 0.97 | 0.96 | 0.95 | 0.91 |
| 5 | 0.97 | 0.99 | 0.96 | 0.95 | 0.95 | - | 0.96 | 0.97 | 0.91 | 0.96 |
| 6 | 0.96 | 0.98 | 0.96 | 0.98 | 0.97 | 0.96 | - | 0.99 | 0.95 | 0.98 |
| 7 | 0.98 | 0.98 | 0.97 | 0.95 | 0.96 | 0.97 | 0.99 | - | 0.95 | 0.91 |
| 8 | 0.98 | 0.98 | 0.93 | 0.92 | 0.95 | 0.91 | 0.95 | 0.95 | - | 0.96 |
| 9 | 0.97 | 0.99 | 0.97 | 0.96 | 0.91 | 0.96 | 0.98 | 0.91 | 0.96 | - |

From Table 1, we can see that the algorithm shown in the theorem 7.3 has good generalization ability for mnist, almost all result is higher than $90\%$.

## I.2   Experiment on CIFAR10

For CIFAR10, we test all binary classification problems with different label combinations. For each pair of labels, we use 3000 corresponding samples with each label in the original dataset to form a new dataset $\mathcal{D}_{tr}$, and then construct memorization network for $\mathcal{D}_{tr}$ by Theorem 7.3. For each binary classification problem, Table 2 shows the accuracy on the samples with such two labels in testset.

From Table 2, we can see that, most of the accuracies are above $70\%$, but for certain pairs, the results may be poor, such as cat and dog (category 3 and category 5).

Our memorization algorithm cannot exceed the training methods empirically. Training, as a method that has been developed for a long time, is undoubtedly effective. For each pair of labels, we use 3000 corresponding samples with each label in the original dataset to form a training set $D_{tr}$, and train Resnet18 [28] on $\mathcal{D}_{tr}$ (with 20 epochs, learning rate 0.1, use crossentropy as loss function, device is GPU NVIDIA GeForce RTX 3090), the accuracy of the obtained network is shown in Table 3.

Table 2: On CIFAR10, accuracy for all binary classification problems with different label compositions, use memorization algorithm by theorem 7.3. The result in row $i$ and column $j$ is the result for classifying classes $i$ and $j$.

| category | 0 | 1 | 2 | 3 | 4 | 5 | 6 | 7 | 8 | 9 |
|---|---|---|---|---|---|---|---|---|---|---|
| 0 | - | 0.77 | 0.74 | 0.78 | 0.81 | 0.81 | 0.85 | 0.85 | 0.68 | 0.73 |
| 1 | 0.77 | - | 0.78 | 0.75 | 0.82 | 0.78 | 0.82 | 0.87 | 0.79 | 0.63 |
| 2 | 0.74 | 0.78 | - | 0.61 | 0.61 | 0.65 | 0.67 | 0.67 | 0.82 | 0.77 |
| 3 | 0.78 | 0.75 | 0.61 | - | 0.71 | 0.54 | 0.67 | 0.69 | 0.83 | 0.76 |
| 4 | 0.81 | 0.82 | 0.61 | 0.71 | - | 0.66 | 0.62 | 0.65 | 0.82 | 0.79 |
| 5 | 0.81 | 0.78 | 0.65 | 0.54 | 0.66 | - | 0.73 | 0.67 | 0.81 | 0.78 |
| 6 | 0.85 | 0.82 | 0.67 | 0.67 | 0.62 | 0.73 | - | 0.71 | 0.86 | 0.81 |
| 7 | 0.85 | 0.87 | 0.67 | 0.69 | 0.65 | 0.67 | 0.71 | - | 0.82 | 0.73 |
| 8 | 0.68 | 0.79 | 0.82 | 0.83 | 0.82 | 0.81 | 0.86 | 0.82 | - | 0.69 |
| 9 | 0.73 | 0.63 | 0.77 | 0.76 | 0.79 | 0.78 | 0.81 | 0.73 | 0.69 | - |

Table 3: On CIFAR10, accuracy for all binary classification problems with different label compositions, use normal training algorithm. The result in row $i$ and column $j$ is the result for classifying classes $i$ and $j$.

| category | 0 | 1 | 2 | 3 | 4 | 5 | 6 | 7 | 8 | 9 |
|---|---|---|---|---|---|---|---|---|---|---|
| 0 | - | 0.99 | 0.98 | 0.99 | 0.99 | 0.99 | 0.99 | 0.99 | 0.98 | 0.99 |
| 1 | 0.99 | - | 0.99 | 0.98 | 0.99 | 0.99 | 0.99 | 0.99 | 0.99 | 0.99 |
| 2 | 0.98 | 0.99 | - | 0.99 | 0.99 | 0.99 | 0.99 | 0.99 | 0.99 | 0.99 |
| 3 | 0.99 | 0.98 | 0.99 | - | 0.98 | 0.96 | 0.97 | 0.99 | 0.98 | 0.99 |
| 4 | 0.99 | 0.99 | 0.99 | 0.98 | - | 0.99 | 0.99 | 0.99 | 0.99 | 0.99 |
| 5 | 0.99 | 0.99 | 0.99 | 0.96 | 0.99 | - | 0.99 | 0.99 | 0.99 | 0.99 |
| 6 | 0.99 | 0.99 | 0.99 | 0.97 | 0.99 | 0.99 | - | 0.98 | 0.99 | 0.99 |
| 7 | 0.99 | 0.99 | 0.99 | 0.99 | 0.99 | 0.99 | 0.98 | - | 0.99 | 0.99 |
| 8 | 0.98 | 0.99 | 0.99 | 0.98 | 0.99 | 0.99 | 0.99 | 0.99 | - | 0.99 |
| 9 | 0.99 | 0.99 | 0.99 | 0.99 | 0.99 | 0.99 | 0.99 | 0.99 | 0.99 | - |

Comparing Tables 2 and 3, it can be seen that the training results are significantly better.

## I.3 Compare with other memorization algorithm

Three memorization network construction methods are considered in this section: (M1): Our algorithm in theorem 7.3; (M2): Method in [49]; (M3): Method in [55].

In particular, we do experiments on the classification of such five pairs of numbers in MNIST: 1 and 7, 2 and 3, 4 and 9, 5 and 6, 8 and 9, to compare methods M1, M2, M3. The main basis for selecting such pairs of labels is the similarity of the numbers. For any pair of numbers, we label the smaller number as -1 and the larger number as 1. Other settings follow section I.1, and the result is given in Table 4. We can see that our method performs much better in all cases.

From table 4, our method gets the best accuracy. When constructing a memorization network, the methods (M2), (M3) compress data into one dimension, such action will break the feature of the image, so they cannot get a good generalization.

Table 4: On MNIST, accuracy about different memorization algorithm.

| pair (1,7) | Accuracy |
|---|---|
| M1 | 0.98 |
| M2 | 0.51 |
| M3 | 0.46 |
| pair (2,3) | Accuracy |
| M1 | 0.96 |
| M2 | 0.50 |
| M3 | 0.51 |
| pair (4,9) | Accuracy |
| M1 | 0.91 |
| M2 | 0.45 |
| M3 | 0.46 |
| pair (5,6) | Accuracy |
| M1 | 0.96 |
| M2 | 0.59 |
| M3 | 0.47 |
| pair (8,9) | Accuracy |
| M1 | 0.96 |
| M2 | 0.41 |
| M3 | 0.48 |

