# OpenReview forum: "Generalizablity of Memorization Neural Network"
_NeurIPS.cc/2024/Conference — NeurIPS 2024 poster_

### Official Review · Reviewer_8736 · 2024-06-19

**Soundness:** 3
**Presentation:** 4
**Contribution:** 4
**Rating:** 7
**Confidence:** 4

**Summary:**

The paper studies the generalization capabilities of memorization networks, specifically networks that achieve optimal memorization capacity (i.e. O(sqrt(N)) parameters for memorizing N samples). The authors present a memorization algorithm that is based on the construction of Vardi et al. Next, they show that memorization with fixed width or small number of parameters cannot generalize well. They also provide upper and lower bounds for the communication complexity of learning with a memorization algorithm, while also providing an efficient memorization algorithm.

**Strengths:**

- I think this research direction is novel and interesting. It puts a spotlight on memorization results, which may be efficient in terms of expressiveness but their generalization capabilities are unclear.

- The presentation of the paper is good and it is easy to follow, while also providing sufficient proof intuitions.

- Section 5 is particularly interesting since it shows the limitations of efficient memorization results, and specifically that memorization networks cannot generalize well.

- Section 6 is also very extensive, and provides tight sample complexity bounds on memorization learning, that connects the size of the memorization network to the number of required samples for learning.

**Weaknesses:**

I think this paper is very nice overall, however, there are some issues with specific results which I would be happy to see the author’s response about:
- Proposition 3.8 - where is the proof? Also, it is not clear to me why the constructions in [54,48] are probabilistic.

- There is something unclear about the proof of Theorem 5.3. The theorem statement is that for every distribution D and any test set D_tr there exists a memorization network that doesn’t generalize. However, in the proof, the authors begin with a given set D_tr, and then define D based on this training set, i.e. D is defined by some set S which in itself depends on D_tr. I think the statement should be that there exists a distribution D such that for all training set D_tr  there exists a memorization network that doesn’t generalize. Note that the current proof doesn’t show it, because D is constructed such that it depends on D_tr.

- The comparison between interpolation and memorization learning is unclear. Specifically, in what cases do the results on interpolation learning already provide generalization bounds for memorization learning?

Some minor comments:

- Results and comparisons on implicit bias toward margin maximization of neural networks do not appear (e.g. https://arxiv.org/abs/1906.05890, https://arxiv.org/abs/2006.06657). They also study a regime where the optimization converged to 100% accuracy on the dataset and provide generalization guarantees based on margin arguments.

- In the main contribution, point 3, it is better to write lower bound is \Omega rather than O.

**Questions:**

- How large can S_D be? It seems that for any continuous distribution on [0,1]^n, S_D is already exponential in n, hence having such a large sample complexity makes this bound vacuous. Can the authors provide some examples where S_D and N_D are not very far apart?

- What is the size (num of parameters, width, depth) of the network defined in Theorem 7.3?

**Limitations:**

The authors discuss the limitations of their results.

---

> ### Author Rebuttal · Authors · 2024-08-05
>
> The authors thank the reviewer for the valuable and insightful questions and hope that we have answered these questions satisfactorily.
>
> Question 1. Proposition 3.8 - where is the proof? Also, it is not clear to me why the constructions in [54,48] are probabilistic.
>
> We deduce proposition 3.8 from the proof process of theorem in these papers, considering that proposition 3.8 can be directly obtained from the proof in the corresponding paper, the proof is not given. In the revised version of the paper, we will give a sketch of the proof for Proposition 3.8. This randomness comes from a technique of memorization: in the construction of a memorization network, we require a special vector to compress data into one dimension, but this vector is not easy to calculate directly. However, every time we randomly select a vector, there is a non-zero (0.5) chance of selecting the required vector, and the randomness comes from this random selection. We have described some similarities in the proof of Theorem 4.1, including lines 220-223 and 648-649 (and the proof above lines 648-649) of the paper.
>
> Question 2. There is something unclear about the proof of Theorem 5.3.
>
> The $D_{tr}$ and $D$ used in our proof in Appendix E4 is the $D_{tr}$ and $D$ in Theorem 5.3, they can be any distribution $D$ and a dataset $D_{tr}$ selected from it, and we will clarify this point in the future. Our proof is for any distribution $D$. The core of the proof is in lines 1060-1065, and the proof in these lines applies to any distribution. Lemma E2 is also for any distribution $D$. Therefore, the whole proof is really for any distribution. In the revised version of the paper, we will make this clear.
>
> Question 3. The comparison between interpolation and memorization learning is unclear. Specifically, in what cases do the results on interpolation learning already provide generalization bounds for memorization learning?
>
> Generalization bounds for interpolation learning are only established  in linear or approximately linear situations. Our sample complexity results are for all memrozaition networks and hence are valid in all cases.
>
> Question 4. Results and comparisons on implicit bias toward margin maximization of neural networks do not appear.
>
> Thank you for pointing out this. There are many articles that analyze the margin maximization or generalization under gradient. These works can prove the results under some assumptions, such as NTK, Lip conditions, two-layer networks, etc., which are very good. In comparison, our results consider all the memorization algorithms and we have not made so many assumptions, so we  believe that our work has certain advantages. We will add these in the revised version of the paper.
>
> Question 5. The size of $N_D$ and $S_D$.
>
> Firstly, in most cases, we have $N_D<S_D$, and $S_D$ is more susceptible to the influence of separation bound. Second, in the worst-case scenario, they are exponential in n, and this is inevitable as shown in Corollary 6.4, where a lower bound $2^{O(n)}$ for $N_D$ is given. Also, in line 1425, an upper bound $n^{O(n)}$ for $S_D$ is given. Third, for some ``good'' distributions, we believe that $N_D$ and $S_D$ are reasonably small, from the nice practical performance of deep learning.
>
> For some simple situations, we can estimate their values. We use linear data as a simple example to calculate $N_D$ and $S_D$.
>
> If the distribution $D\in R^n$ is linear separable and defined as: $x\in[0,1]^n$ has label $1$ if $\langle\mathbf{1},x\rangle>199n/200$, $x\in[0,1]^n$ has label $-1$ if $\langle\mathbf{1},x\rangle<n/200$, where $\mathbf{1}$ is the vector whose weight is all 1.
>
> Then $N_D=O(n)$ for any $c$, because $D$ is linear separable.
>
> And $S_D=2$. Because the distance between the points with different labels is at least $0.49\sqrt{n}$; and the distance between point with label 1 and point $\mathbf{1}$ is at most  $\sqrt{n/50}$; and the distance between point with label -1 and origin is at most $\sqrt{n/50}$. By $0.49/3.1>\sqrt{1/50}$, then let $S=\\{0,\mathbf{1}\\}$ and by definition 7.1, we know that $S$ is a nearby set of $D$, then $S_D=2$.
>
> Question 6. What is the size of the network defined in Theorem 7.3?
>
> The size of the network is affected by the data distribution. In the worst-case, the network has depth 3, width N, and $O(N^2)$ parameters. As said in lines 103-107, to ensure generalization, we abandoned many classic techniques, which resulted in an increase of parameters. We will add this information in the revised version.

---

> > ### Comment · Reviewer_8736 · 2024-08-10
> >
> > Thank you for your answer, my questions have been answered and I will keep my score.

---

> > > ### Author Response · Authors · 2024-08-11
> > >
> > > Thank you again for your nice insights！

---

### Official Review · Reviewer_QwyD · 2024-07-09

**Soundness:** 3
**Presentation:** 3
**Contribution:** 3
**Rating:** 7
**Confidence:** 3

**Summary:**

This work studies the generalization properties of neural network memorization algorithms. Several results are proved: (1) Construction of a memorization network for a sampled dataset, with an optimal number of parameters. (2) There exists a constant such that for all datasets sampled from a distribution, there exists a memorization network with at most the constant number of parameters. (3) Sample complexity lower and upper bounds for memorization algorithms. (4) An efficient memorization algorithm with a generalization guarantee.

**Strengths:**

Solid mathematical paper with several novel results in DL theory. Mostly clearly written.

**Weaknesses:**

-	The significance of the theoretical results are not very clear because no estimates of $N_\mathcal{D}$ and $S_\mathcal{D}$ are given.

-	Writing can be improved, e.g.: “is also a challenge problem” line 408, “which is more challenging compare to linear model” line 44, “and shows that even if there is enough data” line 63, “In other word” line 87.

**Questions:**

Can estimates of $N_\mathcal{D}$ and $S_\mathcal{D}$ be provided even for simple toy examples (e.g., linearly separable data)?

**Limitations:**

Yes.

---

> ### Author Rebuttal · Authors · 2024-08-05
>
> For Reviewer QwyD:
>
> The authors thank the reviewer for the valuable and insightful questions and hope that we have answered these questions satisfactorily. Also thank you for pointing out the typos in our writing, we will correct them in future versions.
>
> Question 1. No estimates of $N_D$ and $S_D$  are given. Can $N_D$ and $S_D$ be provided even for simple toy examples (e.g., linearly separable data)?
>
> (1): The meaning of $N_D$ and $S_D$ in terms of generalization. $O(N^2_D)$ represents the minimum amount of data required for any memorization algorithm to achieve generalization, while $O(S_D)$ represents the upper bound of the amount of data required for achieving generalization by efficient memorization algorithms. We mainly pointed out the existence of these two values, but estimated them is not the core of the paper, the estimation of them can be a future work.
>
> (2): How big are $N_D$ and $S_D$?
> Firstly, in most cases, we have $N_D<S_D$, and $S_D$ is more susceptible to the influence of separation bound. Second, in the worst-case scenario, they are exponential in n, and this is inevitable as shown in Corollary 6.4, where a lower bound $2^{O(n)}$ for $N_D$ is given. Also, in line 1425, an upper bound $n^{O(n)}$ for $S_D$ is given. Third, for some ``good'' distributions, we believe that $N_D$ and $S_D$ are reasonably small, from the nice practical performance of deep learning and also be showen in the following simple example.
>
> (3): Can we calculate $N_D$ and $S_D$? Unfortunately, because these two values are closely related to the properties of the distribution $D$, calculating these two values for any distribution is not easy, as we said in line 405 to 406. But for some simple situations, we can estimate their values. We use linear data as an example to calculate $N_D$ and $S_D$.
>
> If the distribution $D\in R^n$ is linear separable and defined as: $x\in[0,1]^n$ has label $1$ if $\langle\mathbf{1},x\rangle>199n/200$, $x\in[0,1]^n$ has label $-1$ if $\langle\mathbf{1},x\rangle<n/200$, where $\mathbf{1}$ is the vector whose weight is all 1.
>
> Then $N_D=O(n)$ because $D$ is linear separable.
>
> And $S_D=2$. Because the distance between the points with different labels is at least $0.49\sqrt{n}$; and the distance between point with label 1 and point $\mathbf{1}$ is at most  $\sqrt{n/50}$; and the distance between point with label -1 and origin is at most $\sqrt{n/50}$. By $0.49/3.1>\sqrt{1/50}$, then let $S=\\{0,\mathbf{1}\\}$ and by definition 7.1, we know that $S$ is a nearby set of $D$, then $S_D=2$.

---

### Official Review · Reviewer_R3pV · 2024-07-12

**Soundness:** 4
**Presentation:** 4
**Contribution:** 3
**Rating:** 6
**Confidence:** 4

**Summary:**

This paper provides a first, thorough theoretical understanding of the generalization ability of the memorized neural network, including the existence of the memorized neural network, the matching upper bound and lower bound on the sample complexity, the computational infeasibility of attaining optimal sample complexity, and a computation-efficient but sub-optimal remedy.

**Strengths:**

Overall, the paper is well-written, the results are novel and comprehensive, and it makes a clear contribution to the theoretical community.

**Weaknesses:**

There are some comments, shown in the Questions below, but I recommend accepting this paper.

**Questions:**

Major Comments:
1. I think the hypothesis class D(n,c) is simple to be learned. I guess, for fixed (c, n), if the density in the support of X is bounded from below and above, then the minimax (worst D) error rate in D is the parametric rate? Hence this is a relatively easy class of function to be analyzed, and there is almost no noise, which may weaken the results of this paper.
2. The existence result in Theorem 4.3 is nice. But it should be clear whether the memorization algorithm L is dependent on D, or at least c. It will be good if such an algorithm can obtain memorization agnostic to the knowledge of c.
3. The upper bound and lower bound does not exactly match in its current form in terms of epsilon. I guess the lower bound is not tight, because it does not rule out the possibility that a fixed number of samples can have exact recovery (eps=delta=0), this is relatively loose. Can it be improved?



Minor comments:
1. I think the introduction part can make a clear, or informal definition of what is generalization under the well-separated classification task given it lists all the theoretical results.

**Limitations:**

See the Questions above.

---

> ### Author Rebuttal · Authors · 2024-08-05
>
> The authors thank the reviewer for the valuable and insightful questions and hope that we have answered these questions satisfactorily.
>
> Question 1. Why consider the distribution D(n,c) defined in Definition 3.1?
>
> D(n,c) is indeed a distribution with relatively good properties, as we explain this in Remark 3.2 of the paper, there are three main reasons and advantages to use such a distribution:
>
> (1): Proposition 3.3 shows that there exists a distribution D not in D(n,c),  such that any network is not generalizable over D, so if conisdering general distributions, we cannot have a general generalization theory at all.
>
> (2): D(n,c) is suitable for most of real-world distributions, like image classification.
>
> (3): D(n,c)  is abstracted from previous related work. In order to achieve memorization with networks with sub-linear number of parameters, it requires that data has a positive separation bound, like [48] and [54]. So, when studying the distribution in $D(n,c)$, many related techniques and classic conclusions can be used in the memorization, such as [48] and [54], on the other hand, we need to point out that the existing conclusions cannot directly derive our theorem.
>
> For the relationship between the parametric rate and error rate you mentioned, we need to point out that if we know which regions the network can correctly classify, we can indeed estimate the error rate using density. However, based on our research, we only know that the network has memorized a dataset, and we cannot directly obtain which regions the network can correctly predict.
> Research on more complex distributions or noisy distributions can be a future research direction.
>
> Question 2. In Theorem 4.3, whether the memorization algorithm L is dependent on D, or at least c. It will be good if such an algorithm can obtain memorization agnostic to the knowledge of c.
>
> In the proof, such algorithm is depended on c. But it is easy to make it no longer dependent on c, we will add an additional step to the memorization algorithm, we calculate the shortest distance $c '$ between two samples with different labels in the input dataset at first, easy to see that $c'\ge c$, and then use $c'$ to replacing $c$ in the proof of th4.3. After doing this, such an algorithm is  completely only dependent on the input dataset, and for any distribution, there is an upper bound on the number of parameters of output network.
>
> This is a good suggestion, and we will make modifications in future versions, thank you.
>
> Question 3. The upper bound and lower bound does not exactly match in its current form in terms of epsilon.
>
> You are right that this lower bound only reaches tightness for $N^2_D$, but there is still room to improve the factor $(1-2\epsilon-\delta)$. But currently, we do not achieve this yet.
>
> Question 4. I think the introduction part can make a clear, or informal definition of what is generalization under the well-separated classification task.
>
> We will do this in the revised version.

---

> > ### Comment · Reviewer_R3pV · 2024-08-12
> >
> > Thank you for the clarification, I'll keep the score.

---

> > > ### Author Response · Authors · 2024-08-13
> > >
> > > Thank you again for your nice review !

---

### Official Review · Reviewer_dviZ · 2024-07-12

**Soundness:** 2
**Presentation:** 3
**Contribution:** 3
**Rating:** 6
**Confidence:** 4

**Summary:**

- The authors study the memorization capacity of ReLU networks and its generalization theory for i.i.d. datasets over the compact domain, under a binary classification setting.
- They propose two different memorization capacity upper bound of ReLU networks. One bound depends on the size $N$ of the training dataset ($O(\sqrt{N}\ln{N})$) and the other bound is independent of $N$. Notably, the second bound implies that even with an arbitrarily large dataset sampled i.i.d., a constant number $N_{\mathcal{D}}$ of the parameters is enough to memorize the dataset, where $N_{\mathcal{D}}$ depends only on the true data distribution $\mathcal{D}$.
- They show that memorization is not enough for generalization with concrete examples. First, a generalizable memorization network must have a width of at least $n$ (the data dimension). Second, for almost every data distribution, there exists a non-generalizable memorization network with $O(\sqrt{N}\ln{N})$ parameters ($N$: size of training dataset).
- They study the sample complexity for $(\\epsilon,\\delta)$-PAC-generalizable memorization networks. First, they provide a lower bound of sample complexity for general memorization networks. Second, they provide an upper bound of sample complexity for particular memorization networks with at most $N_{\mathcal{D}}$ parameters. Both bounds are tight in terms of $N_{\mathcal{D}}$ up to logarithmic factors.
- Lastly, they study the algorithm constructing a PAC-generalizable memorization network in poly$(N)$ time.

**Strengths:**

- S1. As far as I know, this is the first work studying the generalizability of the memorization ReLU network, under the binary classification task.
- S2. In terms of ReLU networks, they discover two novel complexity terms $N_{\mathcal{D}}$ (smallest number of parameters to memorize i.i.d. dataset (of any size) from $\mathcal{D}$) and $S_{\mathcal{D}}$ (”efficient memorization sample complexity”; minimum size of the nearby set of  $\mathcal{D}$) which depend only on the true data distribution.
- S3. The paper is well-written and easy to follow.

**Weaknesses:**

- W1. Compactness of the input domain
    - Line 47 of the introduction says the paper will consider the open domain $\mathbb{R}^n$ of input. However, only the compact input domain $[0,1]^n$ is considered throughout the paper. It is unclear whether the results on the compact domain can be easily extended to the  $\mathbb{R}^n$ case. In the approximation literature, it is important whether the input domain is compact or not. Thus, the authors should mention whether the extension to a larger domain $\mathbb{R}^n$ is easy. Or, it would be better to replace $\mathbb{R}^n$ with a compact input domain in the introduction.
- W2. Randomness in Theorem 4.1
    - Theorem 4.1 contains a probabilistic argument. However, it is unclear where the randomness comes from, with the theorem statement alone.
- W3. Efficiency of memorization algorithm in Theorem 4.3
    - It would be meaningful to mention whether the memorization algorithm in Theorem 4.3 is efficient.
- W4. Parameter complexity of efficient memorization algorithm in Theorem 7.3
    - It would be meaningful to clarify how many parameters the network constructed by the efficient memorization algorithm in Theorem 7.3 should have.
- W6. Minor comments on notations/typos (There are so many typos…)
    - Title: “Generalizablity” → “Generalizability”
    - Line 60: “…, (2) of Theorem 1.1…”
    - Please use $\overline{O}$ for upper bounds, $\overline{\Omega}$ for lower bounds, and $\overline{\Theta}$ for tight bounds (up to logarithmic factors).
    - Section 3.1: “$X_0 = x$” and “$n_0 = n$” must be explicitly mentioned.
    - As far as i know, $\mathbb{R}\_{+}$ and $\mathbb{Z}\_{+}$ usually include zero. To exclude zero, $\mathbb{R}\_{++}$ (or $\mathbb{R}\_{>0}$) and $\mathbb{Z}\_{++}$ (or $\mathbb{Z}\_{>0}$) are better to use.
    - Line 250: “generaizable” → “generalizable”
    - Line 274: “The density of distribution $\mathcal{D}$”
    - Line 279: “$A_{\mathcal{D}} (\mathcal{F}) \le 0.51$”
    - Line 291: “generazable” → “generalizable”
    - Line 309: “generalizability,.” → “generalizability.”
    - Line 321: “$N_D$” → “$N_{\mathcal{D}}$” (calligraphic D)
    - Line 325: “bpund” → “bound”
    - Line 337: “$\mathcal{D}_{\rm tr} \sim \mathcal{D}^N$
    - Lines 340 and 472: “Proof Idea.” is not in a bold font.
    - Line 367: What does it mean by “a subset of $(x,y)\sim \mathcal{D}$”?
    - Line 382: Is “$S_{\mathcal{D}} \ge N_{\mathcal{D}}^2$” really true? I guess it should be “$S_{\mathcal{D}} \gtrsim N_{\mathcal{D}}^2$”, ignoring logarithmic factors.
    - Line 619: “Uisng” → “Using”
    - Step Three in the proof of Theorem 4.3 (Appendix C): Please make clear which theorem of [61] you refer to.
    - Line 1052: “Tree” → “Three”
    - Section F: a very non-standard notation is used here: “$C^m_n$” means a binomial coefficient “$\binom{n}{m}$”! This makes the proof really hard to read. Please consider changing the notation.
    - Part 2 in the proof of Theorem 6.1: I guess $12v_1 \le N_{\mathcal{D}} \le 4v_1 \sqrt{k} \ln(\sqrt{k})$ is required for the later use, rather than $3 \le N_{\mathcal{D}} \le 4v_1 \sqrt{k} \ln(\sqrt{k})$. This can be fixed with minor changes.
    - Lines 1148-1149: Make clear that $S(\mathcal{D}) \subset [k]^q$.
    - Equation (4) and the rest of the proof of Theorem 6.1: I think $q^k$ should be replaced with $k^q$. See lines 1166, 1167, 1195, and 1196.
    - Line 1189: “Because” → “This is because”
    - Lines 1189-1192: several “$S_{ss}(\mathcal{D},Z)$”s are written in wrong symbols.
    - Line 1213: “Here, $E\_{\mathcal{D}}(\mathcal{F}) = \mathbb{E}\_{(x,y)\sim \mathcal{D}} [I(\operatorname{Sgn}(\mathcal{F}(x))=y)]$,  $E\_{\mathcal{D}\_{\rm tr}}(\mathcal{F}) = \frac{1}{N} \sum_{(x,y) \in \mathcal{D}\_{\rm tr}} I(\operatorname{Sgn}(\mathcal{F}(x))=y)$ ”.
    - Line 1224: “$n$” → “$m$”
    - Line 1249: “similarto” → “similar to”
    - and many more…

**Questions:**

- Q1. Architectural constraint: ReLU network
    - A clear limitation of this work is that it is limited to a fully-connected ReLU network.
    - Can you imagine how the change in activation function will affect the memorization capacity, $N_{\mathcal{D}}$, and $S_{\mathcal{D}}$?
- Q2. Sample complexity bound gap of memorization algorithms in Section 6
    - Currently, there is an $\epsilon^{-2}$ gap between the upper and lower sample complexity for the generalization of memorization networks. I guess this is quite huge. Do you think this gap can be reduced?
- Q3. Proof of Corollary 6.4
    - The proof is too short. How did you get to the statement of Corollary 6.4 by plugging $k=2^{[\tfrac{n}{\lceil c^2 \rceil}]}$? It seems unclear to me.
- Q4. Algorithm in Theorem 7.3 as an initialization
    - Empirically, what if you run SGD (or other standard optimizers) with logistic loss starting from the network parameters constructed by the algorithm of Theorem 7.3? Will it be better than the result of SGD from random initialization?
- Q5. Generalizability of memorization v.s. non-memorization networks
    - I know this question is somewhat out of topic, but can you discuss the generalizability of memorization and non-memorization networks? Can a non-memorization network generalize better than some memorization networks? If so, when can it happen?

**Limitations:**

The authors have discussed the limitation in Section 8 (Conclusion). The major limitation is that it is unclear how to compute the numerical complexities $N_{\mathcal{D}}$, $S_{\mathcal{D}}$.

---

> ### Author Rebuttal · Authors · 2024-08-05
>
> The authors thank the reviewer for the valuable and insightful questions and hope that we have answered these questions satisfactorily. Also thank you for pointing out the typos in our writing, we will correct them in future versions.
>
> 1. Question (W1): Compactness of the input domain.
>
> The current proofs are for the compact input domain. We will change $R^n$ to $[0,1]^n$ in the Introduction Section.  After evaluating the proofs, we find that only Theorems 5.1, 5.3, 6.1 still hold when we consider the distributions in $ R^n$. On the other hand, the extension to a larger domain $R^n$ is not a good idea, because networks with finite parameters can not approximate many of distributions over $R^n$, so that it is impossible to study guaranteed generalization theorems under such infinite domain, and th4.1, th4.3, th6.5 and th7.3 are incorrect on $R^n$. So, if we need to study about the distributions on $R^n$, we need to add more constraints to the distributions.
>
> 2. Question (W2): Where comes the randomness in Theorem 4.1?
>
> This randomness comes from a technique of memorization: in the construction of a memorization network, we require a special vector to compress data into one dimension, but this vector is not easy to calculate directly. However, every time we randomly select a vector, there is a non-zero (0.5) chance of selecting the required vector, and the randomness comes from this random selection. We have described this in lines 220-223 and 648-649 (and the proof above lines 648-649) of the paper. We will make this clearer in the revised version of the paper.
>
> 3. Question (W3): Efficiency of memorization algorithm in Theorem 4.3?
>
> Theorem 4.3 is an existence result that mainly demonstrates the existence of $N_D$. The efficiency of the memorization algorithms are given in Theorem 6.5 and Theorem 7.3. If $para(L(D_{tr}))\le N_D$, (2) of Theorem 6.5 shows that such memorization algorithms are not efficient.  If $N>O(S_D)$, then Theorem 7.3 shows that there exists an efficient memorization algorithm.
>
> 4. Question (W4): Parameter complexity of efficient memorization algorithm in Theorem 7.3.
>
> The number of parameters are affected by the data distribution. In the worst-case, the network will have $O(N^2)$ parameters. As said in line 103-107, to ensure generalization, we abandoned many classic techniques, which resulted in an increase of parameters. We will add this information in the revised version.
>
> 5. Question (Q1): Architectural constraint: ReLU network
>
> The reason why we consider the fully connected ReLU network is because this type of network is very classic, and it has been widely applied in previous related research.  For other structure networks like CNN or with other activation functions, the conclusions in this paper may be different. The main results in the paper are based on the expressive power of the fully-connected ReLU network, such as VCdim, the ability to fit discrete points and so on, so after replacing the network structure, we can still use the ideas in the paper to re-establish the $N_D$, $S_D$ and so on. This is certainly an important direction for future research. We will mention this in the revised version of the paper.
>
> 6. Question (Q2): There is an $\epsilon^2$ gap between the upper and lower.
>
> You are right. We want to mention that the lower and upper bounds are at the same order of magnitude for the number of samples $N^2_D$. We believe that it is still room to improve the factor $(1-2\epsilon-\delta)$ in the lower sample complexities. But currently, we do not achieve this yet.
>
> 7. Question (Q3): How to get the proof of Corollary 6.4?
>
> According to Lemma E.2, for any $n,c$, we can find $2^{n/[c^2]}$ points with distance $c$ in $[0,1]^n$. Then we take suitable $n,c,k=2^{n/[c^2]}$ to make the contents in lines 1131 to 1133 of the proof of Theorem 6.1 valid. Finally, follow the part 2-4 in the proof of Theorem 6.1, according to line 1159-1162 and the begining of part 4, we can prove Corollary 6.4. We will add more details about the proof in the future.
>
> 8. Question (Q4): Algorithm in Theorem 7.3 as an initialization.
>
> We can only now provide some simple experiments. In the case of a small sample size, using our method as initialization for training may be helpful; while in the case of a large sample size, the training with random initialization can reach a very high level, so using our method for training does not bring significant improvement.
>
> We do binary classification problems on CIFAR-10 dataset by considering 5 pairs binary classification problems: samples with label 2i and label 2i+1, where $i=0,1,2,3,4$. Randomly select 1000 samples from each category as the training set. We compare four methods: train a DNN, train a VGG-16, ours, training with our methods as initialization. The accuracies are given below.
>
> |Pair of Label|CNN|DNN|Ours|Ours+Training|
> |0,1|0.95|0.85|0.76|0.90|
> |2,3|0.84|0.73|0.59|0.75|
> |4,5|0.87|0.72|0.65|0.77|
> |6,7|0.98|0.84|0.69|0.84|
> |8,9|0.92|0.82|0.68|0.84|
>
> It is easy to see that:
> (1): Our methods cannot go beyond training.
> (2): If the dataset is relatively small, using our method as an initialization can help improve accuracy.
> (3): Since the paper focuses on a fully connected network structure, its performance is inferior to that of CNN in image classification tasks.
>
> 9. Question (Q5): Generalizability of memorization v.s. non-memorization networks.
>
> We believe that a non-memorization network can generalize better than some memorization networks mainly in the following situation: The data set contains significant bad data, such as outliers. One weakness of memorization algorithm is that it must fit all the data, and the network will learn non-existent features from outliers, thus harming generalization. In this situation, a non-memorization algorithm can ignore the bad data in certain sense and thus work better.

---

> > ### Comment · Reviewer_dviZ · 2024-08-07
> > **Remaining Questions**
> >
> > Thank you for your kind reply. I am happy with most of the answered questions/comments.
> > I have to take some time to digest the responses for W1, W3, W4, and Q3. I will come back again after carefully checking these responses.
> >
> > Unfortunately, the two questions in my original review are not answered yet, mainly inside the "minor typo/comment" part. Let me bring them here again:
> >
> > 1. Line 367: What does it mean by “a subset of $(x,y)\sim \mathcal{D}$”?
> > 2. Line 382: Is “$S_{\mathcal{D}} \ge N_{\mathcal{D}}^2$” really true? I guess it should be “$S_{\mathcal{D}} \gtrsim N_{\mathcal{D}}^2$”, ignoring logarithmic factors.
> >
> > I am looking forward to further responses to these unanswered questions. Also, if there are missing details for the response above due to the space limit, please leave them as a comment here, then I'll happily read them all.

---

> > > ### Author Response · Authors · 2024-08-07
> > >
> > > I'm glad you replied to us, taking this opportunity, we would like to provide some more detailed responses.
> > >
> > > For Question (W3): This proof of th4.3 uses the method in [61] and gives an algorithm with a large $N'_D$, and such algorithm is effective. But if we want to get an algorithm with a smaller $N_D'$, it's quite difficult, and it is NPC when $N_D'=N_D$.
> > >
> > > For Question (Q4): Due to formatting issues, the table format is not easy to read, we show it at here:
> > >
> > > |Pair of Label|CNN|DNN|Ours|Ours+Training|\\\
> > >  |0,1|0.95|0.85|0.76|0.90|\\\
> > >  |2,3|0.84|0.73|0.59|0.75| \\\
> > > |4,5|0.87|0.72|0.65|0.77|\\\
> > >  |6,7|0.98|0.84|0.69|0.84| \\\
> > > |8,9|0.92|0.82|0.68|0.84|\\\
> > >
> > > For line 367: We want to express a subset composed by samples in a distribution. Strictly speaking for it, if the distribution $D$ is defined on $A\subset [0,1]^n\times\{-1,1\}$, then it means a subset composed of samples in $A$. For example, $D$ is defined as: $P_{(x,y)\sim D}((x,y)=(x_0,-1))=P_{(x,y)\sim D}((x,y)=(x_1,1))=0.5$, then $A=\\{(x_0,-1),(x_1,1)\\}$.
> > >
> > > For line 382: You are right, we mainly want to express a possible relationship between $N_D$ and $S_D$, our expression is not strict, your recommendation is correct.

---

> > > > ### Comment · Reviewer_dviZ · 2024-08-10
> > > > **Thank you**
> > > >
> > > > Thank you for your further reply.
> > > >
> > > > * I guess they meant by Lemma F.1 (instead of Lemma E.2) in #7 of the author's response. Also, in the sentence "Then we take suitable n, c,... valid.", I am curious about what values of $n$ and $c$ are indeed suitable. Thus, I hope the authors will add much more detail to the proof of Corollary 6.4.
> > > >
> > > > Given all the promises about further revisions, I am satisfied with the author's responses. I retain my score.

---

> > > > > ### Author Response · Authors · 2024-08-11
> > > > >
> > > > > Thank you for your nice insights. About how to take the value of $n,c$  in the proof of Corollary 6.4, using the notation in the proof of th6.1, for the situation of $1-2\epsilon-\delta>0$, we can take $n=3[12v_1/(1-2\epsilon-\delta)]+3$ and $c=1$ to make them satisfies line 1130-1133; when $1-2\epsilon-\delta\le0$, the Corollary 6.4 is obvious stand for any $n$ and $c$, because the lower bound is a negative when $1-2\epsilon-\delta\le0$. We will give more detailed to the proof of Corollary 6.4 in the future version.

---

### Decision · Program_Chairs · 2024-09-25

**Decision:**

Accept (poster)

**Comment:**

This paper presented a novel and extensive theoretical studies of the generalization capabilities of memorization networks that interpolate a finite training set. The results provide valuable new insights into the understanding of key properties of neural network generalization.